# Description and evaluation of the community aerosol dynamics model MAFOR v2.0

Matthias Karl[1], Liisa Pirjola[2,11], Tiia Grönholm[3], Mona Kurppa[3], Srinivasan Anand[4], Xiaole Zhang[5], Andreas Held[6], Rolf Sander[7], Miikka Dal Maso[8], David Topping[9], Shuai Jiang[10], Leena Kangas[3], and Jaakko Kukkonen[3,12]

[1]Chemistry Transport Modelling, Helmholtz-Zentrum Hereon, Geesthacht, Germany
[2]Department of Physics, University of Helsinki, Helsinki, Finland
[3]Atmospheric Composition Research, Finnish Meteorological Institute, Helsinki, Finland
[4]Health Physics Division, Bhabha Atomic Research Centre, Mumbai, India
[5]Institute of Environmental Engineering (IfU), ETH Zürich, Zürich, Switzerland
[6]Environmental Chemistry and Air Research, Technische Universität Berlin, Berlin, Germany
[7]Air Chemistry Department, Max-Planck Institute of Chemistry, Mainz, Germany
[8]Aerosol Physics, Faculty of Engineering and Natural Sciences, Tampere University, Tampere, Finland
[9]Department of Earth and Environmental Science, University of Manchester, Manchester, UK
[10]School of Information Science and Technology, University of Science and Technology of China, Hefei, Anhui, China
[11]Department of Automotive and Mechanical Engineering, Metropolia University of Applied Sciences, Vantaa, Finland
[12]Centre for Atmospheric and Climate Physics Research, and Centre for Climate Change Research, University of Hertfordshire, Hatfield, UK.

*Correspondence to:* M. Karl (matthias.karl@hereon.de)

**Abstract.** Numerical models are needed for evaluating aerosol processes in the atmosphere in state-of-the-art chemical transport models, urban-scale dispersion models and climatic models. This article describes a publicly available aerosol dynamics model MAFOR (Multicomponent Aerosol FORmation model; version 2.0); we address the main structure of the model, including the types of operation and the treatments of the aerosol processes. The main advantage of MAFOR v2.0 is the consistent treatment of both the mass- and number-based concentrations of particulate matter. An evaluation of the model is also presented, against a high-resolution observational dataset in a street canyon located in the centre of Helsinki (Finland) during an afternoon traffic rush hour on 13 December 2010. The experimental data included measurements at different locations in the street canyon of ultrafine particles, black carbon, and fine particulate mass $PM_1$. This evaluation has also included an intercomparison with the corresponding predictions of two other prominent aerosol dynamics models, AEROFOR and SALSA. All three models fairly well simulated the decrease of the measured total particle number concentrations with increasing distance from the vehicular emission source. The MAFOR model reproduced the evolution of the observed particle number size distributions more accurately than the other two models. The MAFOR model also predicted the variation of the concentration of $PM_1$ better than the SALSA model. We also analysed the relative importance of various aerosol processes based on the predictions of the three models. As expected, atmospheric dilution dominated over other processes; dry deposition was the second most significant process. Numerical sensitivity tests with the MAFOR model revealed that the uncertainties associated with the properties of the condensing organic vapours affected only the size range of particles smaller than 10 nm in diameter. These uncertainties do not therefore affect significantly the predictions of the whole of the number size distribution and the total

number concentration. The MAFOR model version 2 is well documented and versatile to use, providing a range of alternative parametrizations for various aerosol processes. The model includes an efficient numerical integration of particle number and mass concentrations, an operator-splitting of processes, and the use of a fixed sectional method. The model could be used as a module in various atmospheric and climatic models.

## 1 Introduction

Urban environments can contain high concentrations of aerosol particle numbers as a result of the emissions from local sources, most frequently vehicular traffic (Meskhidze et al., 2019), ship traffic (Pirjola et al., 2014), airports (Zhang et al., 2020), industrial emissions (Keuken et al., 2015) or from all of these sources (Kukkonen et al., 2016). The majority of the urban aerosol particles – in terms of number concentration – are ultrafine particles (UFP), having aerodynamic diameters less than 100 nm (e.g., Morawska et al., 2008). UFPs exhibit high deposition efficiency, large active surface area and are often associated with toxic contaminants, such as transition metals, polycyclic aromatic hydrocarbons, and other particle-bound organic compounds (Bakand et al., 2012). Owing to their small size, inhaled UFPs can penetrate deep in the human lungs, deposit in the lung epithelium and translocate to other organs. Long-term exposure to UFP negatively affects cardiovascular and respiratory health in humans (Wichmann and Peters, 2000; Evans et al., 2014; Breitner et al., 2011). Sub-micrometre soot particles emitted from diesel engines, mainly consisting of light-absorbing black carbon (BC), other combustion-generated carbonaceous materials and condensed organics (Kerminen et al., 1997), often dominate the absorption of solar light by aerosols thereby influencing the visibility in urban areas (Hamilton and Mansfield, 1991). The physico-chemical characteristics of UFP and their dynamic evolution also play an important role in changing the optical properties as they quickly coagulate with each other and larger particles, or grow by the condensation of vapours, into the size range of cloud condensation or ice nuclei, affecting the indirect climate effects of atmospheric aerosol by regulating cloud formation, cloud albedo, and changing the precipitation processes (Andreae and Rosenfeld, 2008).

In urban areas, the temporal variation and spatial inhomogeneity of both the particle number (PN) and particulate matter (PM) concentrations are closely linked to local meteorology and traffic flows (e.g., Kumar et al., 2011; Singh et al., 2014; Kukkonen et al., 2018). For example, particle concentrations in street canyons can be several times higher than in unobstructed locations. PN concentrations in a street canyon depend upon traffic characteristics, building geometry, turbulence that can be induced by traffic, the prevailing winds, and atmospheric stability (e.g., Kumar et al., 2009). However, measurements of particle number and size distributions in urban environments are scarce and the complexity of the urban environment prevents extrapolation from single point measurements to the wider urban area.

A key question in applying aerosol process models is the scarcity of reliable and comprehensive emission data. Kukkonen et al. (2016) presented an emission inventory for particulate matter numbers (PN) in the whole of Europe, and in more detail in five target cities. The modelled PN concentrations (PNC) were compared with experimental data on regional and urban scales. They concluded that it is feasible to model PNC in major cities with reasonable accuracy; however, there were major challenges especially in the evaluation of the emissions of PNC. The rapid transformation of freshly emitted aerosol particles

by condensation/evaporation, coagulation and dry deposition was also found to pose challenges for dispersion modelling on the urban scale.

A substantial fraction of the state-of-the-art chemical transport models contain treatments of aerosol processes (e.g., Kukkonen et al., 2012). However, only a limited number of urban dispersion models can deal with PN dispersion and processes affecting the particle size distribution, especially addressing the modelling of the dispersion of particles in complex urban terrain, such as street canyons (Gidhagen et al., 2004). This has been partly caused by the large effort for model development that is necessary for implementing size-resolved aerosol and particle dynamics models to urban modelling systems.

Modelling of particle transformation in parallel to plume dispersion is necessary to represent the evolution of the particle number and mass size distribution from the point of emission to the point of interest. Since the particle size and composition evolve on a short timescale, it is important to examine the evolution near the source at high spatial and temporal resolution. Modelling studies examining the evolution of particle emissions have used zero-dimensional (0-D) models (Vignati et al., 1999; Pohjola et al., 2003, 2007; Karl et al., 2016), one-dimensional (1-D) models (Fitzgerald et al., 1998; Capaldo and Pandis, 2001; Boy et al., 2006), two-dimensional (2-D) models (Roldin et al., 2011) and three-dimensional (3-D) models (Gidhagen et al., 2005; Andersson et al., 2015). Jacobson and Seinfeld (2004) have modelled the near-source evolution of multiple aerosol size distributions with a 3-D chemistry-transport model (CTM) over a high-resolution limited-area grid, however only a few minutes were simulated. Long range aerosol transport models coupled with numerical weather prediction models can be used to trace the mass and number concentrations of aerosols from point source emissions at the surface and different vertical levels (Fountoukis et al., 2012; Sarkar et al., 2017; Chen et al., 2018). The size distribution of emissions in large-scale models can only be approximated because they need to take into account the size distribution of the primary emitted particles at the point of emissions and the ageing processes that occur at sub-model grid scales (Pierce et al., 2009). Higher temporal resolution is therefore necessary to better characterise primary and secondary particle sources. Computational Fluid Dynamics (CFD) models, notably building-resolving Large Eddy Simulation (LES) models are advantageous in simulating the air flow and dispersion of air pollutants in urban areas. Until now, only a few LES models include modules for treating aerosol particles and their dynamics (Tonttila et al., 2017; Kurppa et al., 2019; Zhong et al., 2020). The implementation of aerosol dynamics into LES models increases their computational load tremendously.

Lagrangian approaches to the fluid flow are often employed in 0-D models that combine a vehicular plume model with an aerosol dynamics model in order to assess the impacts of coagulation, condensation of water vapour, and plume dilution of the particle number size distribution (e.g., Pohjola et al., 2007). On the urban scale, application of Lagrangian models is limited because of the large variability of emission sources and because they do not account for different wind speed or direction at different altitudes. However, the Lagrangian approach is advantageous for the examination of exhaust plumes in street environments, as it allows for the inclusion of more details on the representation of the aerosol dynamics and gas-phase chemistry than would be possible in a 3-D CTM. The traffic exhaust plume can be considered as an isolated air parcel moving with the fluid flow, without mixing with other air parcels on the neighbourhood scale.

The Multicomponent Aerosol FORmation model MAFOR (Karl et al., 2011) is a 0-D Lagrangian type sectional aerosol process model, which includes multiphase chemistry in addition to aerosol dynamics. It has been originally developed to

overcome the limitations of monodisperse models with respect to the simulation of continuous new particle formation in the marine boundary layer. Later, the model has been extended with a module for dilution of particles in urban plumes with particles from background air (Karl et al., 2016). The aerosol dynamics module of MAFOR simultaneously solves the time evolution of particle number, mass concentration and chemical composition distributions in a consistent manner and with high accuracy, including the cases in which the condensation and coagulation processes are competing.

The aerosol dynamics in MAFOR are coupled to a detailed gas-phase chemistry module, which offers full flexibility for inclusion of new chemical species and reactions. There are only a few other aerosol dynamics models for use in atmospheric studies that integrate gas-phase chemistry together with aerosol processes as a function of time. Examples are ADCHEM (Roldin et al., 2011) and AEROFOR (Pirjola, 1999; Pirjola and Kulmala, 2001) that both use the kinetic code developed by Pirjola and Kulmala (1998), originally representing a modified EMEP chemistry scheme (Simpson, 1992). An advantage of AEROFOR is that it allows for multicomponent condensation to an externally or internally mixed particle population. AEROFOR has been applied to study aerosol dynamics and particle evolution under different atmospheric conditions such as arctic, boreal forest, and marine environments (e.g., Pirjola et al., 1998; 2002; 2004; Kulmala et al., 2000) as well as for the study of diesel exhaust particles under laboratory conditions (Pirjola et al., 2015). However, the model has limitations with respect to the treatment of particle phase chemistry and does not solve mass concentration distributions as a function of time.

MAFOR has been proven to be particularly useful for studying changes of the emitted particle size distributions by dry deposition (to rough urban surfaces), coagulation processes, considering the fractal nature of soot aggregates, and by condensation/evaporation of organic vapours emitted by vehicular traffic. The model is very versatile in its application: due to its modular structure, the model user can switch on/off the different aerosol processes or use alternative parameterizations for the same process, depending on the research question.

The first objective of this paper is to present the model's structure, the treatment of aerosol processes, the coupling to multiphase chemistry, and the main updates compared to the first publication of the model (version 1, in Karl et al., 2011). The second objective of the paper is the evaluation of the model performance of MAFOR version 2 with respect to its ability of predicting particle and mass number size distributions. Several of the new features of MAFOR version 2 were investigated in three different numerical scenarios and compared to reference data. Specifically, they included the evaluation of (1) the model's sectional representation of the aerosol size distribution in a scenario of new particle formation in urban areas ("Case 1"); (2) Brownian coagulation under the condition of continuous injection of nanoparticles ("Case 2"); and (3) the dynamic treatment of semi-volatile inorganic gases by condensation and dissolution ("Case 3"). The description and results of the numerical scenarios are given in the Supplementary Materials.

The main performance evaluation of MAFOR version 2 is addressed in a real-world scenario of a street canyon environment, in comparison with other aerosol process models and experimental data. In combination with the plume dispersion module, MAFOR version 1 has previously been evaluated against PN measurements at a motorway (Keuken et al., 2012) and against observed particle size distributions in the exhaust plumes of passenger ships arriving or leaving a ferry terminal (Karl et al., 2020). The real-world scenario in the present study focuses on the application of MAFOR version 2 for plume dispersion in a street canyon, based on a published dataset of observations (Pirjola et al., 2012); from now on referred to as "Urban

Case". Results from the MAFOR model are inter-compared to the aerosol process models AEROFOR and SALSA (Kokkola et al., 2008). The relative importance of aerosol dynamic processes in this scenario is evaluated for the three models, using the dispersion-coagulation model LNMOM-DC model (Anand and Mayya, 2015; Sarkar et al., 2020) as reference for the relevance of coagulation. The performance of the aerosol dynamics models is evaluated based on defined criteria, such as statistical performance indicators, computational demand, and number of model output variables.

Section 2 describes the structure of the community aerosol dynamics model MAFOR version 2, the included physical and chemical processes, and their numerical solution. In addition, previous applications of the model are summarized and the new setup for modelling of the particle evolution in a street canyon is introduced. Sect. 3 presents the methods and the experimental data that are used for evaluation of the model in the Urban Case scenario. Sect. 4 discusses the results from the evaluation and from the comparison with other aerosol dynamics models.

## 2 Model description

MAFOR v2.0 is available as an open source community aerosol model. The publication of MAFOR v2.0 as a community model is driven by the intention to provide both newcomers and experts in aerosol modelling with an easy-to-use stand-alone aerosol box model. A consortium of aerosol scientists guides the development of the community model. For application in atmospheric studies, apart from the SALSA (Kokkola et al., 2008) and PartMC (Riemer et al., 2009), there exist to date no other aerosol dynamics model that is available as open source code. In recent years, several aspects of the MAFOR model have been revised and updated with aerosol process parameterizations published in the peer-reviewed literature. The main new features of MAFOR v2.0 compared to the original version (MAFOR v1.0, Karl et al., 2011) are:

1. Coupling to the chemistry sub-model MECCA (Module Efficiently Calculating the Chemistry of the Atmosphere) of the community atmospheric chemistry box model CAABA/MECCA v4.0 (Sander et al., 2019).

2. Extension of the Brownian coagulation kernel to consider the fractal geometry of soot particles, van der Waals forces and viscous interactions.

3. Inclusion of new nucleation parameterizations for neutral and ion-induced nucleation of $H_2SO_4$-water particle formation (Määttänen et al., 2018a,b) and $H_2SO_4$-water-$NH_3$ ternary homogeneous and ion-mediated particle formation (Yu et al., 2020).

4. The Predictor of Nonequilibrium Growth (PNG) scheme (Jacobson, 2005a) was implemented and linked with the thermodynamic module MESA (Zaveri et al., 2005b) of the MOSAIC (Model for Simulating Aerosol Interactions and Chemistry; Zaveri et al., 2008), to enable dynamic dissolution and evaporation of semi-volatile inorganic gases.

5. Absorptive partitioning of organic vapours to form secondary organic aerosol (SOA), following the formulation of the two-dimensional volatility basis set (2-D VBS; Donahue et al., 2011 within the framework of dynamic condensation/evaporation.

The model can be run in three different types of operation: (1) Simulation of an air parcel extending from the surface to the height of the planetary boundary layer (PBL) for multiple days along a given air mass trajectory, or as a box model at a single geographic location, assuming a well-mixed boundary layer and clear sky conditions. As a variation of this operation type, the multiphase chemistry during a fog cycle with pre-defined liquid water content and pH value of the fog/cloud can be simulated; (2) Chamber experiment simulation, assuming homogeneous mixing of constituents in a defined air volume for a given chamber geometry, considering sink terms and source terms of gases to and from chamber walls, deposition of particles to chamber walls, and constant dilution by replenishment of air; (3) Plume dispersion simulation that considers the evolution of the particle number and mass composition distributions in a single exhaust plume, along one dimension in space, by treating the transformation of emitted gases, condensing vapours and particles concurrent with the dilution with background air during the spread of the plume volume. A special case is the simulation of dilution and ageing in a laboratory system for diesel exhaust, using a simple parameterization for the dilution and cooling processes as described in Pirjola et al. (2015).

In the following sections, a detailed description of the physical and chemical processes and their numerical solution will be given. The focus is on presenting the new features that have been implemented after version 1.0. We begin with a review of the currently available aerosol process models in Sect 2.1. Sect. 2.2 gives an overview of the structure and workflow of the MAFOR model. Sect. 2.3 describes the multiphase chemistry processes and each of the individual aerosol transformation processes in the model. Sect 2.4 explains the dynamic treatment of semi-volatile inorganic gases in more detail. Sect. 2.5 presents SOA formation by absorptive partitioning of organic vapours according to the 2-D VBS. The numerical solution of the aerosol dynamics in the model is given in Sect. 2.6. A brief overview of previous applications of the model in plume dispersion scenarios is given in Sect. 2.7.

Throughout the paper, index $q$ ($q = 1, \ldots, N_C$) is used to denote chemical constituents, with $N_C$ being the number of constituents in the aerosol. Index $i$ ($i = 1, \ldots, N_B$) is used to denote the size section of the particle distribution and $N_B$ is the number of size sections (bins). A list of acronyms and mathematical symbols is given in Appendix A.

### 2.1 Review of current aerosol process models

Table 1 provides a comparison of selected aerosol dynamics models that are currently used in studies of atmospheric aerosols. According to their representation of the particle size distribution, aerosol dynamics models can be divided into sectional, modal, monodisperse and moment models (refer to Whitby and McMurry (1997) for detailed review).

Sectional models (Gelbard and Seinfeld, 1990; Warren and Seinfeld, 1985; Jacobson and Turco, 1995; Pirjola and Kulmala, 2001; Korhonen et al., 2004) place a grid on the independent variable space (e.g. particle diameter or volume). The aerosol size distribution is approximated by a finite number of size sections (bins) whose locations on the grid can either vary with time or be fixed.

Modal models (Wright et al., 2001; Vignati et al., 2004) represent the particle distribution as a sum of modes, each having a lognormal or similar size distribution, typically described by mass, number and width. Modal size distributions can be solved very efficiently, which makes them favourable candidates for global 3-D CTM models. However, the accuracy of the modal method is lower compared to the sectional method, especially if the standard deviation (width) of the modes is treated as

constant (Zhang et al., 2002). In monodisperse models (Pirjola et al., 2003), all particles in each mode have the same size, but can have different composition.

Moment models (McGraw, 1997) track a few low-order moments of the particle population, but do not explicitly resolve the size distribution. Anand and Mayya (2009) have developed a formalism based on an analytical solution of the coagulation-diffusion equation for estimating the survival fraction of aerosols in dispersing puffs and plumes under the assumption of an initially Gaussian distributed particle number concentration and spatially separable size spectra. The parameterization scheme has been further developed and is termed "Log Normal Method Of Moments – Diffusion Coagulation" (LNMOM-DC) model, enabling the simultaneous treatment of aerosol coagulation and dispersion in an expanding exhaust plume.

The sectional aerosol dynamics model MAFOR allows for multicomponent condensation of vapours [sulphuric acid ($H_2SO_4$), methane sulfonic acid (MSA), ammonia ($NH_3$), amines, nitric acid ($HNO_3$), hydrochloric acid (HCl), water ($H_2O$) and nine different organic compounds] to an internally mixed aerosol that includes all atmospherically relevant aerosol constituents, i.e. sulfate, ammonium, nitrate, methane sulfonate ($MSA_p$), sea salt, soot, primary biological material, and mineral dust. The assumption of internally mixed particles, i.e. that all particles in the same size bin have same chemical composition, lowers the accuracy in cases of high humidity in air, because the ability to take up water can vary considerably for particles of the same size that have different composition (Korhonen et al., 2004). However, handling multivariate distributions that allow for same-sized particles with different hygroscopic properties involve large storage and computation requirements. The particle-resolved model PartMC-MOSAIC (Riemer et al., 2009; Tian et al., 2014) stores the composition of many individual aerosol particles (typically about $10^5$) within a well-mixed computational volume. The computational burden is reduced by simulating the coagulation stochastically, assuming coagulation events are Poisson distributed with a Brownian kernel.

The size-segregated aerosol model UHMA (Korhonen et al., 2004), another sectional aerosol dynamics model, has demonstrable good performance in reproducing new particle formation, and solves the evolution of particle number and surface size distribution together with the composition distribution. In UHMA, the discretization of particle sizes are based on the volume of the particle core. A shortcoming of UHMA is that it does not explicitly solve the mass concentration change of individual aerosol components with time; whereas MAFOR takes into account that the condensation or evaporation of an individual component results in the growth/shrinkage of the (total) mass concentration size distribution, affects the total aerosol mass, and moves the component's mass concentration distribution on the diameter coordinate.

M7 (Vignati et al., 2004) and SALSA (Kokkola et al., 2008), owing to their computationally efficiency – have been implemented into 3-D aerosol-climate models (Bergman et al., 2012). SALSA is a sectional aerosol module, developed with the specific purpose for implementation in large scale models. Implementation examples are UCLALES-SALSA (Tonttila et al., 2017), in PALM (Kurppa et al., 2019) and in ECHAM-HAMMOZ (Kokkola et al., 2018). The focus of the implementation of SALSA is the description of the aerosol processes with sufficient accuracy, which is important for understanding the aerosol-cloud interactions and their impacts on global climate. SALSA includes aerosol microphysical processes nucleation, condensation, hydration, coagulation, cloud droplet activation and oxidation of sulphur dioxide ($SO_2$) in cloud droplets. The main advantage of SALSA is that particle size bin width does not have to be fixed and lower size resolution can be used in the particle size range less affected by microphysical processes.

**Table 1.** Comparison of selected zero-dimensional aerosol dynamics models for atmospheric simulation studies.

| **Model name, reference** | **Code availability** | **Aerosol processes** | **Meteorol. driver** | **Aerosol size distribution** | **Particle phase chemistry** | **Gas-phase chemistry** | **Numerical solution** | **Model output** |
|---|---|---|---|---|---|---|---|---|
| MAFOR This work and Karl et al. (2011) | open source, GPL-3 license | nucleation, coagulation, condensation, dry dep. | trajectories or plume dispersion | logarithmic, fixed sectional | PNG scheme, liquid-phase chemistry in fog droplets | Mainz Organic Mechanism, DMS and amine chemistry | KPP-2.2.3 Rosenbrock solver and Euler forward differences | number and mass size distribution, composition distribution |
| AEROFOR Pirjola (1999); Pirjola and Kulmala (2001) | no | nucleation, coagulation, condensation, dry dep. | trajectories or plume dispersion | logarithmic, fixed sectional | none | modified EMEP scheme, DMS and iodine chemistry | NAG library FORTRAN-routine D02EJF | number and surface size distribution, composition distribution |
| SALSA Kokkola et al. (2008) https://github.com/UCLALES-SALSA/SALSA-standalone | open source, Apache license 2.0 | nucleation, coagulation, condensation, dry dep. | 3-dimensional atmospheric models and LES models (e.g. PALM) | volume ratio, moving centre or fixed sectional | thermodynamic equilibrium of soluble compounds | none | Euler forward differences | number and volume size distribution, composition distribution |
| UHMA Korhonen et al. (2004) | no | nucleation, coagulation, condensation, dry dep. | trajectories or plume dispersion | logarithmic, hybrid, moving centre or retracking | thermodynamic equilibrium of soluble compounds | none | 4th order Runge-Kutta | number and surface size, composition distribution |
| ADCHEM Roldin et al. (2011) | no | nucleation, coagulation, condensation, dry dep. | trajectories; built-in atmos. transport and diffusion | logarithmic, full stationary, moving centre or full moving | PNG scheme, thermodynamic equilibrium for SIA | modified EMEP scheme | MATLAB® ode15s solver | number size distribution |
| M7 Vignati et al.(2004) GMXe Pringle et al. (2010) | no | nucleation, coagulation, condensation | 3-dimensional atmospheric models | superstition of seven log-normal distributions | none | sulphate chemistry | Euler Backward Iterative (EBI) method | number size distribution |
| PartMC-MOSAIC Riemer et al. (2009) http://lagrange.mechse.illinois.edu/partmc/ | PartMC is open source / MOSAIC code upon request to R. A. Zaveri | coagulation, gas-particle transfer | Lagrangian parcel framework | Individual aerosol particles (about $10^5$) | aerosol chemistry model MOSAIC | | stochastic simulation algorithm for coagulation | number and mass size distribution, composition distribution |
| LNMOM-DC Anand and Mayya (2015); Sarkar et al. (2020) | no | coagulation | plume/puff dispersion | monodisperse, log-normal distribution | none | none | N/A | PN survival fraction; PNC; number size distribution |

### 2.2 Model structure

Figure 1 illustrates the model structure of MAFOR v2.0. The model consists of three basic modules: (1) a chemistry module; (2) an aerosol dynamic module; and (3) a plume dispersion module. MAFOR is coupled with the chemistry sub-model MECCA v4.0 that allows the dynamic generation of new chemistry solver code and photolysis routines after adding new species and/or reactions to the chemistry mechanism. The newly generated code is packaged into a FORTRAN library that is included during the compilation of MAFOR, avoiding the need to build the MECCA interface each time when changes are made in the model code.

The chemistry module of MAFOR calculates time-varying gas-phase concentrations and aqueous phase concentrations (in the droplet mode) by solving the non-linear system of stiff chemical ordinary differential equations (ODE). The photolysis module JVAL (Sander et al., 2014) is used to calculate photolysis rate coefficients for photo-dissociation reactions. JVAL includes the JVPP (JVal PreProcessor) which pre-calculates the parameters required for calculating photolysis rate coefficients based on absorption cross sections and quantum yields of the atmospheric molecules. The kinetic pre-processor KPP v2.2.3 (Sandu and Sander, 2006) is used to transform the chemical equations into program code for the chemistry solver. The numerical integration of the ODE system of gas-phase and aqueous phase reactions is done with Rosenbrock 3 using automatic time step control. The chemistry module also includes the emission and dry deposition of gases.

The aerosol dynamics module includes homogeneous nucleation of new particles according to various parameterizations, Brownian coagulation, condensation/evaporation, dry deposition, wet scavenging, and primary emission of particles. The composition of particles in any size bin can change with time due to multicomponent condensation and/or due to coagulation of particles. The aerosol dynamic solver updates number and component mass concentrations in the order: (1) condensation/evaporation, (2) coagulation, (3) nucleation, (4) dry and wet deposition, and (5) emission. It returns updated number concentration, updated component mass concentration per size bin and the updated gas-phase concentration of condensable and nucleating vapours.

The plume dispersion module calculates the vertical dispersion of a Gaussian plume as a function of $x$ (the downwind distance from the point of emission) and the dilution rate for the particle and gas concentrations in the plume. Temperature in the plume and the plume height is varying with time according to prescribed dispersion parameters. In case the MAFOR model would be included into a dispersion or climate modelling system, the plume dispersion model in Fig. 1 would be replaced by the advection-diffusion modules of that system.

The model starts with the initialization of the particle number and mass composition distributions and gas-phase concentrations. In the plume simulation, the aerosol distribution and gas-phase concentrations of the background air and dispersion parameters are initialized based on the user input. Meteorological conditions are updated on an hourly basis. It is possible to tailor the properties of the (lumped) organic compounds for the simulation to best represent the conditions in a chamber experiment or specific atmospheric region. As the model begins the integration over time, each process is solved using operator splitting, in the order: plume dispersion, chemical reactions, and aerosol dynamics. The changed gas-phase concentrations from the chemistry module are used in the aerosol dynamic module in the condensation/evaporation and nucleation processes.

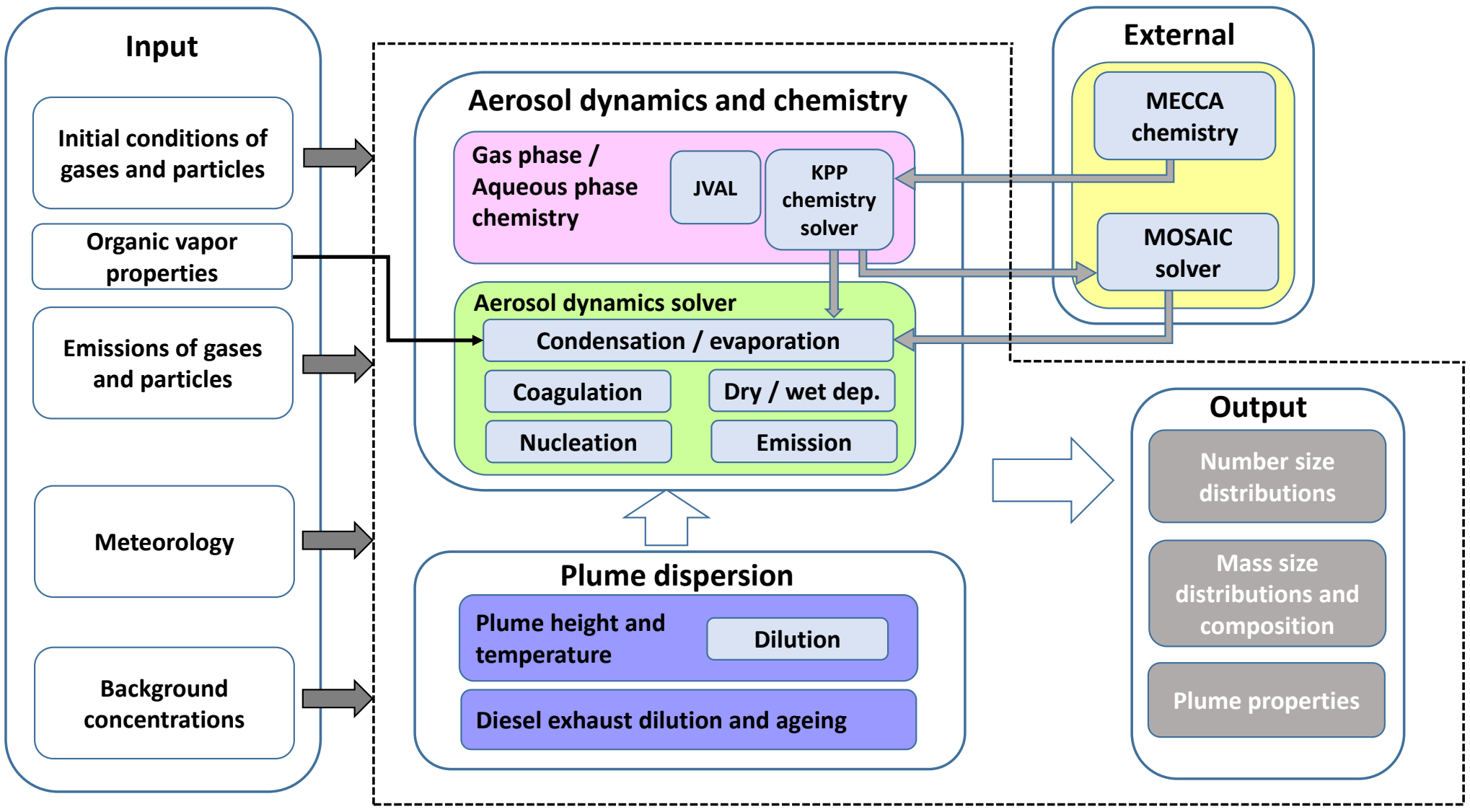

**Figure 1.** Illustration of the model structure. Input data is on the left side. Area with a dashed outline contains the MAFOR model. External modules: interface to MECCA v4.0 and interface to MOSAIC solver (the models are not part of MAFOR). MECCA v4.0 is used to create the modules for the KPP chemistry solver, and JVAL solver provides photolysis rate constants.

Pre-existing mass and number are input in the calculation of aerosol dynamic processes. The module first calculates the mass concentration of liquid water in each size section and consequently the wet diameter of particles, which is used for the calculation of aerosol dynamic processes. The dilution of particles is calculated after the number and mass concentrations of the current time step have been updated.

MAFOR has an interface to the MOSAIC model (Zaveri et al., 2008) for the treatment of condensation/evaporation of semi-volatile inorganic gases. This interface encapsulates a reduced version of the MOSAIC solver code in an external FORTRAN library. The thermodynamic module of MOSAIC is the Multicomponent Equilibrium Solver for Aerosols (MESA) model (Zaveri et al., 2005b). MESA is used here to calculate aerosol phase state, the activity coefficients of electrolytes in the aqueous solution, the equilibrium concentration of ammonium ($NH_4^+$) in all size bins, and the parameters for dynamic growth by dissolution. An operator-split aerosol equilibrium calculation in MESA is performed to recalculate electrolyte composition and activity coefficients in each size bin. Finally, the MOSAIC interface provides the parameters required to determine the solubility terms in the PNG scheme (Jacobson, 2005b). In the PNG scheme, condensation (dissolution) and evaporation of $HNO_3$, HCl and $H_2SO_4$ is solved first. Following the growth calculation for all acid gases, $NH_3$ is equilibrated with all size bins, conserving charge among all ions. In this method, ammonia growth is effectively a time-dependent process, because the equilibration of $NH_3$ is calculated after the diffusion-limited growth of all acids. The PNG scheme allows operator-split to be done at long

time step (e.g. 150–300 s) between the growth calculation and the equilibrium calculation without causing oscillatory solutions when solving the condensation/evaporation of acid and base as separate processes (Jacobson, 2005b).

Two aspects in the implementation of the dynamic partitioning of inorganic and organic aerosol components in MAFOR v2.0 advance beyond the original concepts:

1. The condensation and dissolution of $HNO_3$ and HCl was modified compared to the original PNG scheme. Condensation of the two gases to a particle size bin is applied when solid is present in the bin, using the minimum saturation vapour concentration. This leads to more nitrate mass to transfer to the aerosol phase compared to the original PNG scheme, which only considers solubility.

2. The coupling of the mass-based formulation from the 2-D VBS framework (Donahue et al., 2011) for organic aerosol phase partitioning, considering non-ideal solution behaviour, with the dynamics of organic condensation and evaporation according to a so-called hybrid approach, addressing the critical role of condensable organics in the growth of freshly nucleated particles.

### 2.3 Processes included in the model

#### 2.3.1 Multiphase chemistry

The gas-phase and aqueous-phase chemistry mechanism is based on the MECCA chemistry sub-model of CAABA/MECCA v4.0 (Sander et al., 2019). In addition to the basic tropospheric chemistry it contains the Mainz Organic Mechanism (MOM) as oxidation scheme for volatile organic compounds (VOC), including alkanes, alkenes (up to four carbon atoms), ethyne (acetylene), isoprene, several aromatics and five monoterpenes. Most of the VOC species of MOM are available for initialization in simulations with MAFOR. Diurnal variation of photolysis rates are based on Landgraf and Crutzen (1998) with the updates included in the JVAL photolysis module (Sander et al., 2014), such as updated UV/VIS cross sections as recommended by the Jet Propulsion Laboratory (JPL), Evaluation no. 17 (Sander et al., 2011). The chemistry mechanism of MECCA was extended by a comprehensive reaction scheme for dimethyl sulphide (DMS) adopted from Karl et al. (2007) and oxidation schemes of several amines: methylamine, dimethylamine, trimethylamine (Nielsen et al., 2011), 2-aminoethanol (Karl et al., 2012b), amino methyl propanol, diethanolamine, and triethanolamine (Karl et al., 2012c). In total, the current chemistry mechanism of MAFOR v2.0 contains 781 species and 2220 reactions in the gas phase, as well as 152 species and 465 reactions in the aqueous phase. Initial concentrations of relevant gas-phase species, their dry deposition rate and their emission rate can be provided by the model user.

The aqueous phase chemistry is currently restricted to the liquid phase of coarse mode aerosol (short: droplet mode). The composition of the liquid phase may be initialized with concentrations of the most relevant cations and anions. Transfer of molecules between the gas phase and the aqueous phase of coarse mode aerosol and vice versa is treated by the resistance model of Schwartz (1986) which considers gas-phase diffusion, mass accommodation and the Henry's Law constants. The mass transfer coefficient $k_{m,q}$, a first order loss rate constant, describes the mass transport of compound $q$ from the gas phase to
the aqueous phase:

$$k_{m,q} = \left( \frac{r_d^2}{3D_q} + \frac{4r_d}{3c_{m,q}\alpha_{l,q}} \right)^{-1}, \tag{1}$$

where $D_q$ is the molecular diffusion coefficient in the gas phase, $c_{m,q}$ is the molecular speed and $\alpha_{l,q}$ is the mass accommodation coefficient (adsorption of the gas to the droplet surface), and $r_d$ is the droplet radius (mean radius of the monodisperse droplet mode). The first term represents the resistance caused by gas phase diffusion, while the second term represents the interfacial mass transport. It is assumed that the liquid aerosol (cloud/fog droplet) behaves as an ideal solution and that no formation of solids occurs in the solution.

The change of gas-phase and aqueous phase concentrations, $C_{g,q}$ and $C_{aq,q}$, of a (soluble) compound with time due to chemical reactions in a system with equilibrium partitioning is then described by:

$$\frac{dC_{g,q}}{dt} = Q_{g,q} - k_{m,q}LWC\left(C_{g,q} - \frac{C_{aq,q}}{H_{A,q}}\right) \tag{2a}$$

and

$$\frac{dC_{aq,q}}{dt} = Q_{aq,q} + k_{m,q}\left(C_{g,q} - \frac{C_{aq,q}}{H_{A,q}}\right), \tag{2b}$$

where $Q_{g,q}$ and $Q_{aq,q}$ are the gas phase and aqueous phase net production terms in chemical reactions, respectively, and $LWC$ is the liquid water content. The dimensionless Henry's law coefficient, $H_{A,q}$, for the equilibrium partitioning is independent of the liquid water content. Aqueous phase partitioning parameters and aqueous phase reactions are adopted from the MECCA chemistry module, extended with a treatment of organic molecules in the aqueous phase from Ervens et al. (2004) and amines in the aqueous phase (Ge et al., 2011; Karl et al., 2012c).

### 2.3.2 Condensation/evaporation

The growth of particles through multicomponent condensation is implemented in MAFOR according to the continuum/transition regime theory corrected by a transitional correction factor (Fuchs and Sutugin, 1970). The scheme used for condensation/evaporation is the Analytical Predictor of Condensation (APC; Jacobson, 2005b) for dynamic transfer of gas-phase molecules to the particles over a discrete time step.

The difference between partial pressure of a condensable compound in air and vapour pressure on the particle surface is the driving force for condensation/evaporation in the model. Condensation/evaporation is solved by first calculating the single particle molar condensation growth rate $I_{q,i}$ ($m^3\,s^{-1}$) for each compound $q$ in each size bin $i$, given by:

$$I_{q,i} = \frac{d\upsilon_{q,i}}{dt} = \left(48\pi^2\upsilon_i\right)^{1/3} D_q\beta_{q,i}\upsilon_{g,q}\frac{N_A}{10^6 MW_q}\left[C_{g,q} - S'_{q,i}C_{eq,q}\right], \tag{3}$$

where $\upsilon_i$ is the particle volume, $\upsilon_{g,q}$ is the molecular volume of the condensing vapour, and $C_{eq,q}$ (in $\mu g\,m^{-3}$) is the saturation vapour concentration over a flat solution of the same composition as the particles. The factor $N_A/10^6 MW_q$ is for conversion from mass-based to molecular units, where $N_A$ is the Avogadro constant ($N_A = 6.022\times10^{23}\,mol^{-1}$) and $MW_q$ is the molecular

weight of the condensing vapour ($g\,mol^{-1}$). The diffusion coefficient $D_q$ is estimated using an empirical correlation by Reid et al. (1987). The equilibrium saturation ratio of the condensing vapour, $S'_{q,i}$, is determined by the Kelvin effect and Raoult's law: $S'_{q,i} = \gamma_{q,i} Ke$, with the molar fraction in the particle phase, $\gamma_{q,i}$, and the Kelvin term $Ke$.

The transitional correction factor $\beta_{q,i}$ is (Fuchs and Sutugin, 1970):

$$\beta_{q,i} = \frac{Kn+1}{1+\left(\frac{4}{3\alpha_q}+0.377\right)Kn+\frac{4}{3\alpha_q}Kn^2}, \tag{4}$$

where $\alpha_q$ is the mass accommodation (or sticking) coefficient fo compound $q$. The default values for the accommodation coefficient are 0.5 for $H_2SO_4$ and 0.13 for MSA. The model user can replace these values by unity. The accommodation coefficient of organic vapours and all other inorganic vapours is assumed to be equal to unity. The Knudsen number is $Kn = \lambda_v / r_i$ and $\lambda_v$ is the mean free path of vapour molecules and $r_i$ is the particle radius.

The Kelvin effect due to curvature of particles is considered for the condensation/evaporation of all vapours. Inclusion of the Kelvin term reduces the condensation flux of vapours to particles smaller than 10 nm diameter in size. The Kelvin term $Ke$ is expressed as:

$$Ke = \exp\left(\frac{2\sigma_q 10^{-3} MW_q}{RT\rho_{L,q} r_i}\right), \tag{5}$$

where $R$ is the universal gas constant ($R = 8.3144\,kg\,m^2\,s^{-2}\,K^{-1}\,mol^{-1}$), and $T$ is the air temperature (K), $\sigma_q$ is the surface tension ($kg\,s^{-2}$), $\rho_{L,q}$ is the density of the pure liquid ($kg\,m^{-3}$), and $r_i$ is particle radius in size bin $i$ (m). Surface tension and density of the pure liquid for the condensing vapours are given in Table 2. The vapour pressure of the lumped organic compounds is modified by their molar fraction in the particle phase (according to Raoult's law), and by their molar volume and surface tension according to the Kelvin effect. The condensation flux of $H_2SO_4$ and MSA is corrected by the effect of hydrate formation following Karl et al. (2007). For organic vapours, the revised flux formulation by Lehtinen and Kulmala (2003) is used, which accounts for the molecule-like properties of the small particles, by modification of the transitional correction factor, Knudsen number and mean free path.

The condensation of $NH_3$ is coupled to the concentration of acid gases ($H_2SO_4$, $HNO_3$ and HCl). If the $NH_3$ concentration is at least twofold compared to $H_2SO_4$ concentration, then two $NH_3$ molecules are removed from the gas phase, assuming formation of ammonium sulfate [$(NH_4)_2SO_4$]. If there is excess of $NH_3$ available for reaction with $HNO_3$ to produce ammonium nitrate ($NH_4NO_3$), then each $HNO_3$ molecule removes one $NH_3$ molecule from the gas phase. $NH_3$ can also react with HCl to produce ammonium chloride ($NH_4Cl$). The formation of $NH_4NO_3$ and/or $NH_4Cl$ then determines the saturation vapour pressures of $NH_3$, $HNO_3$, and HCl. At equilibrium, the relation between the saturation concentration and the gas-solid equilibrium coefficients $K_{p,NH_4NO_3}$ and $K_{p,NH_4Cl}$, together with the mole-balance equation, can be used to obtain the analytical solution for the saturation concentration of $NH_3$ (i.e. $C_{eq,NH_3}$), as follows:

$$C_{eq,NH_3} C_{eq,HNO_3} = K_{p,NH_4NO_3} \tag{6a}$$

$$C_{eq,NH_3} C_{eq,HCl} = K_{p,NH_4Cl} \tag{6b}$$

$$C_{g,\mathrm{NH_3}} - C_{eq,\mathrm{NH_3}} = C_{g,\mathrm{HNO_3}} - C_{eq,\mathrm{HNO_3}} + C_{g,\mathrm{HCl}} - C_{eq,\mathrm{HCl}} \quad (6c)$$

$$C_{eq,\mathrm{NH_3}} = \frac{C_0}{2} + \frac{1}{2}\sqrt{C_0^2 + 4\left[K_{p,\mathrm{NH_4NO_3}} + K_{p,\mathrm{NH_4Cl}}\right]} \quad (6d)$$

with

$$C_0 = C_{g,\mathrm{NH_3}} - C_{g,\mathrm{HNO_3}} - C_{g,\mathrm{HCl}}. \quad (6e)$$

The saturation concentrations of $HNO_3$ (i.e. $C_{eq,\mathrm{HNO_3}}$) and HCl (i.e. $C_{eq,\mathrm{HCl}}$) are obtained accordingly. The reaction of alkylamines with $HNO_3$ to alkyl ammonium nitrate is treated in analogy to the ammonia-nitric acid system. Alternatively, the PNG scheme, applicable across the entire relative humidity range, can be used to solve the growth by dissolution of $HNO_3$ and HCl, and equilibration of $NH_3$, as will be described in Sect. 2.4.

Saturation vapour pressures of the organic compounds are based on the $C^0$ values (pure-compound saturation mass concentration) provided by the model user. Typical $C^0$ values are shown in Table 2. Alternatively, the absorptive partitioning of organics is considered using the 2-D VBS method, as will be described in Sect. 2.5.

The gas-phase concentration of a condensing vapour with respect to condensation/evaporation and gas-phase chemistry is predicted according to:

$$\frac{dC_{g,q}}{dt} = Q_{g,q} - 4\pi D_q \sum_{i=1}^{N_B} r_i N_i \beta_{q,i}\left[C_{g,q} - S'_{q,i} C_{g,eq,q}\right], \quad (7)$$

where $N_i$ is the number concentration of particles ($m^{-3}$). The second term on the right-hand-side (RHS) in this equation represents the condensation/evaporation flux to a particle population, as defined in Eq. (3).

The change of the particle phase mass concentration, $m_{q,i}$ of the compound in each size bin with time due to condensation/evaporation is described by:

$$\frac{dm_{q,i}}{dt} = \frac{d\upsilon_{q,i}}{dt} \cdot \frac{N_i}{\upsilon_{g,q}} \cdot \frac{10^6 MW_q}{N_A} = k_{T,q,i}\left[C_{g,q} - S'_{q,i} C_{g,eq,q}\right] \quad (8a)$$

with

$$k_{T,q,i} = 4\pi r_i N_i D_q \beta_{q,i}, \quad (8b)$$

where $k_{T,q,i}$ is the mass transfer rate ($s^{-1}$) of gas to the particles of a size bin.

A non-iterative solution for the gas phase and particle phase concentration in each bin due to condensation over time is obtained by making use of the mass balance equation of the final aerosol and gas phase concentrations (Jacobson, 2005b). Details of the APC solver are given in Appendix B.

The condensation of $H_2O$ is accounted for by assuming the particles to be in equilibrium with the ambient water vapour. The uptake of water is calculated based on equilibrium thermodynamics (Binkowski and Shankar, 1995) using empirical

**Table 2.** Molecular properties of the condensing vapours. Saturation concentration $C^0$ is provided by the model user for the lumped organic compounds.

| Compound | Molecular weight $[\mathrm{g\,mol^{-1}}]$ | Surface tension $[\mathrm{kg\,s^{-2}}]$ | Density pure liquid $[\mathrm{kg\,m^{-3}}]$ | Accommod. coefficient [–] | Saturation vapour pressure $p_s^0(298\,\mathrm{K})$ [Pa] | Saturation concentration $C^0(298\,\mathrm{K})$ $[\mathrm{\mu g\,m^{-3}}]$ |
|---|---|---|---|---|---|---|
| $H_2SO_4$ | 98.08 | 0.052 [a] | 1851 [a] | 0.5 \| 1 | $4.05\times10^{-3}$ [b] | 160 |
| MSA | 96.11 | 0.053 [c] | 1507 [d] | 0.13 \| 1 | $9.85\times10^{-2}$ [c] | 3820 |
| $HNO_3$ | 63.0 | 0.1084 | 1725 | 1 | [e] | [e] |
| HCl | 36.5 | 0.1084 | 1725 | 0.15 | [e] | [e] |
| $NH_3$ | 17.0 | 0.1084 | 1725 | 1 | [e] | [e] |
| Amine | 63.0 | 0.1084 | 1725 | 1 | [f] | [f] |
| BSOV | 170 | 0.048 [g,h] | 1570 [h] | 1 | $3.06\times10^{-5}$ | 2.1 |
| BLOV | 170 | 0.048 [g,h] | 1570 [h] | 1 | $4.37\times10^{-7}$ | 0.03 |
| BELV | 372 | 0.048 [g,h] | 1570 [h] | 1 | $9.0\times10^{-10}$ | 0.0001 |
| ASOV | 137 | 0.048 [g,h] | 1570 [h] | 1 | $1.8\times10^{-5}$ | 1.0 |
| ALOV | 137 | 0.048 [g,h] | 1570 [h] | 1 | $2.0\times10^{-7}$ | 0.01 |
| AELV | 338 | 0.048 [g,h] | 1570 [h] | 1 | $9.0\times10^{-10}$ | 0.0001 |
| PIOV | 296 | 0.029 | 792 | 1 | $8.05\times10^{-4}$ | 100 |
| PSOV | 366 | 0.031 | 778 | 1 | $3.80\times10^{-6}$ | 0.6 |
| PELV | 450 | 0.032 | 810 | 1 | $9.97\times10^{-10}$ | 0.0002 |

Footnotes:

[a] Vehkamäki et al. (2002) using unity mole fraction of $H_2SO_4$.

[b] temperature-dependent expression from Bolsaitis and Elliott (1990) using unity mole fraction of $H_2SO_4$.

[c] temperature-dependent expression from Kreidenweis and Seinfeld (1998).

[d] Wyslouzil et al. (1991).

[e] equation (6) with $K_{p,NH_4NO_3}$ and $K_{p,NH_4Cl}$ from Zaveri et al. (2008).

[f] treated in analogy to the ammonia-nitric acid system.

[g] temperature-dependent surface tension for pure succinic acid from Hyvärinen et al. (2006).

[h] value for the organic vapours BSOV, BLOV, BELV, ASOV, ALOV, AELV can be replaced by model user.

polynomials (Tang and Munkelwitz, 1994) for the mass fraction of solute as a function of water activity. Polynomials for ammonium nitrate and ammonium sulfate are adopted from Chan et al. (1992). The water uptake of (soluble) semi-volatile organics is treated as sodium succinate with polynomials adopted from Peng and Chan (2001) and water uptake of sea-salt particles is treated as sodium chloride (NaCl) according to Tang et al. (1997).

#### 2.3.3 Nucleation

New particles are introduced into the atmosphere either by direct emission or by in-situ nucleation of semi-volatile or low-volatile vapours. Nucleated particles (critical clusters) have initial sizes in the order of a few nanometres or less, which is much smaller than typical primary emission particle size ranges. A competition between growth by condensation and loss by coagulation determines the survival probability of a nucleated particle through a certain size range, usually up to 100 nm. Since freshly nucleated particles are small, they are highly diffusive and have a high propensity to collide with pre-existing particles. Nucleation in the atmosphere is a dynamic process that involves interactions of precursor vapour molecules, small clusters and pre-existing particles (Zhang et al., 2012). However, the atmospheric nucleation mechanism is still surrounded with uncertainties. Several options of parameterized nucleation mechanisms can be chosen in the model; Table 3 provides a list of the available mechanisms.

Sulphuric acid is a highly probable candidate for atmospheric nucleation (Kulmala et al., 2004). Sihto et al. (2006) reported that nucleation mode particle concentrations observed in a boreal forest (Hyytiälä, Southern Finland) typically depend on $H_2SO_4$ concentration via a power-law relation with the exponent of 1 or 2. The proposed theory (Kulmala et al., 2006) of atmospheric nucleation by *cluster activation* (option 5) or *kinetic nucleation* (option 1) could be used to explain the observed behaviour. Charged clusters formed on ions are more stable and can grow faster than neutral clusters. Ion-mediated nucleation (IMN) considers the role of ubiquitous ions in enhancing the stability of pre-nucleation clusters (Yu and Turco, 2001). The ionization rate of air is about 2 ion pairs $cm^{-3}\,s^{-1}$ at ground level and increases up to 20–30 ion pairs $cm^{-3}\,s^{-1}$ in the upper troposphere. A constant ionization rate of 2 ion pairs $cm^{-3}\,s^{-1}$ is used in all nucleation parameterizations that consider charged clusters in MAFOR. The *combined nucleation* scheme (option 7) is a combination of IMN and cluster activation (Karl et al., 2011; hereafter K2011) providing an upper estimate to the nucleation rate at low $H_2SO_4$ concentrations under tropospheric conditions.

Binary homogeneous nucleation (BHN) of $H_2SO_4$-$H_2O$ may be the prevailing mechanism in the upper troposphere, and in some cases, classical BHN theory has successfully explained the observed formation rates of new particles (Weber et al., 1999; Pirjola et al., 1998). BHN is implemented in MAFOR based on the parameterization of Vehkamäki et al. (2002; hereafter V2002) which takes into account the effect of hydrate formation (Jaecker-Voirol et al., 1987; Noppel et al., 2002), extended to temperatures above 305 °C (Vehkamäki et al., 2003), suitable for predicting the particle formation rate at high temperatures in exhaust conditions (option 2).

Määttänen et al. (2018a; hereafter M2018) presented new parameterizations of neutral and ion-induced $H_2SO_4$-$H_2O$ particle formation (option 11), valid for large ranges of environmental conditions, which have been validated against a particle formation rate data set generated in Cosmics Leaving OUtdoor Droplets (CLOUD) experiments. The implementation of the M2018 parameterization in MAFOR v2.0 has been tested in an urban background scenario ("Case 1"); giving a maximum particle formation rate of 0.95 $cm^{-3}\,s^{-1}$ when $H_2SO_4$ concentration peaked at $5 \times 10^7\,cm^{-3}$ (Supplement, Sect. S2).

Participation of a third compound to the nucleation process might explain discrepancies between $H_2SO_4$-water nucleation theories and laboratory measurements and field studies. Ternary homogeneous nucleation (THN) involving $NH_3$ is a strong

Geoscientific Model Development
EGU
Discussions

**Table 3.** Nucleation options in the MAFOR model.

| Option no. | Nucleation mechanism | References |
|---|---|---|
| 1 | kinetic $H_2SO_4$ | Kulmala et al. (2006) |
| 2 | binary homogeneous $H_2SO_4$-$H_2O$ | Vehkamäki et al. (2002; 2003) |
| 3 | THN; homogeneous $H_2SO_4$-$H_2O$-NH3 | Merikanto et al. (2007; 2009) |
| 4 | TIMN; homogeneous and ion-mediated $H_2SO_4$-$H_2O$-$NH_3$ | Yu et al. (2018); Yu et al. (2020) |
| 5 | activation $H_2SO_4$ | Kulmala et al. (2006) |
| 6 | kinetic amine-$HNO_3$ | Karl et al. (2012b) |
| 7 | combination $H_2SO_4$ (activation and ion-mediated) | Karl et al. (2011) |
| 8 | OS1; activation organic-$H_2SO_4$ | Karl et al. (2012a); Paasonen et al. (2010) |
| 9 | OS2; kinetic organic-$H_2SO_4$ | Karl et al. (2012a); Paasonen et al. (2010) |
| 10 | OS3; total organic-$H_2SO_4$ | Karl et al. (2012a); Paasonen et al. (2010) |
| 11 | neutral and ion-induced $H_2SO_4$-$H_2O$ | Määttänen et al. (2018a; 2018b) |
| 12 | HET; organic-$H_2SO_4$ in diesel exhaust | Pirjola et al. (2015) |
| 13 | ACDC/THN; homogeneous $H_2SO_4$-$H_2O$-$NH_3$ | Henschel et al. (2016); Baranizadeh et al. (2016) |

option, due to the abundance of $NH_3$ in the atmosphere and its ability to lower the partial pressure of $H_2SO_4$ above the solution surface. Merikanto et al. (2007) revised the classical theory of THN by including the effect of stable ammonium bisulphate formation (option 3), resulting in predicted nucleation rates that are several orders of magnitude lower compared to the original ternary nucleation model by Napari et al. (2002). More recently, the particle formation rates for THN have been updated based on simulations with the Atmospheric Cluster Dynamics Code (ACDC; Olenius et al., 2013) using quantum chemical input data (option 13). ACDC simulates the dynamics of a population of molecular clusters by numerically solving the cluster birth-death equations. Details of the ACDC simulations of the ternary $H_2SO_4$-$NH_3$-$H_2O$ system can be found in Henschel et al. (2016; hereafter H2016). The ACDC/THN lookup-table published by Baranizadeh et al. (2016) was implemented in MAFOR v2.0 allowing for the interpolation of particle formation rates under various conditions. MAFOR v2.0 also includes an implementation of the lookup-table parameterization of ternary nucleation (TIMN, option 4) by Yu et al. (2020; hereafter Y2020). TIMN includes both ion-mediated and homogeneous ternary nucleation of $H_2SO_4$-$NH_3$-$H_2O$. At very low $NH_3$ concentration ($[NH_3] \leq 10^5\ cm^{-3}$), TIMN predicts nucleation rates according to BHN. Hence, the TIMN scheme offers the clear advantage that it can be directly applied to calculate nucleation rates in the whole troposphere in 3-D models.

Figure 2 compares the most relevant parameterizations for the particle formation from sulphuric acid nucleation under conditions relevant for the "Urban Case" scenario ($T = 262$ K and RH = 80 %) as a function of the $H_2SO_4$ concentration. The $H_2SO_4$ concentration for which the particle formation rate reaches $J_{nuc} = 1\ cm^{-3}\ s^{-1}$ is $3.2 \times 10^6$, $4.6 \times 10^6$, $1.8 \times 10^7$, $7.4 \times 10^7\ cm^{-3}$, and $6.0 \times 10^7$ for K2011, M2018, Y2020 (at $[NH_3] = 10^5\ cm^{-3}$), H2016 (at $[NH_3] = 2 \times 10^6\ cm^{-3}$) and V2002, respectively. K2011 gives highest nucleation rates at low $H_2SO_4$ concentrations and shows an almost linear dependence on $[H_2SO_4]$, because this parameterization does not consider kinetic limitation. The M2018 curve shows two turning points: the first at

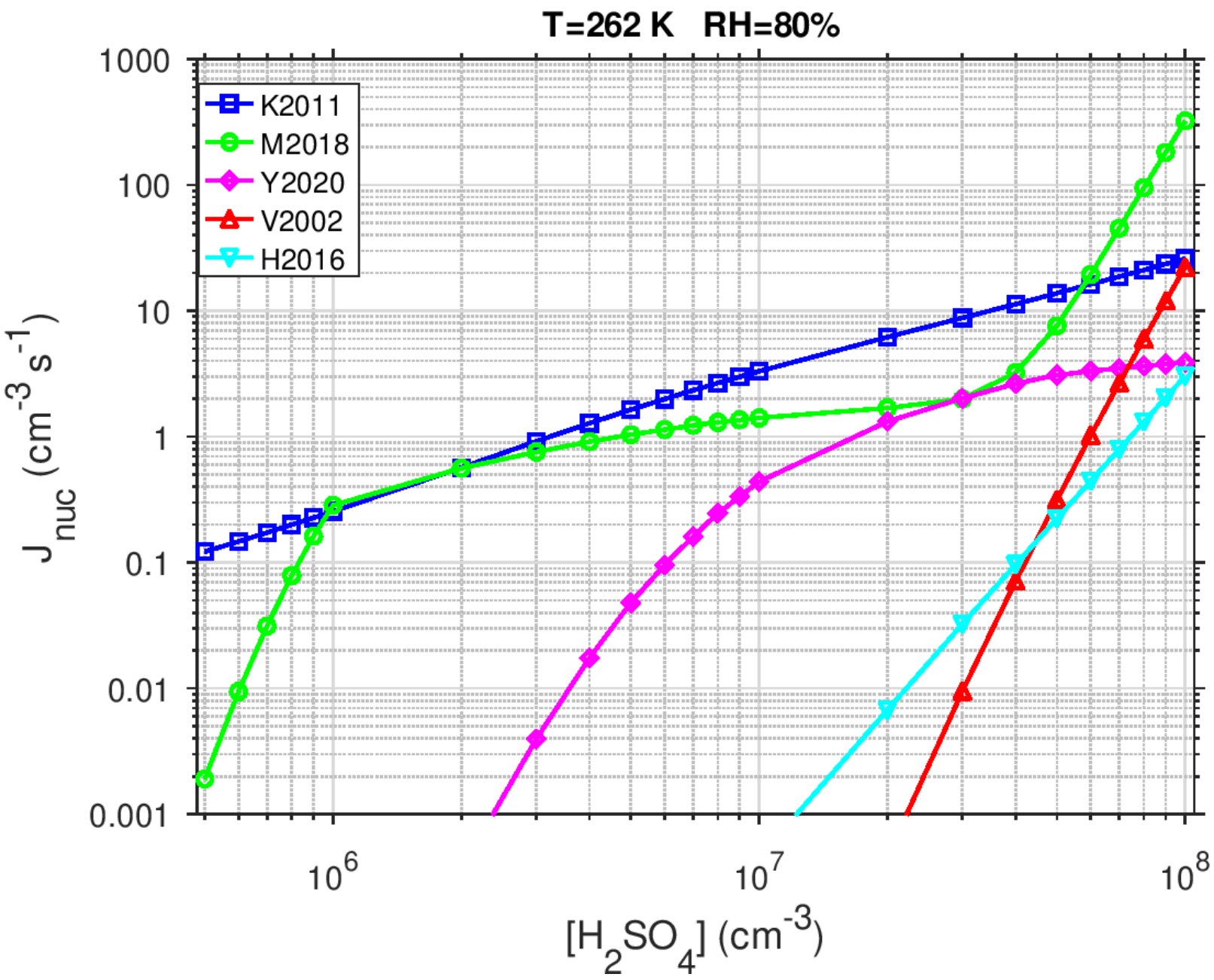

**Figure 2.** Predicted nucleation rate $J_{nuc}$ ($cm^{-3}\,s^{-1}$) as function of the concentration of $H_2SO_4$ (at $T = 262$ K and RH = 80%) calculated with different parameterizations for particle formation through sulphuric acid: combined activation and IMN (K2011), neutral and ion-induced BHN (M2018), TIMN at $NH_3 = 1\times10^5\,cm^{-3}$ (Y2020), THN at $NH_3 = 2\times10^6\,cm^{-3}$ (H2016) and classical BHN (V2002).

[$H_2SO_4$] $\sim 1\times10^6\,cm^{-3}$, when ion-induced nucleation reaches the kinetic limit, and the second at [$H_2SO_4$] $\sim 3\times10^7\,cm^{-3}$, when neutral BHN starts to dominate the total particle formation rate. The Y2020 parameterization is very sensitive to [$H_2SO_4$] at low $H_2SO_4$ concentrations but becomes insensitive to [$H_2SO_4$] at high concentrations due to the limitation of nucleation by ionization rate. Particle formation rates from M2018 at high [$H_2SO_4$] are an order of magnitude higher than those predicted from the earlier V2002 parameterization.

Direct evidence for the participation of low-volatile organic vapours in the nucleation process comes from laboratory experiments (e.g., Metzger et al., 2010) that revealed higher nucleation rates compared to $H_2SO_4$ alone when the concentration of organics was increased. Paasonen et al. (2010) proposed different empirical parameterizations for the nucleation of organic-$H_2SO_4$ clusters, analogous to the kinetic and cluster activation mechanisms for $H_2SO_4$ clusters (Kulmala et al., 2006). From their proposed organic-$H_2SO_4$ nucleation mechanisms, three are included in MAFOR: (1) activation of not-identified clusters by both $H_2SO_4$ and organics (OS1, option 8); (2) homogeneous heteromolecular nucleation between $H_2SO_4$ and organic molecules combined with homogeneous homomolecular nucleation of $H_2SO_4$ according to kinetic nucleation theory (OS2, option 9); and (3) homogeneous nucleation of the organic in combination with the nucleation routes of OS2 according to kinetic nucleation theory (OS3, option 10). The same low-volatility organic vapour (SOA-precursor BLOV) is used in all three parameterizations; it may also be involved in particle growth by condensation. Further nucleation options are: organic-$H_2SO_4$

nucleation in diesel exhaust (HET, option 12), as suggested in Pirjola et al. (2015), and kinetic nucleation of amine-$HNO_3$ (option 6) proposed by Karl et al. (2012b) for amine photo-oxidation experiments.

### 2.3.4 Coagulation

Coagulation of particles leads to a reduction in the total number of particles, changes the particle number size distribution and the chemical composition distribution, but leaves the total particle mass concentration unchanged. Coagulation is more efficient between particles of different sizes (inter-modal coagulation) than between same-sized particles (self-coagulation). The rate of coagulation is a product of size and diffusion coefficient: large particles provide a large collision surface and the smaller particles have high mobility (Brownian motion). For instance, a particle of 10 nm diameter size coagulates about 170 times faster with a 1 µm particle than with another 10 nm particle (Ketzel and Berkowicz, 2004). Thermal coagulation of particles caused by Brownian motion of the particles is considered with an accurate treatment in MAFOR: A semi-implicit solution is applied to coagulation (Jacobson, 2005b). The (non-iterative) semi-implicit solution yields an immediate volume-conserving solution for coagulation with any time step. Brownian coagulation coefficients between particles in size bin $i$ and $j$ are calculated according to Fuchs (1964). For particles in the transition regime, the Brownian coagulation coefficient can be calculated with the interpolation formula of Fuchs (1964):

$$K_{ij}^{B} = \frac{4\pi\left(r_i + r_j\right)\left(D_{m,i} + D_{m,j}\right)}{\dfrac{r_i + r_j}{r_i + r_j + \sqrt{\delta_{m,i}^2 + \delta_{m,j}^2}} + \dfrac{4\left(D_{m,i} + D_{m,j}\right)}{\sqrt{\overline{v}_{p,i}^2 + \overline{v}_{p,j}^2}\left(r_i + r_j\right)}}, \quad (9)$$

where $\delta_m$ is the mean distance from the centre of a sphere reached by particles leaving the sphere's surface and traveling a distance of particle mean free path. Further, $r$ is particle radius, $D_m$ is the particle diffusion coefficient and $\overline{v}_p$ is the mean thermal speed of a particle with index $i$ and $j$ for the respective size bin. Details on the Brownian coagulation algorithm are given in Appendix C.

Brownian coagulation is well understood for coalescing particles of spherical shape. Soot particles in diesel exhaust, however, are fractal-like agglomerates that consist of nano-sized primary spherules. In the direct exhaust plume, the fractal shape of the freshly emitted soot particles larger than 50 nm might increase their effective surface area that acts as a coagulation sink for the smaller particles (Ketzel and Berkowicz, 2004). The coagulation rate for agglomerate particles depends on particle mobility and the effective collision diameter, where it is usually assumed that the collision diameter is equal to either the mobility diameter or the outer diameter (Rogak and Flagan, 1992).

The effect of fractal geometry on coagulation is treated in the model by considering the effect of shape on radius, diffusion coefficient and the Knudsen number in the Brownian coagulation kernel. It is assumed that the collision radius, $r_c$, is equal to the outer radius, $r_\mathrm{f}$, of the agglomerate, defined as:

$$r_c = r_\mathrm{f} = r_s \times n_s^{1/D_\mathrm{f}}, \quad (10)$$

where $n_s$ is the number of primary spherules in the aggregate, $r_s$ is the radius of spherules and $D_\mathrm{f}$ is the fractal dimension. The model user is asked to provide values for $r_s$ and $D_\mathrm{f}$ for the fractal (soot) particles. In accordance with Lemmetty et al. (2008),

the effective density of fractal (soot) particles larger than the primary spherules is expressed as:

$$\rho_{\mathrm{eff}} = \rho_s r_s \left(\frac{D_{p,i}}{d_s}\right)^{D_{\mathrm{f}}-3}, \tag{11}$$

where $D_{p,i}$ is particle diameter of size bin $i$, while $d_s$ and $\rho_s$ are diameter and density of the primary spherules (for soot: 1200 kg m$^{-3}$), respectively.

The Brownian coagulation kernel is modified for fractal geometry with (Jacobson and Seinfeld, 2004):

$$K_{ij}^{B} = \frac{4\pi\left(r_{c,i}+r_{c,j}\right)\left(D_{m,i}+D_{m,j}\right)}{\dfrac{r_{c,i}+r_{c,j}}{r_{c,i}+r_{c,j}+\sqrt{\delta_{m,i}^{2}+\delta_{m,j}^{2}}}+\dfrac{4\left(D_{m,i}+D_{m,j}\right)}{\sqrt{\bar{v}_{p,i}^{2}+\bar{v}_{p,j}^{2}}\left(r_{c,i}+r_{c,j}\right)}}, \tag{12}$$

with the mean distance, $\delta_m$, from the particle's centre and the Knudsen number for air evaluated at the mobility radius. Here, the particle diffusion coefficient is evaluated at the mobility radius. For $D_{\mathrm{f}} = 3$ (spherical shape), the fractal radius, mobility radius, area-equivalent radius, and collision radius are identical and equal to the volume-equivalent radius, hence Eq. (12) simplifies to the Brownian kernel for spheres.

The performance of the model's coagulation process was analysed in the simulation of a chamber experiment in the presence of continuous emission of nanoparticles ("Case 2"). For details, we refer to Supplement Sect. S.4. When assuming compact spherical particles, the simulation of the evolution of the particle size distribution due to Brownian coagulation is in good agreement with the modelled particle size spectra and total particle number concentrations for the same case published in Anand et al. (2012). When fractal particles are considered in the model ($D_{\mathrm{f}} = 1.75$), the resulting particle size distribution is similar as in the same case of Anand et al. (2012), however, growth of the fractal particles into a secondary mode is less efficient. Differences in the coagulation efficiency probably lie in the details of the implementation of the fractal geometry in the coagulation kernel, although the same particle morphology was used in the present study. The accuracy of the coagulation solution in MAFOR with respect to particle mass conservation is sufficiently high, with an error of less than 0.5 %.

Two forces that increase/decrease the rate of aerosol coagulation are van der Waals forces, which results from the interaction of fluctuating dipoles, and viscous forces, which arise from the fact that velocity gradients induced by a particle approaching another particle in a viscous medium affect the motion of the other particle. It has been shown that van der Waals forces can enhance the coagulation rate of particles with diameter < 50 nm by up to a factor of five (Jacobson and Seinfeld, 2004). Viscous forces retard the rate of van der Waals force enhancement in the transition and continuum regimes (Schmitt-Ott and Burtscher, 1982).

In MAFOR, the correction of the Brownian kernel for van der Waals and viscous forces is done as in Jacobson and Seinfeld (2004). An interpolation formula for the van der Waals/viscous collision kernel $K_{k,j}^{V}$ between the free-molecular and continuum

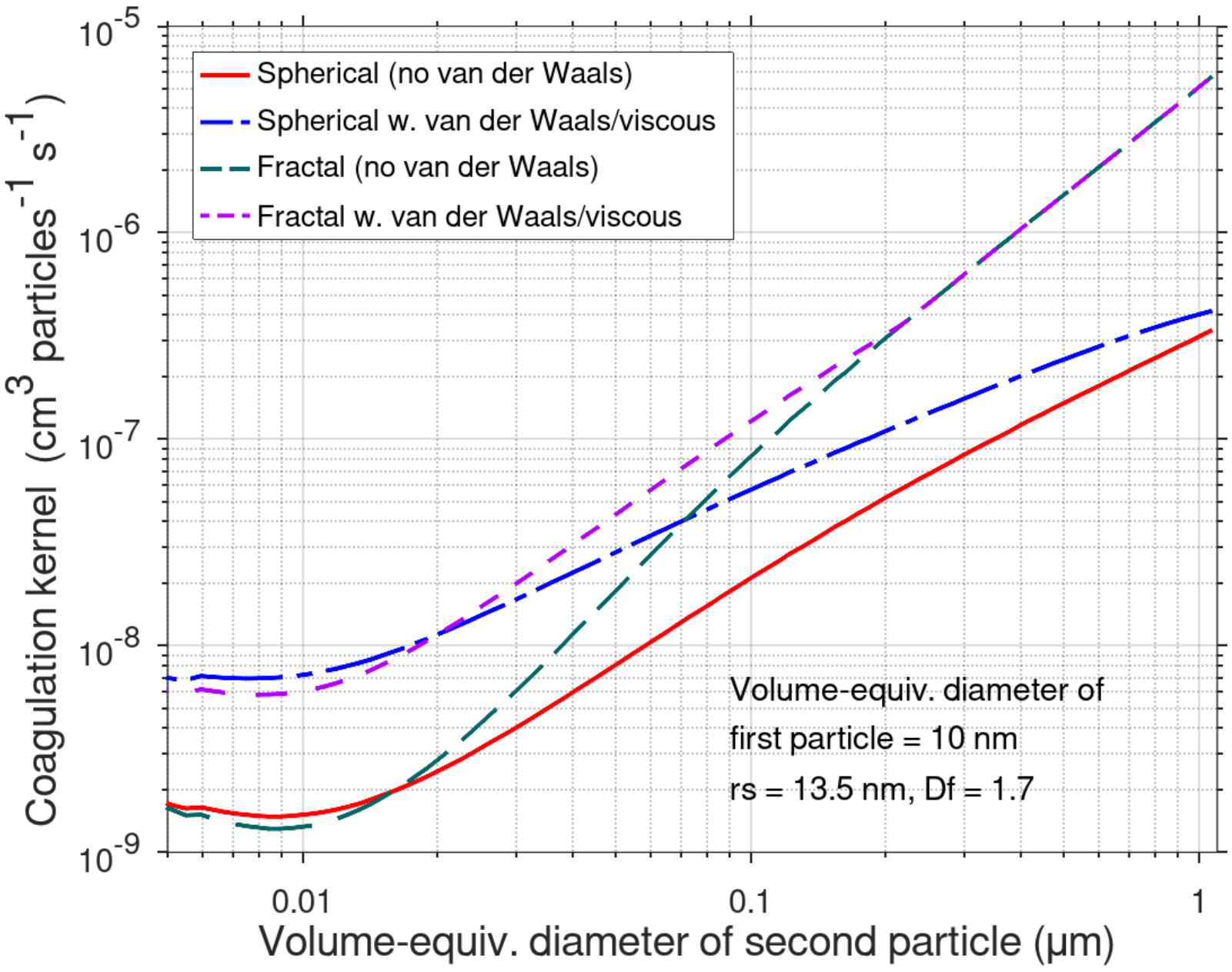

**Figure 3.** Modelled effect of fractal geometry and van der Waals/viscous forces when the volume-equivalent diameter is 10 nm and the volume-equivalent diameter of the second particle varies from 5 to 1000 nm.

regimes is applied (Alam, 1987; Jacobson and Seinfeld, 2004):

$$K_{i,j}^{V} = K_{i,j}^{B} \times \left\{ \frac{W_{c,i,j}\left[1 + \dfrac{4\left(D_{m,i}+D_{m,j}\right)}{\sqrt{\overline{v}_{p,i}^{2}+\overline{v}_{p,j}^{2}}\left(r_i+r_j\right)}\right]}{1 + \dfrac{W_{c,i,j}}{W_{k,i,j}} + \dfrac{4\left(D_{m,i}+D_{m,j}\right)}{\sqrt{\overline{v}_{p,i}^{2}+\overline{v}_{p,j}^{2}}\left(r_i+r_j\right)}} - 1 \right\}. \tag{13}$$

The quotient inside the curly brackets is the enhancement factor due to van der Waals/viscous forces. The correction factors $W_k$ for the free-molecular regime and $W_c$ for the continuum regime are given in Appendix D. Figure 3 shows the predicted effect of van der Waals forces and viscous forces on Brownian coagulation for spherical as well as for fractal particles ($r_s = 13.5$ nm and $D_\mathrm{f} = 1.7$) when the volume-equivalent diameter of the first particle is 10 nm.

Brownian motion by far dominates the collisions of sub-micrometre particles in the atmosphere. The coagulation of particles in turbulent flow is affected by two mechanisms: spatial fluctuations of the turbulent flow and particle inertia, which cause the larger particles not to follow the flow. Since turbulent shear coagulation is only important for particles larger than several $\mu$m diameter sizes under conditions characterized by intense turbulence (Pnueli et al., 1991), its treatment is not considered in the model.

#### 2.3.5 Dry deposition and wet scavenging of particles

Different mechanical processes contribute to the deposition of particles, mainly Brownian diffusion, interception, inertial impaction and sedimentation. The effectiveness of the deposition process is usually described with the dry deposition velocity, $V_d$, which depends on the properties of the deposited aerosol particle, the characteristics of the air flow in the atmospheric surface layer and inside the thin layer of stagnant air adjacent to the surface (the so-called quasi-laminar sub-layer) and the properties of the surface. Three dry deposition schemes are included in the model: (1) Schack et al. (1985; hereafter SPF1985), (2) Kouznetsov and Sofiev (2012; hereafter KS2012), and (3) Hussein et al. (2012; hereafter HS2012). All schemes calculate size-dependent dry deposition velocities of particles.

The SPF1985 scheme considers dry deposition of particles by Brownian diffusion, interception and gravitational settling. This parameterisation is derived for deposition to completely rough surfaces, based on the analysis of several field studies.

The KS2012 scheme can consider the deposition to a vegetation canopy and can be used for smooth and rough surfaces. In the KS2012 scheme, the deposition pathway is split into the aerodynamic layer between heights $z_1$ and $z_0$ and the in-canopy layer. Within the aerodynamic layer the Monin-Obukhov profiles of turbulence are assumed. The in-canopy layer is assumed to be well mixed and to have a regular wind speed $U_{\text{top}}$ ($U_{\text{top}}$ is the wind speed at top of the canopy, i.e. at height $z_C$). The deposition in the in-canopy layer is treated as a filtration process. KS2012 defines a collection length scale to characterize the properties of rough surfaces. This collection length depends on the ratio $U_{\text{top}}/u^*$ and the effective collector size, $d_{col}$, of the canopy.

The HS2012 scheme is based on a three-layer deposition model formulation with Brownian and turbulent diffusion, turbophoresis and gravitational settling as the main particle transport mechanisms to rough surfaces. An effective surface roughness length $F^+$ is used to relate the roughness height to the peak-to-peak distance between the roughness elements of the surface.

The model user defines the roughness length, friction velocity near surface, and other parameters specific for the dry deposition schemes in an input file.

Figure 4 shows a numerical comparison of the deposition schemes for a typical rough urban surface, representative of a street canyon, using friction velocity, $u^* = 1.33\,\text{m}\,\text{s}^{-1}$ and roughness length $z_0 = 0.4\,\text{m}$. This example is chosen to illustrate the differences in the size-dependence of the dry deposition velocity, when all parameterizations are used with identical meteorological parameters and particle density. Effects of buildings on deposition are not considered.

Size-dependent deposition velocities calculated with the SPF1985 and KS2012 schemes agree within a factor of two, except for large particles. Both curves have a minimum in the size range 0.2–0.5 µm diameter. For the HS2012 scheme, an upper limit value of the effective surface roughness length ($F^+ = 2.75$) was chosen, adequate for dry deposition to rough environmental surfaces, that results in higher deposition velocities than for the other two schemes. For particles in the size range below 0.7 µm, the calculated deposition velocities are nearly independent of particle size.

Wet scavenging of particles is described with a simple parameterization of the scavenging rate for in-cloud removal of particles by accretion based on Pruppacher and Klett (1997). Nucleation mode particles are not scavenged. The wet scavenging

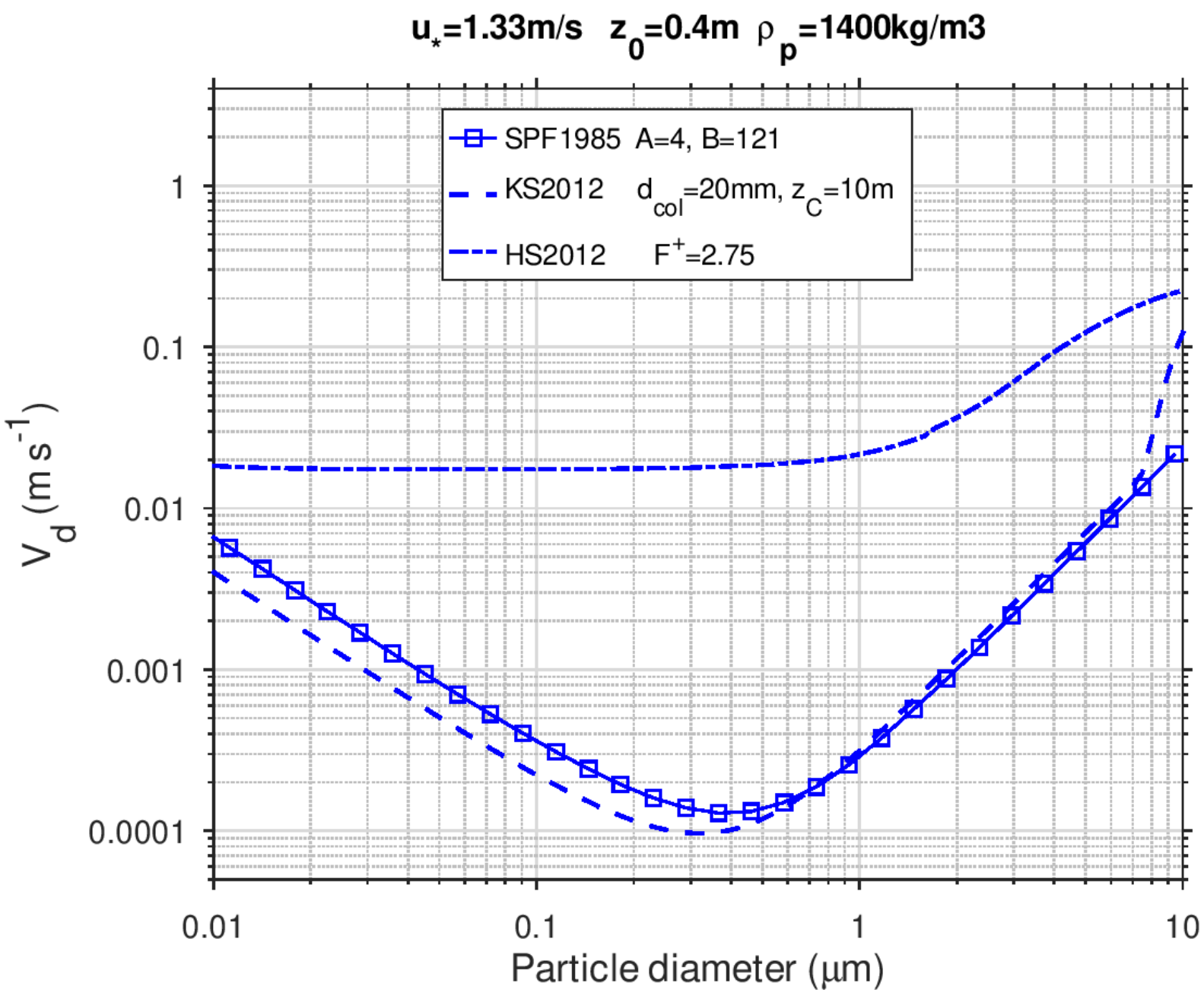

**Figure 4.** Dry deposition of particles over rough urban surface calculated with the SPF1985 scheme (solid line with squares), KS2012 scheme (lower dashed line), and HS2012 scheme (upper dashed line), using $u^* = 1.33\,\mathrm{m\,s^{-1}}$, $z_0 = 0.4\,\mathrm{m}$ and average particle density of $1400\,\mathrm{kg\,m^{-3}}$ . Specific parameter values are given in the legend.

rate of particles, $\lambda_{\mathrm{wet}}$, ($\mathrm{s^{-1}}$) is parameterized as:

$$\lambda_{\mathrm{wet}} = f_c \cdot 3.49 \times 10^{-4} \cdot P^{0.79}, \tag{14}$$

where $f_c$ is the volume fraction occupied by clouds, assumed to be 0.1, typical for the marine boundary layer. The precipitation rate $P$ ($\mathrm{mm\,h^{-1}}$) can be provided in the input by the model user and may vary with time.

### 2.3.6 Emission of particles

Emissions of primary particles are controlled by an input file. The prescribed particle emissions can occur either at a constant rate during the entire simulation period or time-varying as in the simulation of the “Urban Case”. The emitted size spectrum of particles and their chemical composition are defined by the model user.

Emissions of marine sea-salt particles are calculated on-line using the emission parameterization from Spada et al. (2013) which combines the number flux parameterizations of Mårtensson et al. (2003), Monahan et al. (1986) and Smith et al. (1993). Sea-salt particles are assumed to be composed of NaCl. A treatment of primary organic aerosol (POA) particle emissions from the ocean surface will be developed in the future. The parameterization of Spada et al. (2013) describes the size distribution of sea-salt particle emissions in terms of number for the diameter size range 0.2–10.0 µm. Sea-salt particle emissions in the model depend on wind speed (provided in the meteorological input), sea surface temperature (SST; user-provided value) and salinity

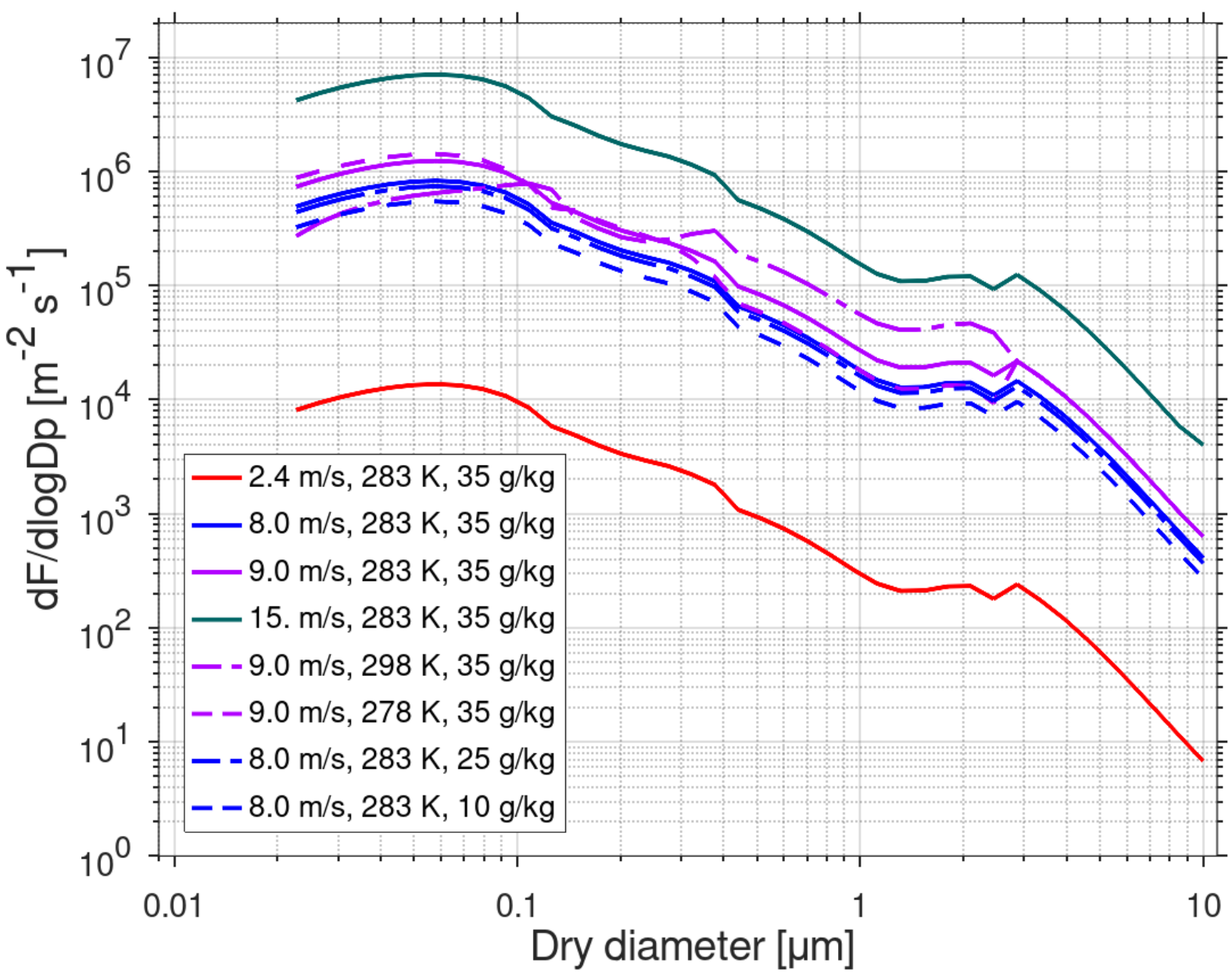

**Figure 5.** Sea-salt particle source function (size-dependent number flux, $F$) at different wind speed, sea surface temperature and salinity with the parameterization by Spada et al. (2013). The effect of wind speed is shown with the green, violet, blue and red solid lines (at SST = 283 K and salinity of 35 g kg$^{-1}$). The effect of SST is shown with the solid and dashed violet lines (at 9 m s$^{-1}$ and salinity of 35 g kg$^{-1}$). The effect of salinity is shown with the solid and dashed blue lines (at 8 m s$^{-1}$ and SST = 283 K).

(user-provided value). The wind speed dependence is described by the whitecap coverage relating to the 10 m wind speed and the fraction of the sea surface covered by whitecaps. Figure 5 shows the size-dependent sea-salt particle flux as a function of particle size for different conditions.

### 2.4 Dynamic partitioning of semi-volatile inorganic gases

Several aerosol models rely on thermodynamic equilibrium principles to predict the composition and physical state of inorganic atmospheric aerosols. Examples for thermodynamic equilibrium aerosol models, commonly applied in 3-D CTM, include EQUISOLV II (Jacobson, 1999), MARS (Binkowski and Shankar, 1995), ISORROPIA (Nenes et al., 1999), and AIM (Wexler and Clegg, 2002). However, in cases where the equilibrium timescale is long compared to the residence time of particles in a given environment, the thermodynamic equilibrium is not a good approximation (Meng and Seinfeld, 1996). A dynamic partitioning approach for the formation of secondary inorganic aerosol (SIA) is therefore preferable and is expected to give results that are more realistic.

To enable dynamic partitioning of semi-volatile inorganics in the model, the APC scheme for condensation/evaporation (Sect. 2.3.2) was extended with the PNG scheme (Jacobson, 2005a). The PNG scheme involves four steps: (1) calculation of

the growth of semi-volatile acidic gases by dissolution at moderate and high aerosol *LWC* (determined as total liquid water over all sizes); (2) calculation of the growth of semi-volatile acidic gases by condensation at low *LWC*; (3) calculation of the growth of non-volatile gases (such as $H_2SO_4$ when forming ammonium sulfate) at all *LWC*; and (4) equilibration of $NH_3/NH_4^+$ and pH between the gas phase and all particle size bins while conserving charge and moles.

In this implementation, the PNG scheme is coupled with the iterative equilibrium code MESA (Zaveri et al., 2005b) that calculates internal aerosol composition and the size-dependent solubility terms. Figure 6 illustrates the workflow for the coupling between the PNG scheme and the thermodynamic equilibrium module of the MOSAIC model. MESA computes aerosol phase state, temperature-dependent equilibrium coefficients, activity coefficients of electrolytes (solutes) and water activity coefficient in all size sections for solid, liquid and mixed phase aerosols. MESA solves the solid-liquid equilibrium by applying a pseudo-transient continuation technique to the set of ODE describing the precipitation reactions and dissolution reactions for each salt until the system satisfies the equilibrium or mass convergence criteria. The internal aerosol composition in MESA includes sodium ($Na^+$), chloride ($Cl^-$), potassium ($K^+$), calcium ($Ca^{2+}$), magnesium ($Mg^{2+}$), sulfate ($SO_4^{2-}$), $NH_3$/ammonium($NH_4^+$), and $HNO_3$/nitrate ($NO_3^-$) in the ionic, liquid and/or solid phases. MESA employs the Multicomponent Taylor Expansion Method (MTEM; Zaveri et al., 2005a) for estimating activity coefficients of electrolytes. MTEM calculates the mean activity coefficient of the electrolyte in a multicomponent solution on the basis of its values in binary solutions of all the electrolytes present in the mixture.

The PNG scheme solves the growth of particles by dissolution of semi-volatile compounds (here $HNO_3$ and HCl) when the *LWC* is moderate or high (here: $> 0.01\,\mu g\,m^{-3}$), i.e. a liquid solution pre-exists on the particle surface. The concentration change of particle compound $q$ (here either the dissolved, undissociated nitric acid plus the nitrate ion or the undissociated hydrochloric acid plus the chloride ion) due to dissolution in one size bin is:

$$\frac{dm_{q,i,t}}{dt} = k_{T,q,i,t-\Delta t}\left[C_{g,q,t} - S'_{q,i,t-\Delta t}\frac{m_{q,i,t}}{H'_{q,i,t-\Delta t}}\right], \tag{15a}$$

where $S'_{q,i}$ accounts for the Kelvin effect and $H'_{q,i}$ is the dimensionless effective Henry's law coefficient for the respective size bin. However, if a solid pre-exists in a particle size bin, condensation occurs, and:

$$\frac{dm_{q,i,t}}{dt} = k_{T,q,i,t-\Delta t}\left[C_{g,q,t} - S'_{q,i,t-\Delta t}C_{eq,q,i,t-\Delta t}\right]. \tag{15b}$$

The saturation vapour concentration $C_{eq,q,i}$ (short: SVC) varies continuously over the aerosol size distribution as a function of particle composition. The size-dependent SVC and the effective Henry's law coefficient are calculated in the MOSAIC solver at the beginning of the time step. The size-dependent SVC of $HNO_3$ and of HCl is determined by several processes (gas-ion reaction, solid-gas equilibrium, and solid-ion reactions). The minimum SVC arising in any of the processes is chosen for the calculation of the condensation term when a solid is present in a particle size bin. The gas concentration $C_{g,q}$ and the total dissolved concentration are unknowns in Eq. (15).

Integration of Eq. (15a) for one size bin over a time step $\Delta t$ gives (Jacobson, 2005a):

$$\frac{dm_{q,i,t}}{dt} = \frac{H'_{q,i,t-\Delta t}C_{g,q,t}}{S'_{q,i,t-\Delta t}} + \left(m_{q,i,t-\Delta t} - \frac{H'_{q,i,t-\Delta t}C_{g,q,t}}{S'_{q,i,t-\Delta t}}\right)\times\exp\left(\frac{-\Delta t k_{T,q,i,t-\Delta t}S'_{q,i,t-\Delta t}}{H'_{q,i,t-\Delta t}}\right). \tag{16}$$

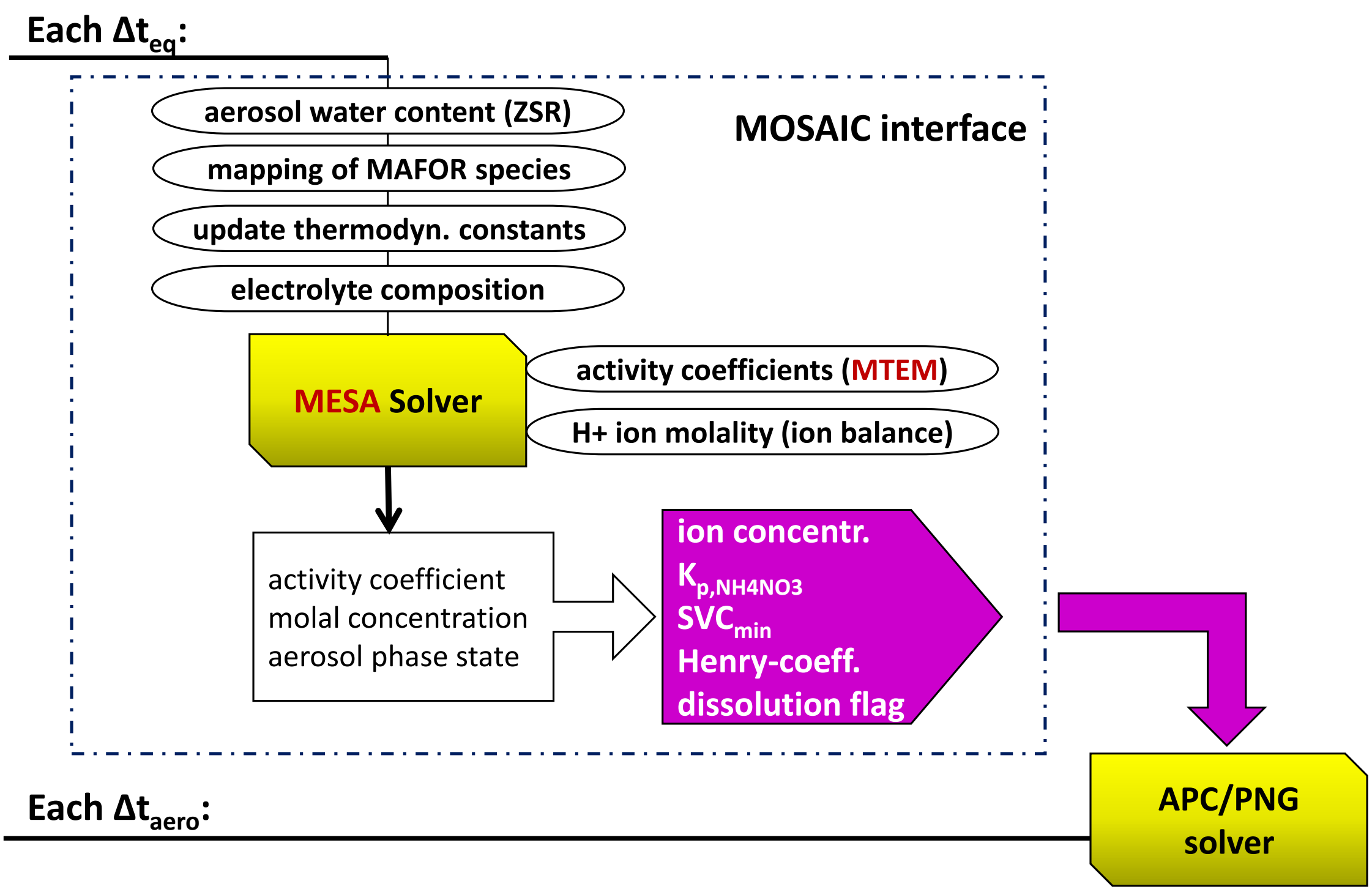

**Figure 6.** Workflow of the dynamic partitioning of semi-volatile inorganic gases. The MOSAIC interface is called every $\Delta t_{eq} = 120$ s, while the PNG solver is called every time step of the aerosol dynamic solver ($\Delta t_{aero}$). The MOSAIC interface outputs the gas-solid equilibrium coefficient for ammonium nitrate, the minimum saturation vapour concentration ($SVC_{min}$), the effective Henry's law coefficient, the ion concentrations and a dissolution flag (indicating if solid is present in a size bin or not) for each size bin of the particle population.

The final gas concentration of the semi-volatile acid and final particle concentration in each bin is obtained analogous to the APC scheme; with the solution described in Appendix B. The solution is unconditionally stable and mole conserving.

When the *LWC* is below 0.01 $\mu g\,m^{-3}$, the growth of nitric acid is treated as a condensation process rather than a dissolution process. The saturation vapour concentrations of $HNO_3$ and HCl are calculated considering the gas-solid equilibrium of ammonium nitrate and the gas-solid equilibrium of ammonium chloride as described in Jacobson (2005b). The solution for the coupled ammonia-nitric acid-hydrochloric acid system is then obtained from Eq. (6) and the growth by condensation is treated in the APC solver (Sect. 2.3.2). The condensation/evaporation of low volatility or non-volatile gases, such as $H_2SO_4$ and high molecular weight organics, is solved as a condensation process among all size bins independent of the aerosol *LWC*.

Following the growth calculation for the acidic gases, $NH_3$ is equilibrated with all ions and solids in all size bins of the aerosol phase, conserving charge among all ions, also for those that enter the liquid solution during the dissolution and condensation process. $NH_3$ is equilibrated with all size bins of the aerosol phase simultaneously, resulting in an exact charge balance among all ions in the solution and conserves mass of $NH_3$ between the gas phase and all particle size bins.

Following the ammonia calculation, an operator-split internal aerosol equilibrium calculation in the MESA solver is performed to recalculate aerosol ion, liquid, and solid composition, activity coefficients and Henry's law coefficients, accounting for all species in solution in each size bin. In order to reduce the computational time, the liquid solution terms and composition are updated at longer time intervals than the aerosol dynamic solver time step ($\Delta t_{\mathrm{aero}}$). The operator-split time interval between growth and equilibrium is 115 s in the current implementation. An advantage of the PNG scheme is that it can be applied at a long time interval (several minutes) without causing oscillatory behaviour in the numerical solution (Jacobson, 2005a). Such oscillatory behaviour at a long time step was observed in an earlier dissolution solver (Jacobson, 1997b), that did not treat the condensation (dissolution) of acid and base separately.

The coupled PNG-MOSAIC system was tested in numerical scenario calculations with different initial concentrations of $NH_3$ and $HNO_3$ at RH = 90 % ("Case 3"), as described in Supplement Sect. S3. The initial conditions for Case 3 were adopted from the tests of the PNG-EQUISOLV II scheme presented in Jacobson (2005a). In a simulation with high gas-phase concentrations of both compounds, an equilibrium was reached within about 6 h, and the time-dependent summed concentrations of inorganic aerosol species matched the equilibrium levels from EQUISOLV II fairly well. Under low nitrate conditions, the performance of the PNG-MOSAIC scheme is very accurate. Under low ammonia conditions, the simulated time-series of summed concentrations of inorganic aerosol species from MAFOR are smooth, showing no sign of oscillation, and the model achieves similar accuracy as PNG-EQUISOLV II.

### 2.5 Absorptive partitioning of organic vapours

The new concept for SOA formation in MAFOR v2.0 relies on the 2-D VBS framework introduced by Neil Donahue and co-workers (Donahue et al., 2011). This classification uses the carbon oxidation state and the saturation concentration of the pure compound to define the organic aerosol composition in a two-dimensional space. The 2-D VBS is able to represent the variety of organic aerosol components in the atmosphere and their conversion due to ageing chemistry.

A hybrid approach of condensation/evaporation (Sect. 2.3.2) and the absorptive partitioning into an organic liquid is used to treat condensation to an organic mixture considering non-ideal solution behaviour. For absorptive partitioning, the equilibrium gas-phase concentration (or saturation concentration) of the condensing organic vapour can be obtained from the following relation (Bowman et al., 1997):

$$C_{eq,q} = \frac{1}{K_{\mathrm{om},q}} \cdot \frac{m_{tot,q}}{f_{\mathrm{om}} m_{tot,p}}, \tag{17}$$

where $m_{tot,p}$ is the total particle mass concentration, $m_{tot,q}$ is the total mass concentration of compound $q$ in the particle, $f_{\mathrm{om}}$ is the fraction of absorbing organic material in the aerosol, and $K_{\mathrm{om},q}$ ($\mathrm{m^3\,\mu g^{-1}}$) is the absorption partitioning coefficient of the

compound. Using the relation for the mass-based absorption partitioning, (Donahue et al., 2006), Eq. (17) can be rewritten as:

$$C_{eq,q} = C_q^* \cdot \frac{m_{tot,q}}{f_{\mathrm{om}} m_{tot,p}}, \tag{18}$$

with the effective saturation mass concentration $C_q^*$ (in µg m$^{-3}$) of compound $q$:

$$C_q^* = C_q^0 \gamma_{\mathrm{om},q}, \tag{19}$$

where $\gamma_{\mathrm{om},q}$ is the activity coefficient of the individual compound (solute) in the organic mixture (solvent). A simplifying assumption of the 2-D VBS framework is that the activity coefficient is a function of the average carbon fraction (O:C) of the organic aerosol as well as the properties of the individual organic solute. Donahue et al. (2011) give an empirical relation to estimate the activity coefficient $\gamma_{\mathrm{om},q}$ for organic mixtures (at 300 K):

$$\log_{10} \gamma_{\mathrm{om},q} = -2 b_{\mathrm{CO}} n_{\mathrm{M}} \left[ \left(\mathrm{f}_{\mathrm{C}}^{q}\right)^2 + \left(\mathrm{f}_{\mathrm{C}}^{s}\right)^2 - 2 \mathrm{f}_{\mathrm{C}}^{q} \mathrm{f}_{\mathrm{C}}^{s} \right], \tag{20}$$

where $b_{\mathrm{CO}}$ is an empirical constant for the carbon-oxygen non-ideality ($b_{\mathrm{CO}}$ = -0.3), $n_{\mathrm{M}}$ is the size of the solute calculated as sum of carbon and oxygen atoms, $\mathrm{f}_{\mathrm{C}}^{q}$ is the carbon fraction of the individual solute and $\mathrm{f}_{\mathrm{C}}^{s}$ is the carbon fraction of the solvent. The activity coefficient for compound $q$ depends exponentially on the size of the solute while the non-ideality is driven by the differences between the carbon fraction in the solvent and the solute. The formulation of the activity coefficient neglects the role of water or other inorganics in the absorbing material. The effect of these constituents may be treatable within the 2-D VBS framework in the future.

Three classes of organic compounds are represented in the model: oxidized secondary biogenic organics, oxidized secondary aromatic organics and primary emitted organics. Each class is divided in three volatility levels resulting in a total of nine lumped gaseous SOA precursors. Formation of secondary organic compounds is coupled to the gas-phase chemistry of biogenic VOCs (isoprene, monoterpenes) as well as aromatic VOCs (toluene, xylene, trimethylbenzene). The lumped SOA precursors are produced in the gas-phase oxidation reactions via their molar stoichiometric yields. They can undergo oxidative ageing and/or oligomerization. Primary emitted organics can either undergo oxidative ageing or fragmentation. Figure 7 presents a scheme of SOA formation reactions in the model.

Extremely low volatility organic compounds (ELVOC) may play an important role in new particle formation. Ehn et al. (2014) have demonstrated the significant formation of ELVOC with a branching ratio of ca. 7 % in the reaction of $\alpha$-pinene with ozone ($O_3$). The compounds have been identified as highly oxygenated molecules (HOM). Their formation is induced by one attack of ozone in the initial reaction of the monoterpene, followed by an autoxidation process involving molecular oxygen. In the model, the production of ELVOC from monoterpenes (represented by BELV) is simplified by assuming direct formation in the reaction of the monoterpene with $O_3$. The formation of HOM in the reaction of aromatics with hydroxyl (OH) radicals occurs either via an autoxidation mechanism or via multi-generation OH-oxidation steps (Wang et al., 2020). Again, only direct formation of ELVOC (represented by AELV) in the initial reaction of toluene with OH radicals is implemented here. The model further assumes that BELV and AELV are the products from the oligomerization reaction of more volatile organics. It is possible to implement a more detailed treatment of the autoxidation mechanism in the future.

**VOC oxidation**

BVOC + Ox → BRO2 + … + $\alpha_1$ BELV
BRO2 + HO2 → $\alpha_2$ BSOV+ $\alpha_3$ BLOV
BRO2 + NO → $\alpha_4$ BSOV+ $\alpha_5$ BLOV
AVOC + Ox → ARO2 + … + $\beta_1$ AELV
ARO2 + HO2 → $\beta_2$ ASOV+ $\beta_3$ ALOV
ARO2 + NO → $\beta_4$ ASOV+ $\beta_5$ ALOV

**Oxidative ageing**

| | | |
|---|---|---|
| BSOV + OH → BLOV | $k_{agin}$ = 4x10$^{-11}$ cm$^3$ s$^{-1}$ | [Tsimpidi et al., 2010] |
| ASOV + OH → ALOV | $k_{agin}$ = 4x10$^{-11}$ cm$^3$ s$^{-1}$ | [Tsimpidi et al., 2010] |
| BLOV + OH → BELV | $k_{agin}$ = 4x10$^{-11}$ cm$^3$ s$^{-1}$ | [Tsimpidi et al., 2010] |
| ALOV + OH → AELV | $k_{agin}$ = 4x10$^{-11}$ cm$^3$ s$^{-1}$ | [Tsimpidi et al., 2010] |
| PIOV + OH → PLOV | $k_{agin}$ = 2x10$^{-11}$ cm$^3$ s$^{-1}$ | [Lambe et al., 2009] |
| PSOV + OH → PELV | $k_{agin}$ = 2x10$^{-11}$ cm$^3$ s$^{-1}$ | [Lambe et al., 2009] |

**Oligomerization**

| | | |
|---|---|---|
| BSOV → BELV | $k_{olig}$ = 9.6x10$^{-6}$ s$^{-1}$ | [Carlton et al., 2010] |
| BLOV → BELV | $k_{olig}$ = 9.6x10$^{-6}$ s$^{-1}$ | [Carlton et al., 2010] |
| ASOV → AELV | $k_{olig}$ = 9.6x10$^{-6}$ s$^{-1}$ | [Carlton et al., 2010] |
| ALOV → AELV | $k_{olig}$ = 9.6x10$^{-6}$ s$^{-1}$ | [Carlton et al., 2010] |

**Fragmentation**

| | | |
|---|---|---|
| PELV → PSOV | $k_{frag}$ = 5.0x10$^{-4}$ s$^{-1}$ | [Lim and Ziemann, 2009] |

**Figure 7.** Chemical reactions involved in SOA formation. BRO2 and ARO2 stand for all the peroxy radicals of the respective biogenic or aromatic VOC. The molar stoichiometric yields $\alpha_1, \ldots \alpha_5$ and $\beta_1, \ldots, \beta_5$ are the formation yields of SOA precursors in the gas-phase reaction of biogenic and aromatic VOCs, respectively. Oligomerization and fragmentation reactions are approximated with first order rate constants (Tsimpidi et al., 2010; Lambe et al., 2009; Carlton et al., 2010; Lim and Ziemann, 2009). The nine lumped organics are: BSOV (biogenic semi-volatile compound), BLOV (biogenic low volatility compound), BELV (biogenic extremely low volatility compound), ASOV (aromatic semi-volatile compound), ALOV (aromatic low volatility compound), AELV (aromatic extremely low volatility compound), PIOV (primary intermediate volatility compound), PSOV (primary semi-volatile compound), and PELV (primary extremely low volatility compound).

The implementation of the 2-D VBS framework requires a series of input parameters for each SOA precursor, namely: number of carbon atoms, number of oxygen atoms, saturation concentration $C^0$, and enthalpy of vaporization. The user-provided $C^0$ value (in µg m$^{-3}$) of the lumped organic compound is then used to compute the saturation vapour concentration according to Eqs. (17)–(20).

## 2.6 Numerical solution of the aerosol dynamics

The model solves the particle number and mass concentration distribution of a multicomponent aerosol using the full stationary (fixed) sectional method. The fixed sectional method (Gelbard and Seinfeld, 1990; Tsang and Rao, 1988) is computationally efficient and advantageous when treating continuous nucleation of new particles, relevant for the modelling of new particle

Open Access EGU

formation. The method is also convenient for the combined treatment of nucleation, emission, coagulation and particle transport, because the particle volume in one size section is always constant (Korhonen et al., 2004). This is achieved by a splitting procedure for the particle growth that determines the fraction of particles in one size bin that will grow to the next size bin. However, this splitting procedure is prone to numerical diffusion causing a wider particle size distribution with lower peak concentrations than the accurate solution. Relevant alternative sectional methods are the full-moving structure (Gelbard, 1990), the hybrid structure (Jacobson and Turco, 1995) and the moving centre structure (Jacobson, 1997a), which all eliminate the numerical diffusion arising from the splitting between size sections. The full-moving structure allows the particles to grow to their exact size. However, the full-moving structure causes problems if new particle formation is considered. The disadvantage of the hybrid structure is that if the particles gain or lose non-volatile material, they must be fitted back to the fixed grid. The moving centre structure allows the particle size to vary in a section within certain boundaries. It causes some numerical diffusion due to averaging of moved particles with pre-existing ones in a section.

Korhonen et al. (2004) tested different sectional structures in the simulation of the particle distribution during a new particle formation event and found that the hybrid structure was most vulnerable to numerical diffusion upon particle growth. The moving centre structure permitted fairly realistic treatment of the particle evolution (Korhonen et al., 2004). The ADCHEM model uses the moving centre structure due to its good performance when the size distribution is represented by only a few size sections (see Table 1). In the SALSA model, the moving centre structure is used for particles below 730 nm in diameter, whilst for particles larger than that, fixed size sections are used. In SALSA, the particle size spectrum is divided into three subranges based on the size. This enables variation in including or excluding microphysical aerosol processes and chemical components in simulations in each subrange based on the relevance of the process in the range. For instance, in the lowest subrange cloud-processing can be neglected and particles contain only sulfate and organic matter.

Because of the advantages when simulating new particle formation, the fixed structure has been chosen for MAFOR (Karl et al., 2011). A fixed sectional grid on the diameter coordinate is used where the number of size sections can be selected by the model user. By using a high number of size sections, the numerical diffusion can be largely reduced. Karl et al. (2011) showed that in an 80-hour simulation of the particle distribution in the arctic marine boundary layer, the final number distribution for the model using 60 size bins closely agreed to the solution of the model using 120 sections. To determine the number of size bins that are necessary to accurately represent an urban particle size distribution, numerical calculations using different number of size sections were performed (Supplement, Sect. S2). This test ("Case 1") confirmed that the model using 60 bins performs very well in comparison to a sectional representation using 160 bins (the reference in Case 1), although slight spreading of the nucleation mode due to numerical diffusion could be noted. For lower size resolution, the discretization errors were more relevant, leading to a broader nucleation mode with peak diameter at smaller size.

In model simulations, size bins are evenly distributed on a logarithmic scale, ranging from the smallest diameter of 1 nm to the largest diameter of 10 µm. It is possible to use a different maximum diameter (in the range 1–10 µm). Typical model applications in plume dispersion simulations use 120 size sections to represent the aerosol size distribution in the size range 0.001–1.0 µm, resolving the nucleation mode at molecular level. Simulations are initiated with the particulate mass concentrations of the aerosol constituents in four aerosol modes: nucleation mode (Nuc; diameter range 1–25 nm), Aitken mode (Ait;

diameter range 25–100 nm), accumulation mode (Acc; diameter range 100–1000 nm) and coarse mode (Coa; diameter range 1–10 µm). The initial mass concentrations of the lognormal modes are distributed over the size bins (Jacobson, 2005b):

$$m_{q,i} = \frac{M_{\mathrm{A},q}\Delta D_{p,i}}{D_{p,i}\sqrt{2\pi}\ln\sigma_{\mathrm{A}}}\exp\left[-\frac{ln^2\left(D_{p,i}/GMD_{m,\mathrm{A}}\right)}{2\ln^2\sigma_{\mathrm{A}}}\right] \tag{21}$$

where $D_{p,i}$ is particle diameter of section $i$, $\Delta D_{p,i}$ the corresponding width of the section; $M_{\mathrm{A},q}$ and $\sigma_{\mathrm{A}}$ are the mass concentration of the constituent $q$ and geometric standard deviation of the lognormal mode A, respectively. The initial number concentration in each mode is then matched by varying the geometric-mean mass diameter, $GMD_{m,\mathrm{A}}$.

Due to full-stationary structure, collision of particles from section $k$ with particles from section $j$ generates a particle which has a volume between those of two sections $i$ and $i+1$, and needs to be partitioned between the two bins, as described in Appendix C. A semi-implicit method is applied to coagulation which yields an immediate volume-conserving solution with any time step (Jacobson, 2005b). Though particle number is not exactly conserved, the error in number concentration reduces when the number of bins to describe the size distribution is increased (Karl et al., 2011). Condensation/evaporation of vapours results in the redistribution of particles between adjacent size sections. Number concentration in section $i$ increases when particles from section $i-1$ grow by condensation or particles from section $i+1$ shrink due to evaporation. It decreases when particles from section $i$ change volume by condensation or evaporation of vapour.

Considering the presence of a supersaturated vapour (e.g. $H_2SO_4$), stable clusters containing a certain number of monomers, $g^*$, will form continuously at the rate of neutral or ion-induced nucleation (see Sect. 2.3.3), denoted by $J_{\mathrm{nuc}}(t)$. Then coagulation, heterogeneous condensation/evaporation of vapour on/from particles of size $i \geq g^*$ and nucleation of $g^*$-mers are distinct processes. The time evolution of the particle number concentration (in $\mathrm{m}^{-3}$) and mass concentration (in $\mu\mathrm{g\,m}^{-3}$) of all aerosol constituents in section $i$ (with $i = g^*, g^*{+}1, \ldots, g^{**}{+}N_B$) can be written as discrete general dynamic equations in Eqs. (22) and (23).

$$\begin{aligned}
\frac{dN_i}{dt} \quad = \quad & \frac{1}{\upsilon_i}\sum_{j=g^*}^{i}\left(\sum_{k=g^*}^{i-g^*} f_{k,j,i}\upsilon_k K_{j,k} N_j N_k\right) \\
& \textit{coagulation gain} \\
& -N_i \sum_{j=g^*}^{N_B+g^*}\left(1-f_{i,j,i}\right)K_{i,j}N_j \\
& \textit{coagulation loss} \\
& -\left(\frac{\sum_{q=1}^{N_C} I_{q,i}}{\upsilon_{i+1}-\upsilon_i}\right)N_i + \left(\frac{\sum_{q=1}^{N_C} I_{q,i-1}}{\upsilon_i-\upsilon_{i-1}}\right)N_{i-1} \\
& \textit{condensation/evaporation} \\
& +J_{\mathrm{nuc}}(t)\delta_{g^*,i} \\
& \textit{nucleation}
\end{aligned}$$

$$
\begin{aligned}
&-\left(\lambda^i_{\mathrm{dry}}-\lambda^i_{\mathrm{wet}}\right)N_i\\
&\textit{dry deposition / wet scavenging}\\
&-\lambda_{\mathrm{dil}}\left(N_i-N_{\mathrm{bg},i}\right)\\
&\textit{dilution}\\
&+\frac{\left(\sum_{q=1}^{N_C}Q^i_{m,q}(t)/\left(\rho_q c_{\mathrm{v}}\right)\right)}{\upsilon_i}\frac{1}{H_{\mathrm{mix}}}\\
&\textit{emission}
\end{aligned}
\tag{22}
$$

$$
\begin{aligned}
\frac{dm_{q,i}}{dt} = &\ \frac{1}{\upsilon_i}I_{q,i}\sum_{q=1}^{N_C}m_{q,i}\\
&\textit{condensation/evaporation acting on total mass}\\
&-\left(\frac{\sum_{q=1}^{N_C}I_{q,i}}{\upsilon_{i+1}-\upsilon_i}\right)m_{q,i}+\left(\frac{\sum_{q=1}^{N_C}I_{q,i-1}}{\upsilon_i-\upsilon_{i-1}}\right)m_{q,i-1}\\
&\textit{condensation/evaporation acting on size distribution}\\
&+\sum_{j=g^*}^{i}\left(\sum_{k=g^*}^{i-g^*}f_{k,j,i}K_{j,k}m_{q,j}\frac{m_{q,k}}{M_k}\right)\\
&\textit{coagulation gain}\\
&-m_{q,i}\left(\sum_{j=g^*}^{N_B+g^*}\left(1-f_{i,j,i}\right)K_{i,j}\frac{m_{q,j}}{M_j}\right)\\
&\textit{coagulation loss}\\
&+g^*\frac{10^6 MW_q}{N_A}J_{\mathrm{nuc}}(t)\delta_{g^*,i}\delta_{q,\mathrm{nuc}}\\
&\textit{nucleation}\\
&-\left(\lambda^i_{\mathrm{dry}}-\lambda^i_{\mathrm{wet}}\right)m_{q,i}\\
&\textit{dry deposition / wet scavenging}\\
&-\lambda_{\mathrm{dil}}\left(m_{q,i}-m_{\mathrm{bg},q,i}\right)\\
&\textit{dilution}\\
&+Q^i_{m,q}(t)/H_{\mathrm{mix}}\\
&\textit{emission}
\end{aligned}
\tag{23}
$$

where $f$ is the volume fraction of the intermediate volume of the colliding particles, $\delta$ is the Kronecker delta function, $\lambda_{\mathrm{dry}}$ ($\mathrm{s^{-1}}$) is the dry deposition rate, $\lambda_{\mathrm{dil}}$ ($\mathrm{s^{-1}}$) is the dilution rate, $N_{\mathrm{bg},i}$ is the number concentration of background particles in the

same size section, $m_{\mathrm{bg},q,i}$ is the mass concentration of background particles of compound $q$ in the same section, $Q^i_{m,q}(t)$ is the mass-based emission rate (µg m$^{-2}$ s$^{-1}$), $H_{\mathrm{mix}}$ is the height of the simulation box (m), $\rho_q$ is the density of compound $q$ (kg m$^{-3}$), and $c_\mathrm{v}$ is a conversion factor to convert kg into µg. In Eq. (23), $M_k$ is the total mass of a particle (µg) in section $k$ (i.e. the sum of the masses of its individual components), $M_j$ is the mass of a particle in section $j$, and $q_{\mathrm{nuc}}$ indicates that the compound is able to nucleate (e.g. $H_2SO_4$). The first term on RHS of Eq. (23) describes the effect of condensation/evaporation of a vapour on the total aerosol mass. The second and third terms on RHS take into account that the mass of the individual constituent increases/decreases and consequently the mass concentration distribution moves on the diameter coordinate.

The discrete equations describing the change of particle number and mass concentration with time are solved with forward finite differences. In plume dispersion simulations, MAFOR uses a time step of 0.1 s for the integration of chemistry and of the aerosol processes, which is sufficiently small, when compared to the typical time scales in the range 0.5–4 s for dilution in exhaust plumes (Ketzel and Berkowicz, 2004). When simulating an air parcel along multiple day trajectories and for chamber experiments, the time step is 5 s.

### 2.7 Previous applications of MAFOR in plume dispersion studies

In this section, published applications of MAFOR version 1 in plume dispersion studies and the previously developed procedure for treating the dilution term in the model are presented. An evaluation of MAFOR version 2, including the new features, against experimental data and two aerosol dynamics models is presented in section 3.

The MAFOR model version 1 has been used in the European TRANSPHORM ("Transport related Air Pollution and Health impacts – Integrated Methodologies for Assessing Particulate Matter") project to examine the influence of aerosol transformation processes on PN concentrations in several European cities (Karl et al., 2016; Kukkonen et al., 2016). Dry deposition and coagulation were found to be generally relevant on the neighbourhood scale, but less so in efficient dispersion conditions. Sensitivity tests with the model showed that coagulation causes removal of particles with $<$ 25 nm diameter between roadside and ambient. Particle removal was further enhanced when the fractal nature of soot aggregates and the combined effect of van der Waals and viscous interactions were considered.

For the treatment of dilution of vehicular exhaust gases and particles in combination with aerosol transformation processes on the neighbourhood scale, it is practical to divide the exhaust dilution near roadways into two distinct dilution stages: the first stage (tailpipe-to-road) characterized by traffic-generated turbulence and a second stage (road-to-ambient) where atmospheric turbulence prevails (Zhang and Wexler, 2004). The dilution ratio in the first stage can reach up to about 1000:1 in around 1–3 s, while the dilution ratio in the second stage is commonly of the order of about 10:1 on a time scale of about 10 min. A detailed simulation of the first stage would require the use of LES to explicitly describe the plume turbulent dispersion and accounting for the fluctuations in the wake of the vehicles (e.g., Chan et al., 2008). However, in practical applications, the early plume phase has been mainly treated using analytic equations for the jet/plume development up to a few seconds (e.g., Vignati et al., 1999). Due to the rapid temperature decrease immediately after exhaust release, the formation of a nucleation mode has already occurred within the time-scale of the first dilution stage (Rönkkö et al., 2007).

In the study of Karl et al. (2016), model simulations with MAFOR for the road-to-ambient particle evolution were initialized with the particle size distribution measurements at the roadside and at an urban background station. It was assumed that emission of primary exhaust particles and nucleation processes had already occurred before the exhaust plume reached the air quality (AQ) monitoring site, located a few metres away from the street. The horizontal particle dilution parameterization was defined by a numerical power function, $y = ax^{-b} = a(Ut)^{-b}$ , where $x$ (in m) is the distance from the roadside and $U$ is the horizontal wind speed ($\mathrm{m\,s^{-1}}$) perpendicular to the road (Pohjola et al., 2007). Typical values of the dispersion parameters $a$ and $b$ were chosen to represent different meteorological dispersion regimes. Assuming a circular plume cross-section, the particle dilution rate as a function of time is then simply $\lambda_{\mathrm{dil}} = b/t$.

The dispersion parameters can either be derived from dispersion models or from concentration measurements (typically of $NO_X$) at several distances perpendicular to the road. The applied treatment of particle dilution assumes a well-mixed state within each cross-wind cross-section of the plume. The simple dilution model coupled with the aerosol dynamics model has been tested and evaluated in an earlier study (Keuken et al., 2012) simulating the particle evolution downwind of a motorway under free dispersion conditions. The comparison of the modelled total PN and size distributions with measurements at different distances from the motorway gave reasonable agreement.

The model has also been applied to study the formation of particles in the exhaust of a diesel engine, equipped with an oxidative after-treatment system (Pirjola et al., 2015), consisting of a dilution unit and an ageing chamber. The rapid dilution and cooling in the dilution unit was described with empirical parameterizations, where temperature follows the exponential curve of the Newtonian cooling and dilution is modelled by using an exponential equation for the dilution ratio, as in Lemmetty et al. (2006). These functions have been implemented in MAFOR and in AEROFOR. Modelled particle number size distributions of the two models were in good agreement with each other and with measurements after 2.7 s exhaust dilution.

In a study of ship exhaust plumes, MAFOR was applied to determine the in-plume number size distribution and chemical composition of ultrafine particles at different distances from passenger ships (Karl et al., 2020). The dilution of aerosol particles in the ship exhaust plume was approximated using dilution parameters provided by the 3-D atmospheric dispersion model EPISODE-CityChem (Karl et al., 2019). The aerosol dynamics model was used to compute the particle number and mass distributions during the second dilution stage, as a function of the distance from the ship stack along the centreline of the ship plume. Dilution in the first stage, when rapid cooling and expansion occurs, was calculated with the jet plume model of Vignati et al. (1999), assuming a circular cross-section of the plume. Neglecting the removal of particles by coagulation during the first-stage dilution was estimated to introduce an error of 10–15 % in the computed PN concentrations. The particle evolution in the ship plume during the second dilution stage was computed with the aerosol dynamics model considering nucleation, condensation/evaporation, coagulation of particles, dry deposition of particles, gas-phase chemistry within the plume and mixing of the air parcel with gases and particles from the background. Modelled PN concentrations agreed within 50 % with measured PN concentrations when a peak in the signal was detected that related to the ship passage.

Recently, the MAFOR model has been utilized to investigate the particle number concentrations induced by aviation emissions in the surrounding communities of Zurich Airport (Zhang et al., 2020). The offline coupling between the atmospheric dispersion model and MAFOR was achieved through the plume dilution curve, which was approximated by fitting a power-law

function using the dispersion results and then adopted by MAFOR for the aerosol dynamics calculations. The plume dilution curve was analysed based on the centreline concentration of the plume. The particle evolution in the aviation exhaust was calculated with the aerosol dynamics code using the obtained dilution curve, in conjunction with meteorological data (humidity, temperature, precipitation and wind speed) and the background PN concentration. *Kinetic nucleation* of $H_2SO_4$, condensation/evaporation, coagulation, deposition and mixing of the air parcel with the background particles were considered in the model simulations. The results suggested that particles between 10 and 30 nm contributed significantly to the particle number concentration. The predicted PN concentrations were within a factor of two of the measurements.

## 3 Methods of evaluation against experimental data

### 3.1 Experimental data for the Urban Case in Helsinki, 2010

The "Urban Case" scenario for the evaluation of the MAFOR model version 2 was developed as a plume dispersion study inside a half-open street canyon, where emission from vehicular traffic and dilution with background air are the key processes in modifying PN concentrations and size distributions. Mobile and stationary measurements during a street canyon campaign (Pirjola et al., 2012) in winter 2010 (November–December 2010) in Helsinki, Finland, performed as part of the Finnish national research program MMEA (Measurement, Monitoring and Environmental Assessment, 2010–2014), were used to construct the Urban Case scenario. Measurements with the mobile laboratory van, called "Sniffer", were obtained while driving back and forth the main street Mannerheimintie (MA) and in the side streets. Stationary measurements were performed at the sidewalks and inner courts. MA passes through the city of Helsinki in the north-western direction. There are four vehicular traffic lanes in the considered street segment (two in each direction), and in addition, there are two tramway tracks in the middle of the street. The mean traffic flow in the busy sections of MA is about 40,000 vehicles per workday and the fraction of heavy duty vehicles has been estimated to be 10 %.

For the Urban Case, measurements on 13 December 2010 in the microenvironment M2 (as defined in Pirjola et al., 2012; see Fig. 8a), during the afternoon traffic rush hour between 5 and 6 pm local time, were selected. The length of this street canyon is 230 m. In M2, the buildings downwind of the main street are oriented perpendicular to MA, and the distance between the buildings is ~22 m (Fig. 8b). On the other side of the street, buildings are parallel to MA. The buildings are ~21 m tall and the width of the canyon is 38 m, leading to the aspect ratio of 0.55. Although the aspect ratio is relatively shallow and MA is a half-open environment at the place of measurements, it can be considered as a street canyon due to the large traffic intensity (Vardoulakis et al., 2003).

Measurements with Sniffer for dispersion studies in M2 were taken during the driving times on the second lane (outwards from the centre of Helsinki, A), during the standing times (5–10 minutes) downwind of MA in the space between the buildings (B, C, and D), and during the driving times on the side street (E, towards the city centre) shown in Fig. 8b. Monitoring with Sniffer included measurements of particles (particle number concentration, size distribution, particulate matter ($PM_{2.5}$), and BC), and gases (NO, $NO_2$, and $NO_X$); see details of the instruments in Pirjola et al. (2012). A weather station on the roof of the van at a height of 2.9 m above ground level provided measurements of the temperature and relative humidity as well as wind

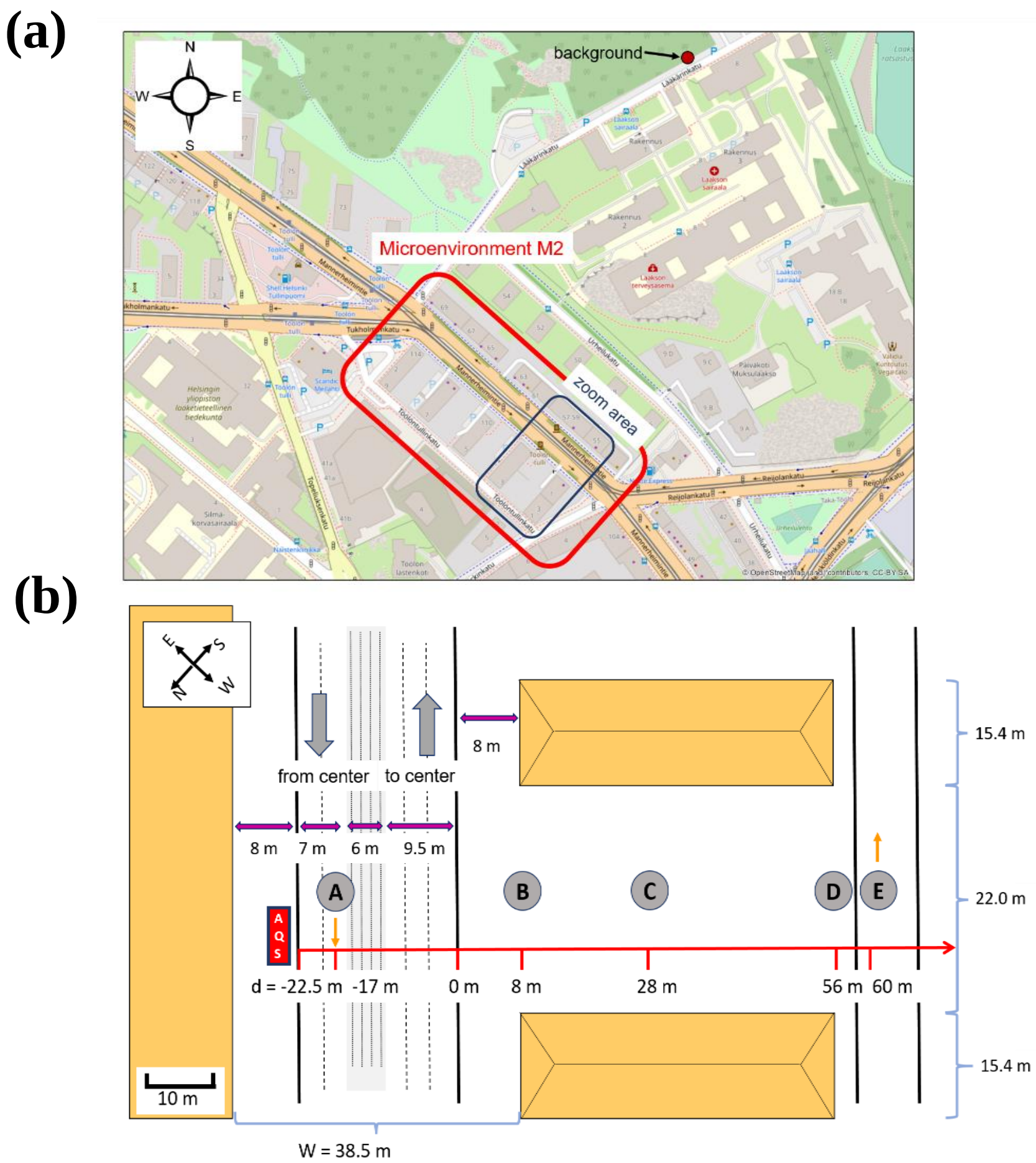

**Figure 8.** Urban case microenvironment M2: a) map showing microenvironment M2, the street canyon zoom area and the background measurement location. (© OpenStreetMap contributors 2021. Distributed under the Open Data Commons Open Database License (ODbL) v1.0. See http://www.openstreetmap.org/); b) the horizontal geometries and Sniffer measurement locations in the M2 zoom area. Red arrow shows distance *d* from kerbside in m.

speed and direction. A GPS device saved the van's speed and location. Background concentrations of particles were measured by Sniffer at Lääkärinkatu; 300 m north from M2; additionally, background air concentrations of $O_3$, NO, and $NO_2$ were monitored at the nearby urban background site Kallio-2 (60° 11' 14.85" N, 24° 57' 02.04" E). Measurements of NO, $NO_2$, $PM_{2.5}$, $PM_{10}$, and BC from air quality monitoring station (AQS) operated by the Helsinki Region Environmental Service Authority (HSY), located at the pavement of M2 (60° 11' 24.51" N, 24° 54' 56.81" E) (Fig. 8b) (Fig. 8b) were also available.

Hourly meteorological data was estimated in this study, using the meteorological preprocessor MPP-FMI (Karppinen et al., 2000). The MPP-FMI results for the selected day are based on meteorological measurements at Helsinki Vantaa airport (60.3267 N; 24.95675 E), a site which has been found to be meteorologically representative for the whole of the Helsinki Metropolitan Area. Data from MPP-FMI includes the parameters defining the atmospheric stability, in addition to wind data. However, the meteorological data measured at Sniffer during the standing times in M2 was used whenever possible, as it better represented the local conditions in the street canyon. The dispersion situation for the Urban Case scenario is evaluated at Sniffer inlet height for particles, i.e., at a height of 2.4 m above the ground level.

### 3.2 Configuration of the simulation

In the Lagrangian air parcel simulation we assume that the initial height of the air parcel volume corresponds to the situation where vehicular exhaust gases and particles have been diluted in a time scale of less than 0.5 s after release from tailpipe (Pohjola et al., 2007), and the process of initial nucleation in the exhaust has been finalized. The initial air parcel height was assumed to be 0.80 m (Pohjola et al., 2007). As in previous plume dispersion studies for exhaust dilution near roadways (see Sect. 2.7), a two-stage dilution process was applied for the Urban Case scenario. The initial air parcel ("sub-scale box" in Fig. 9) is initialized with concentration of particles and gases in the background air. In the first dilution stage, the dispersion of the plume and the growth of the (diluted) exhaust plume is calculated with the jet plume model of Vignati et al. (1999) which takes into account the turbulence generated by traffic, the atmospheric turbulence and the entrainment of fresh air due to the jet effect of the exhaust gas. In the second dilution stage, when the air parcel reaches the kerbside and is further transported to the ambient environment, atmospheric turbulence dominates the plume dispersion. Growth of the air parcel and dilution parameters are calculated with a line source dispersion model that considers the geometry of the street canyon.

The combination of the dispersion model and the aerosol process models was straight forward: the jet plume model and the street canyon dispersion model provided the required parameters for the dilution function of the Lagrangian air parcel, while the aerosol process models then allowed to analyse the aerosol transformation within the temporally expanding volume of the plume. Figure 9 illustrates the coupling of the plume dispersion models with the aerosol dynamics models. The dilution of particles in the moving air parcel is divided in two regimes, i.e. the first between the sub-scale box from emission source to kerbside and the second between kerbside and the ambient environment ("street environment box"). The change of particle number concentration in a size section due to dilution with background air during the first stage is expressed by:

$$\left.\frac{dN_i}{dt}\right|_{\mathrm{dil1}} = -\frac{\left(N_i - N_{bg,i}\right)}{D_{\mathrm{R}}^2} \cdot \frac{dD_{\mathrm{R}}}{dt}. \tag{24}$$

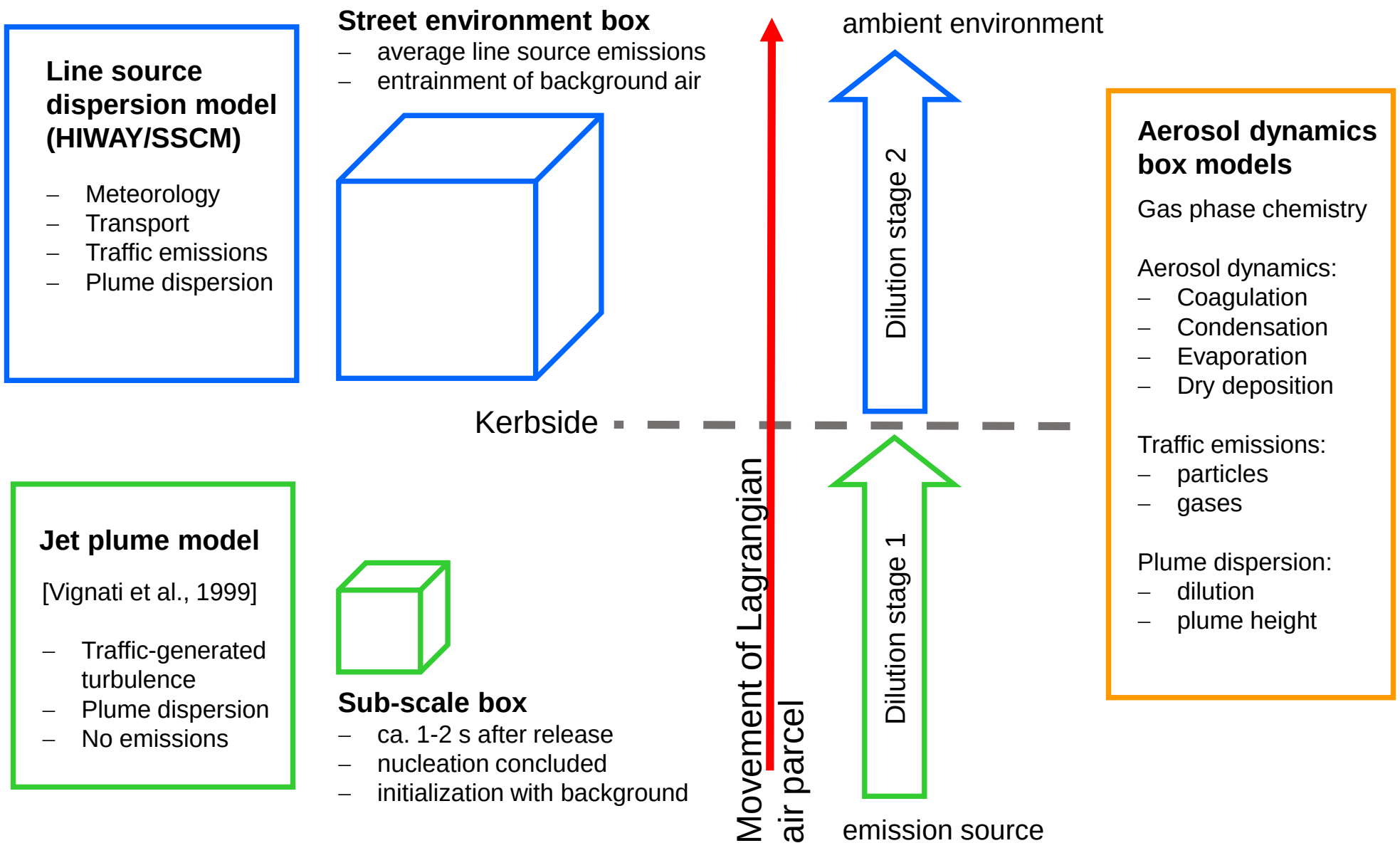

**Figure 9.** Coupling of the dispersion models and aerosol dynamics models.

During the second stage, it is expressed by:

$$\left.\frac{dN_i}{dt}\right|_{\mathrm{dil2}} = -\frac{b}{t}\left(N_i - N_{bg,i}\right). \tag{25}$$

The dilution ratio $D_{\mathrm{R}}$ in the vehicle exhaust plumes increases approximately linearly with time during the first seconds of the dilution. Details on the calculation of the plume height as function of the air parcel transport time and the dilution functions are given in Appendix E. The two dilution functions were implemented in MAFOR and the other Lagrangian-type aerosol process models that were used in the comparison for the Urban Case scenario. The dispersion situation in the street canyon was first evaluated using the simplified street canyon model (SSCM), a component of the urban dispersion model EPISODE-CityChem (Karl et al., 2019). This street canyon model follows in most aspects the Operational Street Pollution Model (OSPM; Berkowicz et al., 1997), but simplifies the geometry of the street canyon. Then the dilution parameters for the second stage were derived from the simulated concentrations obtained from the street canyon model using line source emissions of total PN in both directions of the street.

In the Lagrangian simulation, a continuous flux of vehicular emissions to the moving air parcel occurs during the times when the air parcel is transported over the lanes. The air parcel is released at $d$ = -22.5 m ($d$ is the distance from kerbside) and transported over the street (with the street geometry in Fig. 8b). All gaseous and particulate constituents of the air parcel are diluted during the transport, with the rate of dilution changing at kerbside ($d$ = 0 m). The air parcel receives emissions while passing over the two lanes in outwards direction, then is only diluted while passing over the tram tracks, and then receives
**Table 4.** Overview of meteorological input and initial conditions for the "Urban Case".

| Meteorological parameters | | | Initial concentrations | | |
|---|---|---|---|---|---|
| Input parameter | Value | Source | Input parameter | Value | Source |
| air pressure [mbar] | 1025.8 | MPP-FMI | initial $O_3$ [$cm^{-3}$] | $3.7\times10^{11}$ | HSY Kallio-2 |
| air temperature [K] | 260.1 | Sniffer | initial NO [$cm^{-3}$] | $1.8\times10^{11}$ | HSY Kallio-2 |
| rel. humidity [%] | 59.8 | Sniffer | initial $NO_2$ [$cm^{-3}$] | $7.6\times10^{11}$ | HSY Kallio-2 |
| wind speed [$m\,s^{-1}$] | 1.0 | Sniffer | initial $SO_2$ [$cm^{-3}$] | $3.0\times10^{10}$ | Pohjola et al. (2007) |
| wind direction [°] | 25.5 | MPP-FMI | | | |
| mixing height [m] | 168.0 | MPP-FMI | initial SVOC [$cm^{-3}$] | $2.5\times10^{8}$ | Pohjola et al. (2007) |
| friction velocity [$m\,s^{-1}$] | 0.52 | MPP-FMI | | | |
| surface roughness [m] | 0.40 | HIWAY-2 | total PN conc. [$cm^{-3}$] | $2.41\times10^{4}$ | Sniffer measured 300 m north of M2 |
| inverse M.O. length [$m^{-1}$] | $5.4\times10^{-3}$ | MPP-FMI | | | |
| Vert. gradient potential temperature [$K\,m^{-1}$] | 0.104 | MPP-FMI | | | |

again emissions while passing over the three lanes in direction to the city. After passing $d = 0$ m, the air parcel is freely diluted, with no influence from buildings and ground surfaces (smooth terrain assumption).

The composition of the air parcel was initialized with particle size distribution data from Sniffer measurements in the background air, 300 m north of M2 (Fig. 8a). The chemical composition of the initial aerosol was based on the urban background aerosol described in Pohjola et al. (2007; table 2 therein). Table 4 summarizes the meteorological input and initial conditions for the Urban Case scenario.

Emission factors of gases and particulates for the Urban Case were adopted from Kurppa et al. (2020; table 3 therein). Kurppa et al. (2020) applied a number emission factor of $EF_{PN} = 4.22\times10^{15}$ kg fuel$^{-1}$. A fuel consumption per vehicle (veh) of 9.8 l per 100 km is assumed here for conversion of emission factors in unit kg fuel$^{-1}$ to unit veh$^{-1}$ km$^{-1}$. From this we obtain a particle emission factor of $4.14\times10^{14}$ veh$^{-1}$ km$^{-1}$. This emission factor is 34 % lower than the estimate from Gidhagen et al. (2003) of $6.23\times10^{14}$ veh$^{-1}$ km$^{-1}$, that has been used in the model simulations of the LIPIKA campaign (Pohjola et al., 2007). Emission of total particle numbers were distributed over the particle size spectrum by utilizing the number size distribution when Sniffer was driving on Mannerheimintie to North so that the modelled size distribution after 5.5 m distance from start (on the middle of lane 2; $d = -17$ m) matched with the measured size distribution on lane 2.

Exhaust particles were assumed to be composed of organic carbon (OC) and BC with constant modal OC-to-BC ratios; nucleation mode: 100:0, Aitken mode: 80:20, accumulation mode 1 (Acc1): 40:60, accumulation mode 2 (Acc2): 60:40), as in Karl et al. (2016). The emission factors for vehicle exhaust gases, $EF_{NO}$, $EF_{NO2}$, $EF_{H2SO4}$ and $EF_{SVOC}$, and were $4.94\times10^{-4}$, $1.39\times10^{-4}$, $1.0\times10^{-7}$ and $3.9\times10^{-7}$ g m$^{-1}$ veh$^{-1}$, respectively (SVOC is the sum of semi-volatile organic vapours), adopted from Kurppa et al. (2020). The emission factors for the two line sources were then weighted by the vehicle count in each

**Table 5.** Processes and employed parameterizations in each of the aerosol process models.

| Aerosol Transformation processes | MAFOR | AEROFOR | SALSA |
|---|---|---|---|
| Coagulation | Brownian coagulation, spherical particles | Brownian coagulation, spherical particles | Brownian coagulation, spherical particles |
| Condensation / evaporation | $H_2SO_4$, SVOC (primary emitted) | $H_2SO_4$, SVOC (primary emitted) | $H_2SO_4$, SVOC (primary emitted) |
| Dry deposition | Hussein et al. (2012), horizontal surfaces | Schack et al. (1985), horizontal surfaces | Zhang et al. (2001), horizontal surfaces |

direction. Traffic flow was 1462 veh $h^{-1}$ in outward direction and 1085 veh $h^{-1}$ in city direction (Pirjola et al., 2012). Emissions of particles and gases in the outward direction were shared equally between the two lanes and the emissions toward the city were shared equally between the lanes in this direction. To calculate the particle emission rates (particles $cm^{-3}$ $s^{-1}$) and gas emission rates (molecules $cm^{-3}$ $s^{-1}$), the emission factors were divided by the width of the lanes to one direction and by the air parcel box height (plume height), assuming the air in the box is well mixed. The plume height, dilution rate and emission rate of exhaust particles during the Urban Case simulation is plotted in Fig. E1.

### 3.3 Comparison with other aerosol models

Results from simulations of the Urban Case scenario with MAFOR were compared to results from two other aerosol dynamics models, AEROFOR and SALSA. Processes included in the simulation of the Urban Case for the respective aerosol process models are summarized in Table 5. MAFOR, AEROFOR and SALSA consider the condensation of $H_2SO_4$ and organic vapours emitted from the vehicles, in addition to Brownian coagulation and dry deposition. The dilution of particles and gases according to Eqs. (24–25) was implemented in AEROFOR and SALSA, ensuring that the same dilution schemes were applied in all models. The three sectional aerosol dynamics model used 120 bins for the diameter range between 1 and 1000 nm, a model time step of 0.01 s for the aerosol dynamics, and a time step of 0.5 s for changes of the dilution rate. The model evaluation was done without inclusion of sulphuric acid-water nucleation. A preliminary run with MAFOR showed that freshly nucleated particles formed by the atmospheric nucleation of $H_2SO_4$ emitted from the vehicles, based on nucleation rates using the Määttänen et al. (2018a) parameterization, did not grow beyond diameter of 2 nm in size.

Emissions of particles were inserted differently in the models. In AEROFOR and SALSA particle emissions were distributed over the respective size sections, while in MAFOR the emitted particles as function of size were fitted with a log-normal distribution and attributed to four modes in terms of mass and modal composition (see Eq. (21)). SVOC emissions were treated slightly different in the models: in AEROFOR they were represented by one compound with properties of adipic acid, in SALSA as semi-volatile organic carbon (Kurppa et al., 2019), in MAFOR they were split each half to PIOV (intermediate volatility; $C^0 = 1.0\,\mu g\,m^{-3}$ at 298 K) and PSOV (semi-volatile; $C^0 = 0.01\,\mu g\,m^{-3}$ at 298 K).

LNMOM-DC treats simultaneous coagulation and dispersion from a continuous emission source (Anand and Mayya, 2015; Sarkar et al., 2020). With respect to the coagulation-dispersion system, the parameterization scheme for near-source aerosol dynamics was used as reference for the relevance of coagulation in the Urban Case simulation.

## 4 Results

### 4.1 Model evaluation against experimental data

#### 4.1.1 Comparison with other aerosol dynamics models and experimental data

The model performance of MAFOR version 2 was evaluated in terms of total particle number, number size distributions, total particulate matter and composition (only BC), by comparison against experimental data and against results from two other aerosol dynamics models in an urban environment. Model runs for the "Urban Case" were performed with the three aerosol dynamics under identical conditions for plume dispersion, using the same configuration in the models, to the extent that this was possible (Sect. 3.3). The focus of the model evaluation lies on the analysis of aerosol processes that are relevant in urban environments. Experimental data on particle number and mass concentrations from observations within the street canyon M2, obtained with the Sniffer mobile lab were used for the comparison. Statistical performance indicators for the model-observation (M-O) comparison were: index of agreement (IOA), coefficient of efficiency (COE), and the mean absolute error (MAE). The definitions of these indicators is given in Appendix F. In short, IOA is a refined index (Willmott et al., 2012) that spans values between -1 and +1 with values close to 1.0 representing better model performance. A COE value of 1.0 indicates a perfect agreement, while negative values of COE indicate that the model predicts the observed variation less effectively than the mean of the observations. The M-O comparison was based on a four-point dataset obtained at the locations A, B, C, and D (see Fig. 8b) where Sniffer was positioned during the measurement campaign. Location E was excluded from the analysis, because it appears that the measurements at E were affected by emissions from outside the street canyon. The statistics were prepared for each of the models. Note that model results are instantaneous concentrations whereas experimental data represents an average over a longer time period (typically 5–10 min). Therefore, it is worth noticing that the large variation in the traffic situations, especially while Sniffer was driving on the main street and on the side street, might have affected the experimental results.

First, the predicted total PN concentrations from the three aerosol dynamics models were compared against measurements by SMPS (Scanning Mobility Particle Sizer; combined with a nano-SMPS). Fig. 10 shows the modelled time series of total PN from the three models and the measured total PN (including $1\sigma$ standard deviation), as function of downwind distance, which is the distance from the edge of the road ($d$ = -22.5 m; Fig. 8), i.e. the starting point of the simulation, in downwind direction. All models matched the total PN concentration at street level and the reduction of PN concentrations with increasing distance from the street, as the vehicular exhaust plume is diluted in the open space between the buildings. The total PN curve predicted by SALSA deviates from the other models after kerbside; in 120 m downwind distance total PN remains 52 % higher than in the other models. The statistical evaluation revealed that AEROFOR and MAFOR were in slightly better agreement with the measurement data than SALSA, although the differences in performance are small. Measured and modelled concentration

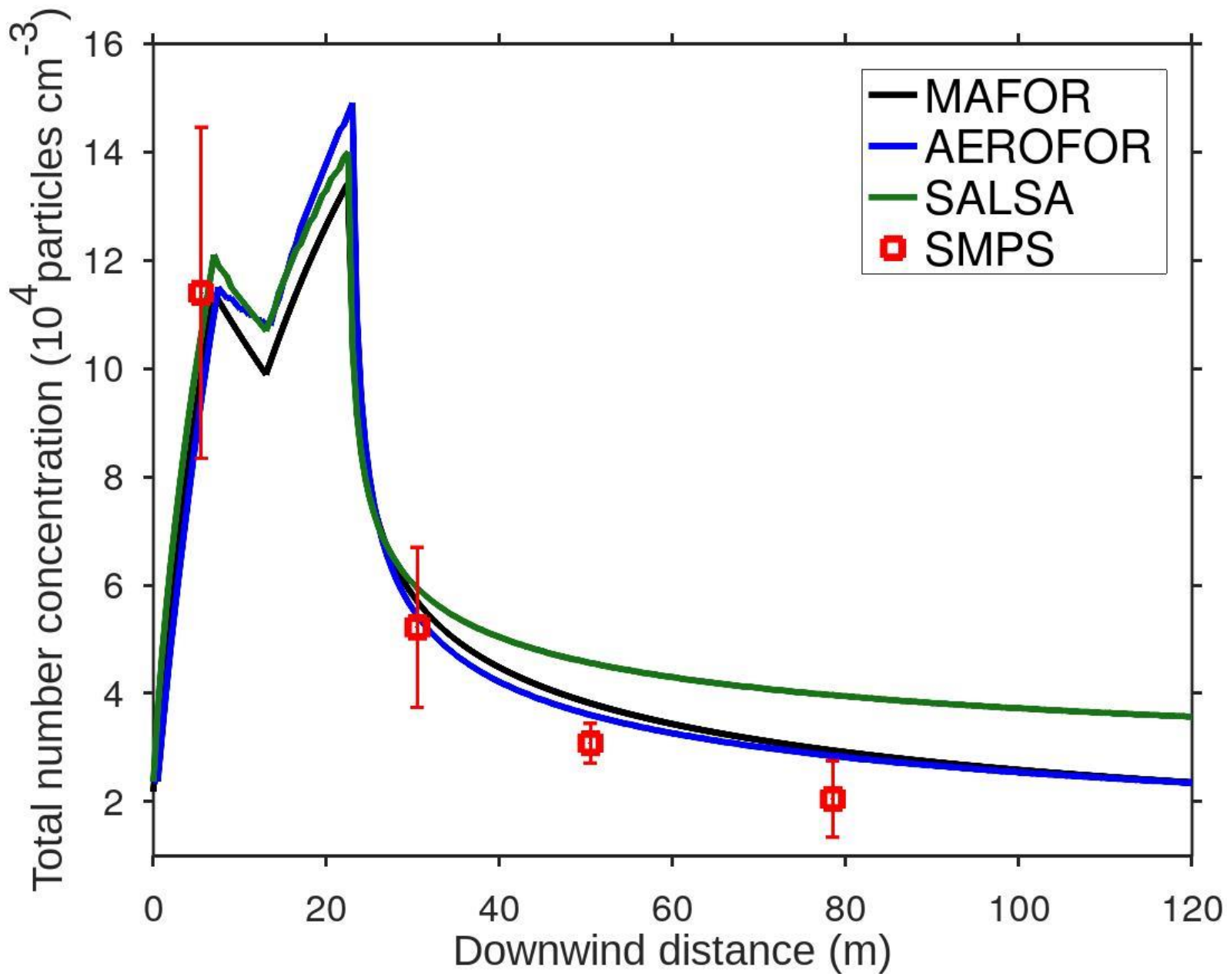

**Figure 10.** Comparison of total particle number concentrations as function of downwind distance in the "Urban Case" scenario. Model results from the aerosol dynamics models MAFOR, AEROFOR and SALSA. Measurement data from SMPS at points A, B, C and D (error bars indicate $1\sigma$ standard deviation).

**Table 6.** Comparison of modelled total number concentration from different aerosol dynamics models and measured data together with statistical indicators. Standard deviation of measurements are given in round brackets.

| Data source | Point A $[cm^{-3}]$ | Point B $[cm^{-3}]$ | Point C $[cm^{-3}]$ | Point D $[cm^{-3}]$ | MAE $[cm^{-3}]$ | IOA [–] | COE [–] |
|---|---|---|---|---|---|---|---|
| MAFOR | $9.85 \times 10^4$ | $5.70 \times 10^4$ | $3.82 \times 10^4$ | $2.95 \times 10^4$ | $0.92 \times 10^4$ | 0.85 | 0.69 |
| AEROFOR | $9.39 \times 10^4$ | $5.43 \times 10^4$ | $3.60 \times 10^4$ | $2.84 \times 10^4$ | $0.83 \times 10^4$ | 0.85 | 0.70 |
| SALSA | $10.6 \times 10^4$ | $5.94 \times 10^4$ | $4.56 \times 10^4$ | 3.97x104 | $1.23 \times 10^4$ | 0.79 | 0.59 |
| Measurements | $11.4 \times 10^4$ $(\pm 3.06 \times 10^4)$ | $5.22 \times 10^4$ $(\pm 1.48 \times 10^4)$ | $3.08 \times 10^4$ $(\pm 0.37 \times 10^4)$ | $2.05 \times 10^4$ $(\pm 0.71 \times 10^4)$ | | | |

values at the four measurement points, together with the statistical performance parameters for all models, are displayed in Table 6.

Next, the modelled and measured particle number size distributions were compared at the four point locations A, B, C and D (Fig. 11). Modelled number size distributions in point A, at street level, to a large extent reflect how the vehicular particle emissions were distributed over the relevant size range. SALSA and AEROFOR, both using a bin-wise distribution of emit-

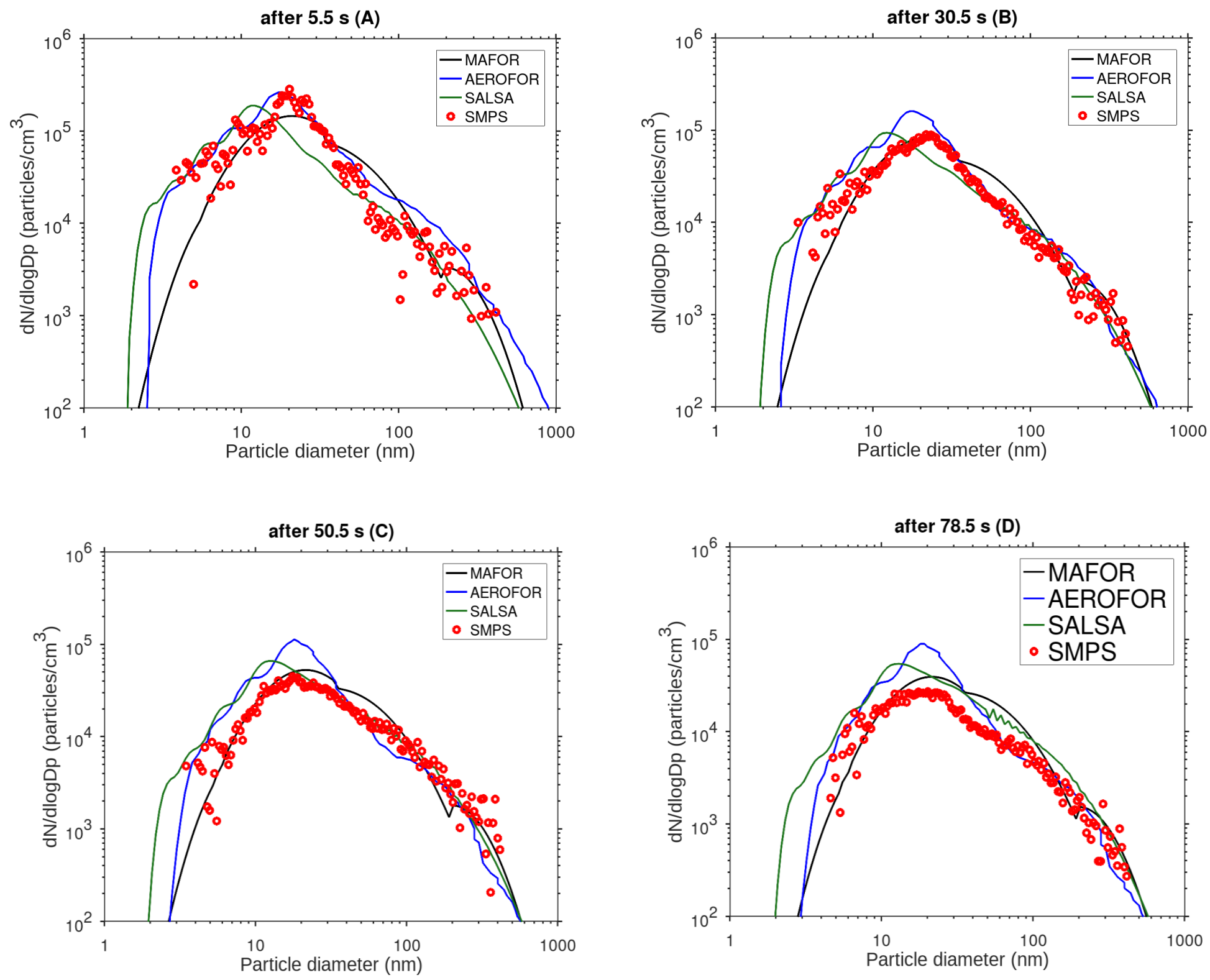

**Figure 11.** Comparison of particle number size distributions in the "Urban Case" scenario. Modelled size distributions from the aerosol dynamics models MAFOR, AEROFOR and SALSA and measured size distributions from SMPS are shown at location point A, B, C and D.

ted particles, better capture the measured size distribution in point A, especially in the size range < 20 nm in diameter, than MAFOR using a mode-wise distribution. Clearly, the bin-wise distribution allows for a more accurate representation of particle emissions. However, the particle size distribution of SALSA does not match the peak of the measured size distribution at 15–30 nm, in contrast to MAFOR and AEROFOR. At the second location, point B, in 8 m distance from the street, particle concentrations have been strongly diluted (Fig. 10) and the modelled distributions are now closer to each other and the measured distribution. In points C and D, both modelled size distributions from AEROFOR and SALSA apparently overestimate number concentrations in the size range 7–20 nm compared to the measurements, indicating that the small particles are not removed efficiently enough. Number concentrations of larger particles (> 100 nm in size) in greater downwind distance (points C and D) show a large variability that was not captured by the models. It cannot be excluded that sources of large particles from outside the street canyon contributed to the number size distribution measured at point C and D.

The measured size distribution from SMPS spans the size range of 3–420 nm in diameter with a size resolution of 138 bins. For the M-O comparison, the modelled size distributions ($dN/d(\log_{10})D_p$) were synchronized to the size resolution of the measured size distribution by linear interpolation. The statistical comparison of size distribution was evaluated separately at points
**Table 7.** Statistical performance indicators for the comparison of modelled and measured number size distributions for each location point A, B, C, D and the average performance (mean IOA and mean COE). MAE is given in particles $cm^{-3}$.

| Model | Indicator | Point A | Point B | Point C | Point D | **Average** |
|---|---|---|---|---|---|---|
| MAFOR | MAE | $2.18 \times 10^4$ | $0.64 \times 10^4$ | $0.50 \times 10^4$ | $0.44 \times 10^4$ | |
| | IOA | 0.79 | 0.85 | 0.77 | 0.71 | **0.78** |
| | COE | 0.58 | 0.70 | 0.55 | 0.41 | **0.56** |
| AEROFOR | MAE | $2.05 \times 10^4$ | $1.56 \times 10^4$ | $1.33 \times 10^4$ | $1.15 \times 10^4$ | |
| | IOA | 0.80 | 0.63 | 0.40 | 0.24 | **0.52** |
| | COE | 0.60 | 0.26 | -0.20 | -0.52 | **0.03** |
| SALSA | MAE | $3.30 \times 10^4$ | $1.14 \times 10^4$ | $0.71 \times 10^4$ | $0.87 \times 10^4$ | |
| | IOA | 0.68 | 0.73 | 0.68 | 0.42 | **0.63** |
| | COE | 0.36 | 0.46 | 0.36 | -0.16 | **0.26** |

A, B, C, and D. Results of the performance evaluation at the four points and the average performance is presented in Table 7. It turns out that MAFOR and AEROFOR performed better in the prediction of the size distribution at street level (point A) compared to SALSA. However, the deviation between modelled size distributions from AEROFOR and the measured ones becomes larger with increasing downwind distance. All models show the weakest predictive capability at point D. Overall modelled size distributions from MAFOR are in good agreement with the measured distributions (IOA range: 0.71–0.85; mean IOA: 0.78) and the model has the smallest MAE at points B–D. MAFOR best reproduced the development of the number size distribution with increasing distance from road edge. The weaker performance of SALSA (mean IOA: 0.63) is mainly due to the lower peak diameter of the modelled size distributions compared to the measured size distributions (Fig. 11).

Modelled and measured total particle mass and BC concentrations were also compared. Modelled $PM_1$ (particles with < 1 µm in diameter) from MAFOR and SALSA was compared against measurement data of $PM_1$ from ELPI (Electrical Low Pressure Impactor), assuming particle density of 1000 kg $m^{-3}$. MAFOR outputs mass concentrations and mass size distributions, while SALSA outputs volume distributions of total mass and components. From AEROFOR no output of particle mass or volume is available. Comparison of $PM_1$ from ELPI to $PM_{2.5}$ measured with DustTrak at Sniffer indicates that the mass of super-micron particles contributed little to $PM_{2.5}$ (Fig. 12a). The DustTrak measurements had large relative uncertainties, which can be attributed to short-term variations caused by passing exhaust plumes at street level, for instance from heavy duty vehicles, or from other sources outside of the street canyon. Measurements of BC with aethalometer similarly show high uncertainty at street level and in point E (Fig. 12b).

Modelled $PM_1$ from SALSA considerably overestimated measured $PM_1$. Modelled $PM_1$ from MAFOR was closer to the measurements, although modelled $PM_1$ at point A was 45 % higher than measured $PM_1$ (Fig. 12a). The statistical indicators show that MAFOR (MAE = 2.29 µg $m^{-3}$, IOA = 0.26, COE = -0.48) and SALSA (MAE = 13.0 µg $m^{-3}$, IOA = -0.76, COE = -

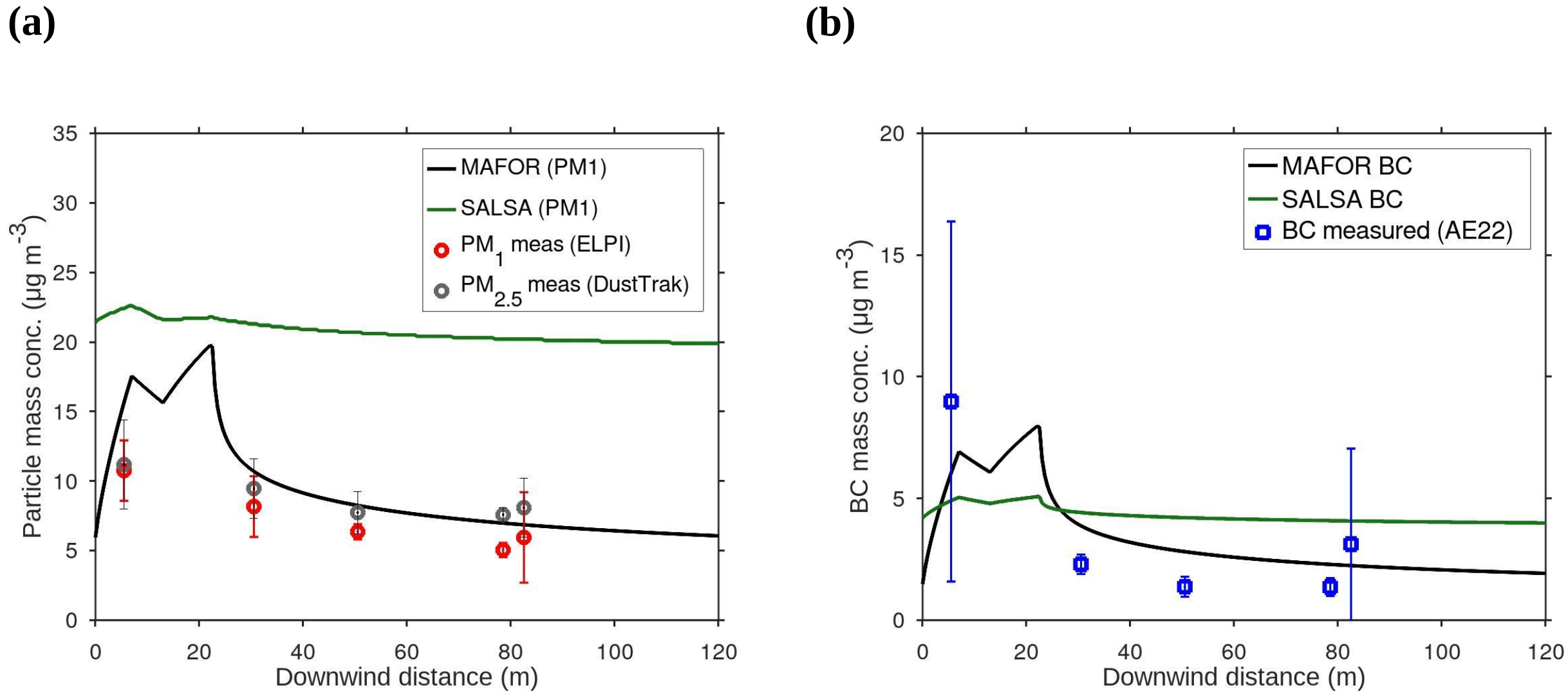

**Figure 12.** Comparison of total particle mass and black carbon concentrations in the "Urban Case" simulation: a) modelled particulate matter ($PM_1$) from MAFOR and SALSA together with measured $PM_1$ from ELPI (assuming particle density of 1000 kg m$^{-3}$) and $PM_{2.5}$ from DustTrak (error bars represent 1$\sigma$ standard deviation); b) modelled BC from MAFOR and SALSA together with measured BC from aethalometer (AE22, Magee Scientific; error bars represent 1$\sigma$ standard deviation). Measurement data was obtained with the mobile lab Sniffer at location points A–E. Note that point E was excluded from the M-O comparison.

7.41) both have a weak performance in predicting the variation of the observations. However, the absolute error of MAFOR model results is still acceptable and the IOA indicates better agreement with observations than the SALSA model.

Measurements of black carbon concentrations show a steeper decline between point A and D than the modelled BC concentrations from the two aerosol process models (Fig. 12b). MAFOR overestimated measured BC concentrations between point B and D, but captured the decreasing trend in measured BC. The statistical evaluation shows that MAFOR (MAE = 1.72 µg m$^{-3}$, IOA = 0.69, COE = 0.37) performs slightly better than SALSA (MAE = 2.94 µg m$^{-3}$, IOA = 0.46, COE = -0.07) in predicting variation of observed BC. Due to the large variation in the uncertainty bars of measured BC, results from the M-O comparison for BC should be regarded with caution.

The comparison of gas phase concentrations of condensing vapours was of particular interest to analyse discrepancies in the magnitude of condensation/evaporation between the models. In the absence of measurements of these compounds, only the model results were compared with each other. Figure 13 shows the comparison of modelled gas phase concentrations of sulphuric acid and semi-volatile organics (sum of condensable organic vapours) calculated by the three aerosol dynamics models. While modelled peak concentrations of condensable vapours at street level were very similar among the models, differences can be noted in greater downwind distance. For $H_2SO_4$, the maximum deviation of a single model from the model

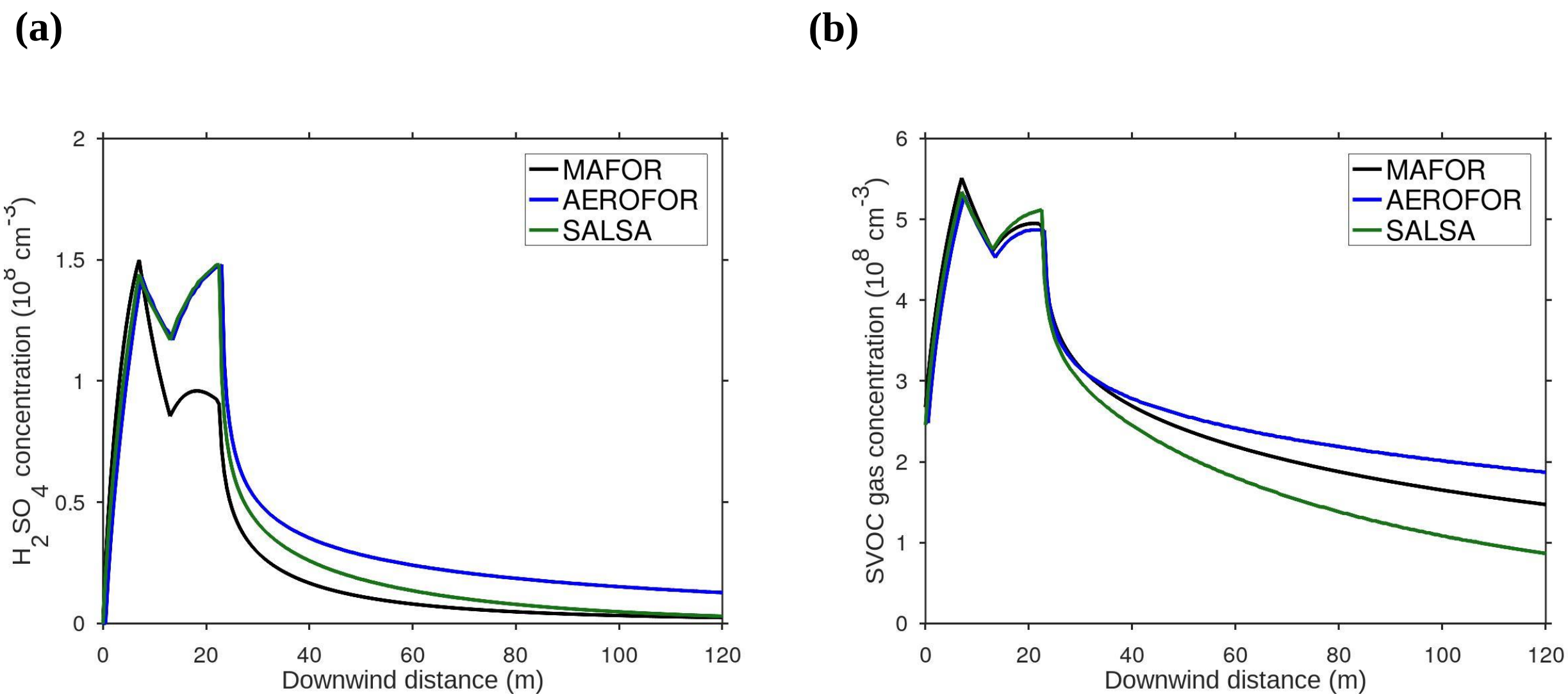

**Figure 13.** Comparison of modelled gas-phase concentrations as function of downwind distance in the "Urban Case" simulation: a) sulphuric acid and; b) sum of semi-volatile organics (here short SVOC).

mean was ±3.0 % at peak concentration, but ±96 % in 100 m distance from road edge. For SVOC, the maximum deviation was ±2.4 % at peak concentration and ±32 % in 100 m distance.

Modelled $H_2SO_4$ from MAFOR shows a notably lower second peak (at around 18 m downwind distance) than the other two models. This appears to be a sign for faster condensation of $H_2SO_4$ to the particle population in the simulation with MAFOR compared to the other models. Applied vapour pressure and accommodation coefficient of $H_2SO_4$ were not identical in the different aerosol models. The relevance of condensation in MAFOR simulations will be discussed in more detail in Sect. 4.1.3.

### 4.1.2 Importance of aerosol processes

The importance of aerosol processes was evaluated for total PN concentrations by comparing the model runs including all processes to model runs excluding one of the aerosol processes, i.e. either condensation/evaporation, dry deposition, or coagulation, and excluding all aerosol processes (dilution only). The evaluation was based on the change of total PN concentration between point A and point D relative to the PN concentration at point A:

$$\Delta PN = (PN_D - PN_A)/PN_A \times 100. \qquad (26)$$

The relative contribution of dilution was calculated $RC_{dilution}\,(\%) = \Delta PN_{dilution}/\Delta PN_{all} \times 100$, whereas the relative contribution $RC_{proc}\,(\%)$ of aerosol processes was defined as:

$$RC_{proc} = \left(\Delta PN_{all} - \Delta PN_{proc}\right)/\Delta PN_{all} \times 100. \qquad (27)$$

**Table 8.** Importance of dilution and aerosol processes in the "Urban Case" scenario: relative changes of total PN concentrations between point A and point D (ΔPN) and relative contribution (RC) of dilution and aerosol processes.

| Process | MAFOR | | AEROFOR | | SALSA | |
|---|---|---|---|---|---|---|
| | ΔPN (%) | RC (%) | ΔPN (%) | RC (%) | ΔPN (%) | RC (%) |
| All processes | -70.1 | – | -69.7 | – | -62.8 | – |
| Dilution | -60.6 | 86.5 | -67.0 | 96.1 | -56.0 | 89.3 |
| Coagulation | -69.9 | 0.23 | -69.5 | 0.37 | -62.3 | 0.77 |
| Cond./evaporation | -70.1 | -0.01 | -69.9 | -0.30 | -62.5 | -0.02 |
| Dry deposition | -60.9 | 13.1 | -67.3 | 3.42 | -56.8 | 9.45 |

Table 8 summarizes the results of the process evaluation. Dilution dominated the change of total PN between street level and neighbourhood scale in the model runs, with a relative contribution in the range 86–96 %. Although the same dilution function has been implemented in the models, PN change in simulations with AEROFOR was more strongly controlled by dilution than in simulations with the other models. In all aerosol dynamics models, dry deposition was the most important aerosol process, while coagulation played a minor role. Dry deposition caused a reduction of total PN concentration ($\Delta PN_{all}$ - $\Delta PN_{deposition}$) by 9 %, 3 %, and 6 %, respectively, in model runs with MAFOR, AEROFOR, and SALSA. Differences in the relative contribution of deposition in the models are most probably due to the fact that different schemes for dry deposition were employed (Table 5). Dry deposition onto the road surface and/or building walls in a street canyon is mainly influenced by traffic movement, and can reduce total PN concentrations by about 10–20 % (Gidhagen et al., 2004; Kurppa et al., 2019).

LNMOM-DC was employed to estimate the relevance of coagulation in the "Urban Case", by modelling the coagulation-dispersion system with identical setup. The change in the total PN due to coagulation at 100 m downwind distance was estimated to be less than 2 %. Due to the small impact of coagulation, LNMOM-DC could not be utilised further to calculate the change in the size distribution parameters due to coagulation.

Condensation/evaporation contributed almost negligible to PN changes, but effectively increased total PN (negative RC value; Table 8). Under inefficient dispersion conditions, increase of total PN due to condensation has been noted previously by Karl et al. (2016), in a study of aerosol processes on the neighbourhood scale. While condensation of vapours is not expected to change the total number concentrations, it serves to increase the volume of particles (Seinfeld and Pandis, 2006) and can modify the shape of particle size distributions. The increase of total PN is related to the competition between condensation and dry deposition or coagulation: small particles that grow by condensation, as the air parcel moves away from the emission source, will be less affected by removal through deposition or coagulation.

The results on the importance of aerosol processes from the three models in this study agrees with the general notion that dilution dominates over other processes, and that dry deposition onto the road surface is the only competitive aerosol process that alters total PN concentrations and size distributions related to vehicular traffic emissions in a street canyon (Kumar et al., 2011).

One method of determining the relative importance of various processes is time scale analysis (Ketzel and Berkowicz, 2004). Time scale analysis for a street canyon in Cambridge, UK, showed that time scales were of the order of 40 s for dilution, 30–130 s for dry deposition on the road surface, and 600–2600 s for the dry deposition on the street walls, about 105 s for coagulation, and about 104–105 s for condensation, respectively (Kumar et al., 2008). The time scale analysis by Nikolova et al. (2014) based on results from CFD modelling for an urban street canyon in Antwerp, Belgium, showed that the time scale for coagulation was about 3 times longer than for dilution, while the time scale for dry deposition was close to that of dilution under low wind speed conditions.

The importance of coagulation in street canyons is subject to ongoing controversy. The relevance of coagulation may depend on a variety of different factors, such as exhaust emissions, the meteorological conditions, canyon geometry and complexity of the area (Kumar et al., 2011). The time scales for self-coagulation and inter-modal coagulation of nucleation mode particles is typically longer than the time scales for dilution (Kerminen et al., 2007; Pohjola et al., 2007). Kerminen et al. (1997) concluded that under conditions characterized by exceptionally slow mixing, simultaneous processing of ultrafine particles by dilution, self- and inter-modal coagulation, as well as by condensation/evaporation can occur. Karl et al. (2016) found that coagulation was relevant for street environments in situations when large numbers of small particles (diameter $< 50$ nm) from vehicle exhaust emissions concurred with a significant PN fraction of larger particles (diameter $> 100$ nm). Kerminen et al. (1997) estimated the time scale for inter-modal coagulation of particles with $D_p = 10$ nm to be 900–1200 s during rush hours, short enough to allow moderate removal of nucleation mode particles by inter-modal coagulation.

#### 4.1.3 Effect or influence of condensation/evaporation of organics

In the following, the relevance of condensation/evaporation of organic vapours in the "Urban Case" scenario is analysed with the MAFOR model. Condensation and evaporation are potentially important processes in the urban case simulation, because condensable vapours are first emitted from the vehicles, then condensing to primary emitted particles inside the street canyon and eventually re-evaporating from the condensed phase as the air parcel moves away from the street. Condensation and evaporation do not change the total number concentrations but will alter the size distributions and particle volume. According to Kumar et al. (2011), the effect of condensation in street canyons is uncertain especially regarding the sub-10nm particles. Evaporation reduces the volume concentration of particles. Partial evaporation can also increase the rate of coagulation by increasing the diffusion coefficient of the remaining particles (Jacobson et al., 2005).

The uncertainties of condensation/evaporation in the models are partly attributable to the algorithm of the condensation process (e.g. mass accommodation coefficient in Eq. (4)) and partly to the properties of the condensing/evaporating vapours (e.g. volatility of the chosen substances, vapour pressures of the liquid). In addition, the emission of semi-volatile organic vapours by vehicles is highly uncertain. Several sensitivity runs were done with MAFOR to evaluate the effect of uncertain parameters in the condensation of organic vapours. The evaluation of modelled size distributions was done by grouping particle sizes into 6 size categories (size classes S1–S6; see Karl et al., 2016).

Sensitivity runs with MAFOR were:
**Table 9.** Effect of the chosen parameters for the condensing organic vapour(s) in the MAFOR model when simulating the "Urban Case" scenario (all processes included). Reference is the model run with all processes presented in Sect. 4.1.1. The size ranges of the six size classes are S1: 1–10 nm, S2: 10–25 nm, S3: 25–50 nm, S4: 50–75 nm, S5: 75–100 nm, S6: > 100 nm.

| Parameter | Change of number concentration ($\Delta$PN in %) | | | | | | Change in diameter ($\Delta D_p$ in %) | | | | | |
|---|---|---|---|---|---|---|---|---|---|---|---|---|
| | S1 | S2 | S3 | S4 | S5 | S6 | S1 | S2 | S3 | S4 | S5 | S6 |
| Reference | -76.2 | -73.1 | -70.9 | -59.1 | -57.9 | -54.8 | 0.9 | 1.9 | 1.9 | 9.5 | 3.4 | 3.8 |
| $C^0$(SVOC) $\times$ 100 | -76.1 | -73.1 | -70.9 | -59.2 | -58.0 | -54.8 | 1.4 | 2.3 | 2.3 | 9.7 | 3.5 | 3.8 |
| Adipic acid | -76.3 | -73.1 | -70.9 | -59.0 | -58.0 | -54.8 | 1.1 | 2.0 | 2.0 | 9.5 | 3.4 | 3.8 |
| $\alpha$ = 0.1 | -76.0 | -73.1 | -70.9 | -59.2 | -58.0 | -54.8 | 8.4 | 2.2 | 2.2 | 9.7 | 3.5 | 3.8 |
| $EF_{SVOC} \times 20$ | -77.0 | -73.2 | -70.8 | -59.0 | -57.7 | -54.8 | 5.7 | 1.3 | 1.3 | 8.9 | 2.8 | 3.7 |
| $EF_{SVOC} \times 50$ | -82.0 | -73.6 | -70.1 | -58.6 | -57.0 | -54.7 | 15.7 | 0.1 | 0.1 | 7.0 | 0.9 | 3.2 |

1. $C^0$(SVOC) $\times$ 100
2. Replace SVOC by adipic acid ($C^0 = 0.95$)
3. Accommodation coefficient for organics: 0.1
4. $EF_{SVOC} \times 20$
5. $EF_{SVOC} \times 50$

The model run with all processes presented in the previous sections is used as reference. Results are shown in Table 9. The sensitivity tests reveal that uncertainties associated with the properties of the organic vapour(s) affect only the sizes of particles that are smaller than 10 nm, and these do not limit the ability to simulate most of the number size distribution and total PN concentration. Even a 20-fold increase of SVOC emissions only affects the sub-10 nm particles. A 50-fold increase of SVOC emissions results in a clear growth of < 25 nm particles, mainly to sizes of 75–100 nm. The chemical composition of the traffic exhaust aerosol at points A and D computed with MAFOR indicates that condensation of organic vapours in the high emission case leads to uniform mass increases in the size range 20–200 nm compared to the reference (Fig. 14).

Modelled and measured mass size distribution of total particles at different distances from the edge of the road the reference run and the sensitivity runs is presented in Appendix G and Fig. G1. The highest emission rate of SVOC clearly leads to an overestimation of the measured mass concentration in the size range below 100 nm diameter. The simulations with MAFOR therefore allow to estimate the magnitude of vehicle-emitted organic vapours to be on the order of $10^{-7}$ to $10^{-6}$ g m$^{-1}$ veh$^{-1}$.

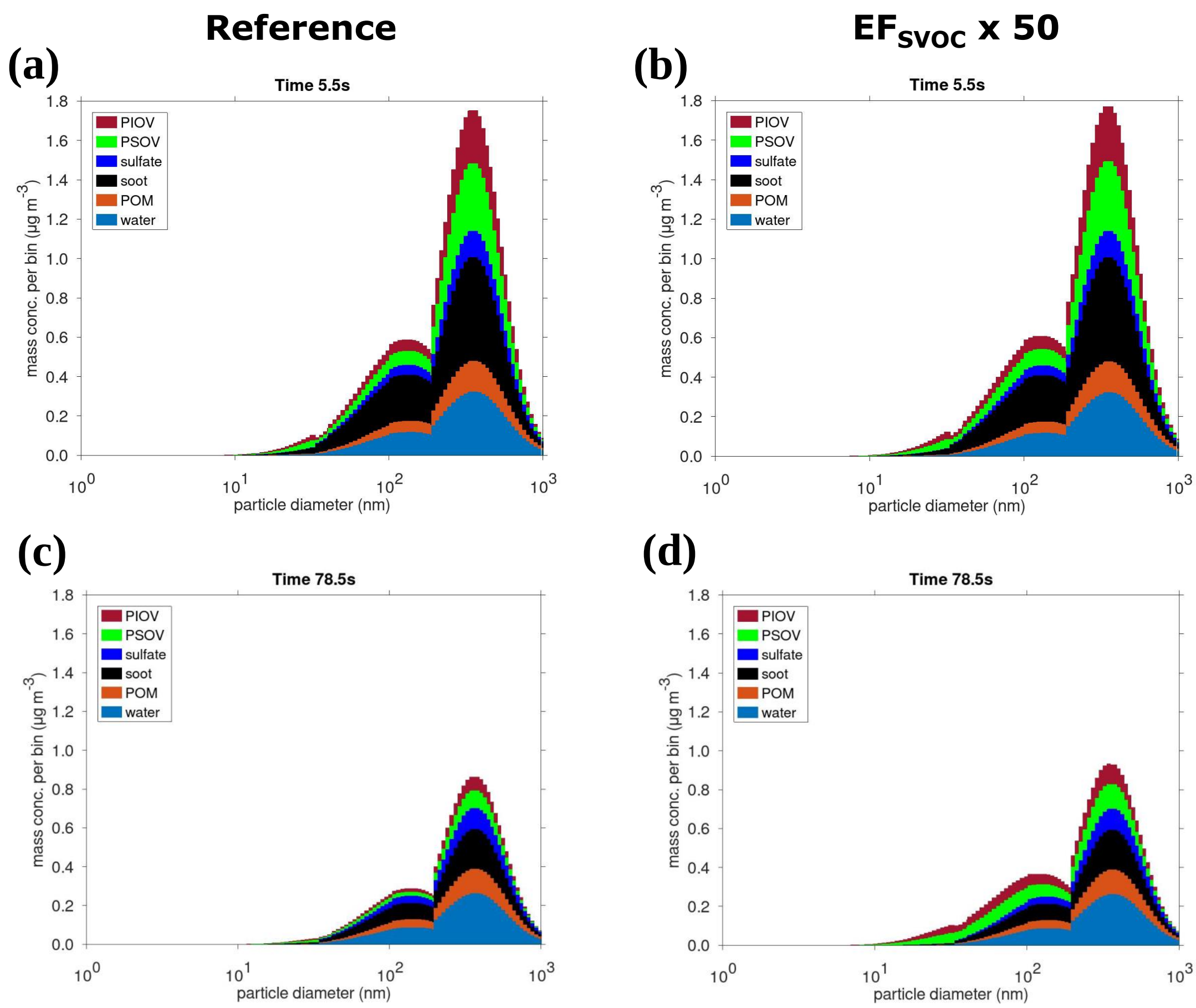

**Figure 14.** Aerosol chemical composition obtained from the MAFOR model, given as mass concentration per size bin, in the "Urban Case" scenario: a) reference simulation at point A; b) simulation $EF_{SVOC} \times 50$ at point A; c) reference simulation at point D; and d) simulation $EF_{SVOC} \times 50$ at point D.

## 4.2 Discussion

### 4.2.1 Uncertainties in the Urban Case scenario

Computation of the aerosol evolution within the street canyon environment of the "Urban Case" scenario involves several assumptions and uncertain parameters. In the following the uncertainties of the processes and the design of the street canyon scenario are discussed.

Dry deposition is identified as the most important aerosol process in the "Urban Case", at the same time the size dependence of the dry deposition velocity is very uncertain. Measurements of dry deposition velocities for one particular surface type generally vary by 1 order of magnitude for a given particle size range of a half logarithmic decade (Petroff et al., 2008). The

HS2012 scheme used in the model is representative for dry deposition to rough environmental surfaces, that results in higher deposition velocities than for the other two aerosol dynamics models. The relative contribution on average of the three models was 9.7 %; together with an uncertainty of ± 60 % (Karl et al., 2016), the RC of dry deposition could be as high as 15 %. The Zhang et al. (2001) parameterization used in SALSA predicts a size-dependent deposition velocity with a minimum at particle diameters of ∼1 µm, however measurements over vegetated surfaces suggest that the deposition velocity minimum occurs closer to ~0.1 µm, at the lower bound of the accumulation mode (Emerson et al., 2020).

Brownian coagulation was identified as a minor aerosol process. While the time scales for coagulation of nucleation mode particles is typically longer than the time scales for dilution, the effect of fractal geometry may enhance the coagulation rates. For small particles, fractal geometry enhances the coagulation kernel with increasing size of the colliding particle compared to spherical shape. A preliminary test of fractal geometry ($r_s = 13.5$ nm and $D_f = 1.7$) in a model run for the Urban Case (all processes included) resulted in PN reduction that was 0.2 % higher compared to compact particles. This suggests a higher importance of coagulation, but does not change the conclusion that coagulation is a minor aerosol process in the Urban Case.

Evaporation might play a role in removing small particles and shrinking larger particles (Harrison et al., 2016), but the low temperature applied in the Urban Case scenario favoured condensation over evaporation. Uncertainties associated with the properties of the organic vapour(s) may affect the sizes of sub-10nm particles. In particular, using a lower mass accommodation coefficient ($\alpha = 0.1$) for the organic vapour(s) may suppress condensation on small particles (Fig. G1), since more vapour molecules reflect from the particle surface back to air. However, molecular dynamics simulations and measurements indicate that the accommodation coefficient of atmospherically relevant organics is consistent with $\alpha = 1$ (nearly perfect accommodation), regardless of the molecular structural properties (Julin et al., 2014).

Traffic-originated particles in the diameter range of 1.3–3.0 nm, so-called nanocluster aerosol (NCA), have been measured in different traffic environments (Rönkkö et al., 2017). Hietikko et al. (2018) reported a clear connection between NCA concentrations and traffic volume in a street canyon. In the M2 street canyon, no significant number concentrations of particles with a diameter of less than 4 nm have been observed. The measurement techniques of the used instruments, i.e. nano-SMPS and ELPI, are not suitable for detection of these small particles. The formation mechanism of NCA particles is not fully understood. It has been hypothesized that depending on the after-treatment systems of vehicles the NCA are non-volatile nano-sized particles formed in the combustion process in the cylinder or exhaust manifold or formed by atmospheric nucleation mechanism during the dilution process of the exhaust (Järvinen et al., 2019; Alanen et al., 2015). The model is not able to simulate solid particles that form in the early stage of the engine exhaust. Neither did the sulphuric acid driven (atmospheric) nucleation produce these small particles (Sect. 3.3). Currently, the relative contribution of traffic-emitted NCA versus atmospheric nucleation to the formation of clusters/particles in this size range is not known and very likely depends on the driving conditions and environmental factors. Based on model calculations, condensational growth of NCA to larger sizes is more important than their removal by coagulation on the street scale (Kangasniemi et al., 2019).

In the coupled dilution–aerosol process modelling of the present study, an average line source is assumed, so that high particle emissions from certain vehicles (e.g. trucks or buses) are not considered. Gidhagen et al. (2004), using a CFD model for a street canyon, find a relative high influence of coagulation on the removal of particles inside a street canyon. For a wind

Open Access EGU

speed of 2 m s$^{-1}$, the effect of coagulation on total PN was 15 % at the leeward side and 21 % at the windward side. Reason for the higher influence of coagulation might be the more realistic simulation of dispersion in the street canyon, resulting in longer residence time of particles inside the street canyon. The CFD simulation considered the plumes of all vehicles inside the street canyon (diluted with clean air), which enhances the effect of removal by coagulation because coagulation is more efficient close to the particle source. The average dilution time scale in the Urban Case (from road edge to point D) was 31 s, close to the dilution time scale of a real street canyon at wind speed of 3 m s$^{-1}$ (Nikolova et al., 2014). For low wind speeds and low traffic intensity the dilution time scale in a street canyon with unit aspect ratio is typically 120 s (Ketzel and Berkowicz, 2004). With a longer residence time in the street canyon, processing of ultrafine particles by coagulation and condensational growth would be more relevant.

Based on the national calculation system for traffic exhaust emissions and energy consumption in Finland (LIPASTO, 2021), the average exhaust emission of $PM_{2.5}$ by vehicles in 2010 was on average 1.5–2.9 times higher than that in 2017 (the reference year of $EF_{PN}$ used in the present study). The decreasing trend is qualitatively in agreement with the corresponding data in figure 6 in Kukkonen et al. (2018); however, that figure only addresses developments until 2014. Ultrafine particles are originating from exhaust emissions, so those have probably diminished in time, mainly due to the implementation of diesel particulate filters. How much exactly is not known; as this depends on the development of engine technology, fuels, and other factors. Model simulations of the Urban Case show that the $EF_{PN}$ from 2017 is in accordance with the total PN concentrations measured in the street canyon.

#### 4.2.2 Discussion of model performance

Statistical performance indicators in the comparison of model data against observation data in the "Urban Case" scenario provide an unambiguous criterion for evaluating the performance of MAFOR in comparison to that of other models. The results on the statistical performance of the model with respect to total PN, number size distribution, $PM_1$ and BC are summarized here:

1. The model reproduced the reduction of total PN concentrations with increasing distance from the street in excellent agreement with the experimental data;

2. The model performs well for the number size distributions at street level and different distances from the street despite the coarser resolution of the particle emission size spectrum from vehicles;

3. The model performed weaker for $PM_1$, however the mean error of the prediction is still acceptable given the high relative uncertainties of the measurements. The low predictability of the observed $PM_1$ variation is partly attributed to the long averaging interval of the measurements (ca. 5–10 min) compared to the instantaneous model simulation;

4. The model performs fairly well for BC, however varying traffic conditions may have affected the measurements, making the M-O comparison for BC less reliable.
Overall, the simulation of the Urban Case demonstrates the good performance of MAFOR v.2 in predicting particle number, size distribution and chemical composition of traffic exhaust aerosol. A strength of the model is the consistent solution of particle number, mass concentration and composition distributions as function of time. An added value of the model is that it can be used to determine the (order or magnitude) emission rate of SVOC by comparison between the modelled and the observed size distribution of total mass.

In addition to the statistical model performance of the aerosol process models presented in Sect. 4.1.1, here we define a set of additional criteria for the overall evaluation. Clearly, this is not a strength-and-weakness analysis because a model user feedback cannot be provided at the current stage. The additional indicators are intended to characterize the capabilities of the models in an objective way and comparable between the models. The selected additional criteria are:

1. Computing time
2. Comprehensiveness of model outputs
3. Representation of aerosol chemical composition

Computing time is an important criteria for comparing the computational efficiency of models and algorithms. Computer models that have an excessive demand of time are less attractive for the model user and are usually not suitable for integration in 3-D models. The computational time on a single CPU for the base simulation of the Urban Case scenario (all processes included) for a plume travel distance of 120 m was 1.5 min for MAFOR (Linux mini PC, 7.6 GB RAM), 1.2 min for SALSA (Linux desktop PC, 32 GB RAM), and 5.2 min for AEROFOR (desktop PC, Windows XP, 2.96 GB RAM, year 2002). Since the different aerosol dynamics models were run on different computers it is not possible to give an accurate ranking of the time required by each model. Nevertheless, roughly comparing the computational times of the models indicates that MAFOR is running with similar speed as SALSA.

Particle number size distribution is the basic output of all models. Additionally, model output of MAFOR comprises size distributions of total mass and the chemical composition (mass fractions). SALSA outputs volume size distributions of particle components, which at known density can be translated to mass concentration. An added value of MAFOR is the capability to resolve the chemical composition of each size section in terms of mass, which allows the size-resolved quantification of the condensed mass of volatile species within the full diameter range.

Regarding the speciation of the aerosol chemical composition in the models, MAFOR has similar degree of detail and capabilities as SALSA, with the addition that two organic vapours (optionally three) of different volatility were used to represent condensation/evaporation of SVOC. AEROFOR used two condensable vapours ($H_2SO_4$ and SVOC) to describe the condensation/evaporation to an internally mixed aerosol, where all particles contained both compounds. In MAFOR and SALSA, the composition of the background aerosol (sulfate, BC, mineral dust, sea-salt, etc.) can be defined separately from the composition of exhaust emissions.

### 4.3 Planned developments for MAFOR

The future development of MAFOR beyond version 2.0 in view of application in urban settings is briefly outlined in the following. Specifically, the further improvement for application of the model in plume dispersion scenarios and the integration in 3-D atmospheric models on the urban scale will be the focus of the planned development for the next versions of MAFOR.

#### 4.3.1 Plume dispersion simulation

The processes relevant for simulating urban cases and the emissions from mobile transport sources are in the focus of the upcoming development. The following topics will be addressed in the continued development of the model:

- Currently, the size spectra of particle emissions can only be represented in four modes. Improving the size resolution of particle emissions (bin-wise) in the model has a high priority.
- Traffic-originated NCA particles may be formed via a delayed primary emission route by rapid nucleation of low-volatility vapours (e.g. primary emitted $H_2SO_4$) during exhaust cooling after release from the tailpipe (Olin et al., 2020) or they are directly emitted as solid particles (Alanen et al., 2015). While the emission of nano-sized solid particles is already implemented (Karl et al., 2013), it is envisaged to implement the delayed primary route in the model to test this hypothesis.
- Ammonia emissions from road traffic is an emerging issue (Farren et al., 2020); $NH_3$ is released from catalyst-equipped gasoline vehicles and light-duty/heavy-duty diesel vehicles that rely on selective catalytic reduction (SCR). Vehicle emissions of $NH_3$ may affect new particle formation via the ternary route and secondary aerosol formation in urban areas. It is planned to activate the PNG-MOSAIC module in plume dispersion runs in order to simulate SIA formation in vehicle exhaust plumes.
- Soot particles acquire a large mass fraction of sulphuric acid during atmospheric ageing. Condensation of $H_2SO_4$ to soot particles was shown to occur at similar rate for a given mobility size, regardless of their morphology (Zhang et al., 2008). Coating of fractal soot agglomerates with $H_2SO_4$ and water is accompanied by restructuring to a more compact form. The change of fractal dimension and effective density during soot ageing will be implemented in the model.
- Additional dilution schemes for ship exhaust for ocean-cruising vessels may be implemented. Chosson et al. (2008) proposed a dilution parameterization for use in CTM based on sophisticated methods to represent dilution in boundary layers by taking into account the initial buoyancy flux of the ship exhaust. For close-to-stack dispersion, the current method in Karl et al. (2020) is considered to be more suitable (Sect. 2.7).
- Particles from ship exhaust can act as cloud condensation nuclei (CCN). Aerosol activation will be implemented in the model based on the scheme of Abdul-Razzak and Ghan (2002) with a sectional representation. Instead of using a single-parameter representation for hygroscopicity growth (Petters and Kreidenweis, 2007), the dynamically calculated concentrations in the liquid droplet will be used.

Open Access EGU

With the proposed implementations, it is assured that the model remains state-of-the-art and could even become a benchmark model for aerosol dynamics process simulations.

#### 4.3.2 Integration in 3-D atmospheric models

Implementation of the presented aerosol dynamics module into 3-D atmospheric dispersion models is facilitated by the operator-splitting of processes and by the efficient integration of particle number and mass concentrations. The fixed sectional method is the most practical way to consider continuous nucleation of new particles together with the atmospheric transport and emission of particles. Coagulation is the process with the highest computational demand due to the representation of collisions of a particle from one size section with particles from all other sections. It will be considered in the future to implement an adaptive time stepping scheme for solving the coagulation process.

With regard to implementation of the aerosol dynamics code into large scale atmospheric models it is of special interest to assess how much one can lower the accuracy of the size distribution description without compromising on the accuracy of the model results. Overall, the size distribution representation using 60 size sections can be recommended for the simulation of long time periods, as the accuracy in terms of size distribution changes and total number concentration is satisfactory while the computational demand is only 10 % higher compared to the lowest tested resolution, as shown in Supplement Sect. S2. Aerosol representations in large scale models are typically limited to less than 20 size classes, as the particles in each size section have to be included in the advection routine and a higher number of advected species increases the computing time. Therefore methods need to be developed for the mapping of the size representation used in the aerosol dynamics code and the advected particle species. The effect of changing the number of size classes in the 3-D model needs to be tested thoroughly.

Studies have demonstrated the relevance of episodes of new particle formation in cities situated in high insolation regions such as southern Europe. Both photo-induced nucleation and traffic emissions play a critical role in determining UFP concentrations in cities (Rivas et al., 2020). In addition, there is the highly dynamic sequence of chemical and physical processes such as condensation, deposition, and coagulation that modulates the number size distributions, making modelling of UFP concentrations on city scale a complex task.

It is planned to integrate the aerosol dynamics code into the open source city-scale model EPISODE-CityChem (Karl et al., 2019). The first requirement is the implementation of a size-resolved particle number emission inventory that compiles PN emission factors and size distributions for different sectors (e.g., Paasonen et al., 2016). The basic assumption of these PN emission inventories is that all primary particles are non-volatile and composed of the same material, although one could assume a certain fraction of particles (in each size section) to be either BC, OC or a different material. According to this definition, volatile particles would always be secondary particles, i.e. forming in the photo-induced nucleation or by condensation of gases already existing in the atmosphere, ignoring that volatile particles may also form rapidly very close to the source of emissions, on the sub-grid scale of the 3-D model (grid resolution typically 100–1000 m). Nevertheless, the division into primary non-volatile particles and secondary volatile particles serves as a good starting point for the implementation of aerosol dynamics in the city-scale model.

There are certain specifications of the MAFOR box model that need to be retained in the large scale model: (1) the structure of four aerosol modes (nucleation, Aitken, Accumulation, coarse) where each mode is divided into the same number of size sections; and (2) the consistency between number and mass calculations. Condensation/evaporation of a chemical species in MAFOR adheres to the mass balance between gas phase and particle phase. Therefore, the mass concentration of the condensing species in each size section has to be considered as additional model species. When for instance 16 tracers for PN (16 size classes) are used, then the condensation of a single gas species will require the addition of 16 tracers for mass concentration. For computational reasons, one should aim to restrict the variety of chemical aerosol components as much as possible, for example by lumping all components of primary emitted particles (BC, primary OC, sea-salt, etc.) into one single non-volatile model species consistent with the PN emission approach outlined above.

The treatment of secondary organic aerosol by a hybrid approach in MAFOR (Sect. 2.5) is already in line with possible implementation in 3-D models. For the implementation in an atmospheric model it is important to connect the vapours to their origin and source region, e.g. biogenic versus anthropogenic, for later research applications. The chemistry solver of the 3-D model needs to be modified to account for chemical reactions that lead to the production of gaseous precursors, or a subset of these, involved in SOA formation (Fig. 7).

## 5 Summary and conclusions

The open source aerosol dynamics model MAFOR v2.0, as a new community model, was described and evaluated against measured data, and the predictions were inter-compared with those of two other aerosol process models.

The main new features of MAFOR v2.0, compared to the original model version (v.1) are the following. (1) The model has been coupled with the chemistry module MECCA, comprising detailed up-to-date photolysis rates of VOC chemistry. This allows the partitioning of chemical species and the subsequent aqueous phase reactions in the liquid phase of coarse mode particles. (2) The model includes a revised Brownian coagulation kernel that takes into account the fractal geometry of soot particles, van der Waals forces and viscous interactions. (3) The model contains a multitude of state-of-the-art nucleation parameterizations that can be selected by the model user. (4) The model has been coupled with PNG-MOSAIC, enabling size-resolved partitioning of semi-volatile inorganics at a relatively long time interval. (5) The model includes a hybrid method for the formation of SOA within the framework of condensation/evaporation. These features make the model well suited for studying changes of the emitted particle size distributions by dry deposition, coagulation, and by condensation/evaporation of organic vapours in urban environments and also for the simulation of new particle formation over multiple days.

A main advantage of MAFOR v2.0 compared to other sectional aerosol process models is the consistent treatment of the mass concentration of particulate compounds and particle numbers, ensuring that number and mass changes occur accordingly over time. Several aspects of the numerical solutions (efficient integration of number and mass concentrations, operator-splitting of processes, use of the fixed sectional method) make the aerosol dynamics code a promising candidate for implementation into large scale atmospheric models.

The performance of MAFOR v2.0 was evaluated against field-scale measurements of plume dispersion in a street environment located in the centre of Helsinki, published by Pirjola et al. (2012). The experimental data was obtained with a mobile laboratory van at different locations in the street environment. The data included particle number measurements in the size range of 3–414 nm, black carbon, and fine particulate mass $PM_1$. The model was also inter-compared with the results from two other aerosol dynamic models (AEROFOR and SALSA). MAFOR reproduced the reduction of total number concentrations with increasing distance from the street in good agreement (IOA = 0.85) with observations. MAFOR performed well in predicting the number size distributions at street level and at different distances from the street (average IOA = 0.78) and was able to reproduce the development of the size distributions with increasing distance better than AEROFOR and SALSA. A limitation of MAFOR is that it represents the particle emission size spectrum as a multi-modal distribution, which may result in an underestimation of the number of small particles, while the total number of emitted particles is not affected. MAFOR predicted the variation of fine particulate matter, $PM_1$ (IOA = 0.25) in the street environment in better agreement with experimental data than SALSA. The difficulty in predicting the variation of observed $PM_1$ is related to the long averaging interval of the mass measurements compared to the model simulations that reflect instantaneous concentrations.

Dry deposition was found to be the only aerosol process that can compete with dilution, in agreement with several previous aerosol process studies in street canyons. Brownian coagulation played a minor role and this was also confirmed by a simulation with the dispersion-coagulation code LNMOM-DC. Longer residence time in the street canyon and higher-than-average emissions from certain vehicles may increase the relevance of self- and inter-modal coagulation of nucleation mode particles. For future aerosol process modelling studies in urban environments it is recommended to (1) select an appropriate deposition scheme based on the environmental conditions; (2) parameterize the dilution rate based on turbulence-resolving CFD simulations; and (3) constrain the particle emission size spectrum by independent measurements in the same environment.

The early phase of the vehicle exhaust plume was not resolved in this study. The vehicle wake is the first spatial scale from where the emitted UFP will disperse into ambient environment (e.g., Kumar et al., 2011). The parcel of exhaust emission at tailpipe contains pre-existing particles from fuel combustion, unburnt droplets from lubricant oil and various precursor gases for condensation. This parcel may already contain traffic-originated particles in the diameter range of 1.3–3.0 nm, so-called nanocluster aerosol (NCA) particles that were previously not detected by the instruments due their small size. Their origin might be either the direct emission of non-volatile particles that formed in the engine or the rapid nucleation of low-volatility vapours during exhaust cooling after tailpipe. The delayed primary emission route to explain the formation of NCA during exhaust cooling should be implemented in MAFOR in the future. The subsequent growth of NCA by organic vapours also needs to be investigated; MAFOR could be an ideal research tool for this, as the model allows to constrain the emission rate of condensable organic vapours based on the measured mass size distribution.

For the consideration of the aerosol processes in urban scale 3-D models, a division into primary non-volatile particles and secondary volatile particles is proposed here as a starting point for the implementation of the aerosol dynamics code. The treatment of primary particles as non-volatile is consistent with current size-resolved PN emission inventories. The volatile particles form by nucleation and both particle types grow by condensation of semi-volatile or low-volatile vapours. The division

enables the mass-conserving approach to condensation/evaporation of vapours, and allows to minimize the total number of aerosol chemical species in the 3-D model.

The continued development of the open source code by the community is guided by a consortium of aerosol scientists. Ultimately, it is intended to establish MAFOR v2.0 as a state-of-the-art benchmark model for evaluating aerosol processes in dispersion studies from local to regional and global scales. We encourage and support the integration of this aerosol dynamics code into urban, regional and global scale atmospheric chemistry transport models, possibly also into earth system models.

*Code and data availability.* The code of the MAFOR v2.0 community model and the relevant data are available at https://doi.org/10.5281/zenodo.5718580. This Zenodo repository contains the source code of the MAFOR model (v1.9.9) as archived snapshot that was used in this study (including the external libraries for MOSAIC and MECCA); the user manual of the model; and the post-processing scripts applied in the data analysis and model evaluation described in this paper. The model code, documentation and the input data are published under the GPL v3.0 license. The experimental data measured by the mobile laboratory Sniffer used in this paper can be downloaded from https://doi.org/10.5281/zenodo.5718580. The project repository for the development of the open source MAFOR v2.0 community model is at https://github.com/mafor2/mafor.

A Docker image with the pre-installed MAFOR model (v1.9.9) based on the official Ubuntu Docker image is available at https://hub.docker.com/repository/docker/matthkarl/ubuntu-mafor2. The image (ca. 1.9 GB) includes following libraries and programs: gcc, gfortran-9, gawk, flex, nano, git, graphviz, and octave.

MOSAIC code is accessible within the chemistry version of the Weather Research and Forecasting model (WRF-Chem), which is publicly available. Model users are required to cite Zaveri et al. (2008) in publications resulting from application of MAFOR when the MOSAIC module was activated.

The model code of AEROFOR can be made available upon request to Liisa Pirjola (liisa.pirjola@helsinki.fi) as a private copy. SALSA as stand-alone model is freely available as an open source code under the Apache License 2.0. The code is available at https://github.com/UCLALES-SALSA/SALSA-standalone. For simulating the urban case with SALSA, a simple driver was written that models the dispersion and emission of aerosol particles and gases and then call SALSA functions for aerosol processes. The driver can be found at https://doi.org/10.5281/zenodo.5718580.

## Appendix A: List of acronyms and nomenclature

A list of the acronyms and abbreviations used in this work is given in Table A1. The nomenclature used in this work is summarized in Table A2.

**Table A1.** List of the acronyms and abbreviations used in this work.

| Acronym | Description |
|---|---|
| Acc | Accumulation mode |
| ACDC | Atmospheric Cluster Dynamics Code |
| ADCHEM | The trajectory model for Aerosol Dynamics, gas and particle phase CHEMistry and radiative transfer |
| AEROFOR | Sectional aerosol dynamics model |
| AELV | Surrogate species for aromatic extremely low volatility organics |
| AIM | Aerosol Inorganics Models |
| Ait | Aitken mode |
| ALOV | Surrogate species for aromatic low volatility organics |
| APC | Analytical Predictor of Condensation |
| AQS | Air quality monitoring station |
| ASOV | Surrogate species for aromatic semi-volatile organics |
| BC | Black carbon |
| BELV | Surrogate species for biogenic extremely low volatility organics |
| BHN | Binary homogeneous nucleation |
| BLOV | Surrogate species for biogenic low volatility organics |
| BSOV | Surrogate species for biogenic semi-volatile organics |
| BVOC | Biogenic volatile organic compound |
| CAABA | Chemistry As A Boxmodel Application |
| CCN | Cloud condensation nuclei |
| CFD | Computational Fluid Dynamics |
| CLOUD | Cosmics Leaving OUtdoor Droplets |
| Coa | Coarse mode |
| COE | Coefficient of efficiency |
| CPU | Central Processing Unit |
| CTM | Chemistry-transport-model |
| ELPI | Electrical Low Pressure Impactor |
| ELVOC | Extremely low volatility organic compound |
| EMEP | European Monitoring and Evaluation Program |
| EQUISOLV II | Equilibrium Solver, updated code |
| FORTRAN | Formula Translation/Translator (high-level programming language) |
| GPL | General Public License |
| HOM | Highly oxygenated molecules |
| HSY | Helsinki Region Environmental Service Authority |
| IMN | Ion-mediated nucleation |
| IOA | Index of agreement |
| ISORROPIA | Thermodynamic equilibrium model for multiphase multicomponent inorganic aerosols |
| JPL | Jet Propulsion Laboratory |
| JVAL | Module calculating photolysis rate constants (J-VALues) |
| KPP | Kinetic pre-processor |

| Acronym | Description |
|---|---|
| LES | Large Eddy Simulation |
| LIPASTO | Calculation system for traffic exhaust emissions and energy use in Finland |
| LNMOM-DC | Log Normal Method Of Moments – Diffusion Coagulation model |
| M7 | Modal aerosol model with seven modes |
| MAE | Mean absolute error |
| MAFOR | Multicomponent Aerosol FORmation model |
| MARS | Model for an Aerosol Reacting System |
| MECCA | Module Efficiently Calculating the Chemistry of the Atmosphere |
| MESA | Multicomponent Equilibrium Solver for Aerosols |
| MMEA | Measurement, Monitoring and Environmental Assessment |
| MOM | Mainz Organic Mechanism |
| MOSAIC | Model for Simulating Aerosol Interactions and Chemistry |
| MPP-FMI | Meteorological preprocessor of the Finnish Meteorological Institute |
| MTEM | Multicomponent Taylor Expansion Method |
| NCA | Nanocluster aerosol |
| Nuc | Nucleation mode |
| ODE | Ordinary differential equations |
| OC | Organic carbon |
| OSPM | Operational Street Pollution Model |
| PartMC | Particle Monte Carlo model |
| PBL | Planetary boundary layer |
| PELV | Surrogate species for primary extremely low volatility organics |
| PIOV | Surrogate species for primary intermediate volatility organics |
| $PM_1$ | Particulate matter with aerodynamic diameter $<1\,\mu m$ |
| $PM_{2.5}$ | Particulate matter with aerodynamic diameter $<2.5\,\mu m$ |
| $PM_{10}$ | Particulate matter with aerodynamic diameter $<10\,\mu m$ |
| PN | Particle number |
| PNC | Particle number concentrations |
| PNG | Predictor of Nonequilibrium Growth |
| POA | Primary organic aerosol |
| PSOV | Surrogate species for primary semi-volatile organics |
| RC | Relative contribution |
| RHS | Right-hand-side |
| SALSA | Sectional Aerosol Module for Large Scale Applications |
| SCR | Selective catalytic reduction |
| SIA | Secondary inorganic aerosol |
| SMPS | Scanning Mobility Particle Sizer |
| SOA | Secondary organic aerosol |
| SSCM | Simplified street canyon model |

| Acronym | Description |
|---|---|
| SST | Sea surface temperature |
| SVC | Saturation vapour concentration |
| SVOC | Sum of semi-volatile organics |
| THN | Ternary homogeneous nucleation |
| TRANSPHORM | Transport related Air Pollution and Health impacts – Integrated Methodologies for Assessing Particulate Matter |
| UFP | Ultrafine particles (aerodynamic diameter < 100 nm) |
| UV/VIS | Ultraviolet/visible |
| VBS | Volatility basis set |
| VOC | Volatile organic compound |

**Table A2.** Nomenclature used in this work.

| Symbol | Description and unit |
|---|---|
| $C_{aq,q}$ | Concentration of compound $q$ in the aqueous phase, $\mu g\, m^{-3}$ |
| $C_{g,q}$ | Concentration of compound $q$ in the gas phase, $\mu g\, m^{-3}$ |
| $C_{eq,q}$ | saturation vapour concentration over a flat solution of the same composition as the particles, $\mu g\, m^{-3}$ |
| $C^0_q$ | Saturation mass concentration of compound $q$, $\mu g\, m^{-3}$ |
| $C^*_q$ | Effective saturation mass concentration of compound $q$, $\mu g\, m^{-3}$ |
| $C_{tot,q}$ | Total concentration of compound $q$ in gas and particles, $\mu g\, m^{-3}$ |
| $c_{m,q}$ | Molecular speed of compound $q$, $m\, s^{-1}$ |
| $c_v$ | Conversion factor to convert kg into µg, $\mu g\, kg^{-1}$ |
| $D_{p,i}$ | Particle diameter of section $i$, m |
| $D_q$ | Molecular diffusion coefficient of compound $q$ in the gas phase, $m^2\, s^{-1}$ |
| $D_{m,i}$ | Particle diffusion coefficient of particles in section $i$, $m^2\, s^{-1}$ |
| $D_R$ | Dilution ratio |
| $f$ | Volume fraction of the intermediate volume of particles |
| $f_{om}$ | Fraction of absorbing organic material in the aerosol |
| $H_{A,q}$ | Dimensionless Henry's law coefficient of compound $q$ |
| $H'_{q,i}$ | Dimensionless effective Henry's law coefficient of compound q for the partitioning to section $i$ |
| $H_{mix}$ | Height of the boundary layer or plume height, m |
| $I_{q,i}$ | Rate of condensation/evaporation of compound $q$ to particles in section $i$, $m^3\, s^{-1}$ |
| $J_{nuc}$ | Nucleation rate, $m^{-3}\, s^{-1}$ |
| $K_{i,j}$ | Coagulation coefficient between particles in section $i$ and $j$, $m^3\, s^{-1}$ |
| $k_{m,q}$ | Mass transfer coefficient for compound $q$ from gas phase to aqueous phase, $s^{-1}$ |
| $k_{T,q,i}$ | Mass transfer coefficient for compound $q$ from gas phase to particles of section $i$, $s^{-1}$ |

| Symbol | Description and unit |
|---|---|
| $Kn$ | Knudsen number |
| $LWC$ | Liquid water content, $m^3\,m^{-3}$ |
| $M_i$ | Total mass of a particle in section $i$, µg |
| $m_{q,i}$ | Mass concentration of compound $q$ in section $i$, $\mu g\,m^{-3}$ |
| $m_{\mathrm{bg},q,i}$ | Mass concentration of compound $q$ in section $i$ in background air, $\mu g\,m^{-3}$ |
| $MW_q$ | Molecular weight of compound $q$, $g\,mol^{-1}$ |
| $N_A$ | Avogadro constant, $mol^{-1}$ |
| $N_i$ | Number concentration of particles in section $i$, $m^{-3}$ |
| $N_{\mathrm{bg},i}$ | Number concentration of particles in section $i$ in background air, $m^{-3}$ |
| $P$ | Precipitation rate, $mm\,h^{-1}$ |
| $Q^i_{m,q}$ | Mass-based emission rate of compound $q$ in particles of section $i$, $\mu g\,m^{-2}\,s^{-1}$ |
| $R$ | Universal gas constant, $kg\,m^2\,s^{-2}\,K^{-1}\,mol^{-1}$ |
| $r_i$ | Radius of particles in section $i$, m |
| $r_{c,i}$ | Collision radius of particles in section $i$, m |
| $r_d$ | Droplet radius, m |
| $S'_{q,i}$ | Equilibrium saturation ratio of compound $q$ over particles of section $i$ |
| $T$ | Air temperature, K |
| $V_{k,j}$ | Intermediate volume for the collision of particles from section $k$ with particles from section $j$, $m^3$ |
| $W_c$ | Correction factor for van der Waals interactions in the continuum regime |
| $W_k$ | Correction factor for van der Waals interactions in the free-molecular regime |
| $\alpha_q$ | Mass accommodation coefficient of compound $q$ on particles |
| $\alpha_{l,q}$ | Mass accommodation coefficient of gas $q$ to the droplet surface |
| $\beta_{q,i}$ | Transitional correction factor for compound $q$ in particles of section $i$ |
| $\gamma_{\mathrm{om},q}$ | Activity coefficient of compound $q$ in the organic mixture |
| $\gamma_{q,i}$ | Molar fraction of compound $q$ in particles of section $i$ |
| $\delta$ | Kronecker delta function |
| $\delta_m$ | Mean distance from the centre of a sphere, m |
| $\lambda_{\mathrm{dil}}$ | Dilution rate, $s^{-1}$ |
| $\lambda^i_{\mathrm{dry}}$ | Dry deposition rate of particles in section $i$, $s^{-1}$ |
| $\lambda^i_{\mathrm{wet}}$ | Wet scavenging rate of particles in section $i$, $s^{-1}$ |
| $\overline{v}_{p,i}$ | Mean thermal speed of a particle in section $i$, $m\,s^{-1}$ |
| $\rho_{\mathrm{eff}}$ | Effective density of fractal particles, $kg\,m^{-3}$ |
| $\rho_{L,q}$ | Density of the pure liquid, $kg\,m^{-3}$ |
| $\rho_{p,i}$ | Average density of particles in section $i$, $kg\,m^{-3}$ |
| $\rho_q$ | Density of compound $q$, $kg\,m^{-3}$ |
| $\sigma_q$ | Surface tension, $kg\,s^{-2}$ |
| $\upsilon_i$ | Volume of particles in section $i$, $m^3$ |
| $\upsilon_{q,i}$ | Volume of compound $q$ in particles of section $i$, $m^3$ |
| $\upsilon_{g,q}$ | Molecular volume of the condensing vapour, $m^3$ |

## Appendix B: Analytical Predictor of Condensation

The Analytical Predictor of Condensation (Jacobson, 2005b) obtains a non-iterative solution for the change of the gas-phase concentration of the condensable compound with time using the mass balance equation of the final aerosol and gas phase concentrations. Based on the mass-balance equation, the total concentration $C_{tot,q}$ of the compound in gas and particles is constrained by:

$$C_{tot,q} = C_{g,q,t} + \sum_{i=1}^{N_B} m_{q,i,t} = C_{g,q,t-\Delta t} + \sum_{i=1}^{N_B} m_{q,i,t-\Delta t}, \tag{B1}$$

where the sub-scripts $t$ and $t-\Delta t$ indicate the current time step and one time step backward.
The final gas-phase concentration in the condensation process at the end of time step $\Delta t$ is:

$$C_{g,q,t} = \frac{C_{g,q,t-\Delta t} + \Delta t \sum_{i=1}^{N_B} \left( k_{T,q,i,t-\Delta t} S'_{q,i,t-\Delta t} C_{eq,q,t-\Delta t} \right)}{1 + \Delta t \sum_{i=1}^{N_B} k_{T,q,i,t-\Delta t}} \tag{B2}$$

The final gas-phase concentration in the dissolution process at the end of time step $\Delta t$ is (Jacobson, 2005a):

$$C_{g,q,t} = \frac{C_{g,q,t-\Delta t} + \Delta t \sum_{i=1}^{N_B} \left( m_{q,i,t-\Delta t} \left[ 1 - \exp\left( - \frac{\Delta t k_{T,q,i,t-\Delta t} S'_{q,i,t-\Delta t}}{H'_{q,i,t-\Delta t}} \right) \right] \right)}{1 + \Delta t \sum_{i=1}^{N_B} \left( \frac{k_{T,q,i,t-\Delta t}}{S'_{q,i,t-\Delta t}} \left[ 1 - \exp\left( - \frac{\Delta t k_{T,q,i,t-\Delta t} S'_{q,i,t-\Delta t}}{H'_{q,i,t-\Delta t}} \right) \right] \right)}, \tag{B3}$$

where $C_{g,q,t-\Delta t}$ is the gas-phase concentration of compound $q$ at the end of the chemistry time step, $k_{T,q,i}$ is the mass transfer rate of the gas to particles, $S'_{q,i}$ is the equilibrium saturation ratio, and $H'_{q,i}$ is the effective Henry's law coefficient. Substituting Eq. (B3) back into Eq. (15a) gives the final total particle concentration in each size bin for the dissolution growth (Sect. 2.4).

The concentration calculated from Eq. (B2) for condensation/evaporation cannot fall below zero, but can increase above the total mass of the compound. Therefore, the gas-phase concentration is limited by $C_{g,q,t} = min(C_{g,q,t}, C_{tot,q})$. The updated gas-phase concentration in Eq. (B2) is then substituted in Eq. (8) to compute the final aerosol component concentration:

$$m_{q,i,t} = m_{q,i,t-\Delta t} + \Delta t k_{T,q,i,t-\Delta t} \left[ C_{g,q,t} - S'_{q,i,t-\Delta t} C_{eq,q,i,t-\Delta t} \right]. \tag{B4}$$

Problematic is that Eq. (B4) can result in negative aerosol mass concentration or in a concentration that exceeds the maximum (i.e. the total compound concentration). Therefore two limits have to be placed subsequently after the computation of Eq. (B4). The first limit is $m_{q,i,t} = max(m_{q,i,t}, 0)$ and the second limit is (Jacobson, 2005b):

$$m_{q,i,t} = \frac{\left[ C_{g,q,t-\Delta t} - C_{g,q,t} + \sum_{i=1}^{N_B} max\left( m_{q,i,t-\Delta t} - m_{q,i,t}, 0 \right) \right]}{\sum_{i=1}^{N_B} \left[ max\left( m_{q,i,t} - m_{q,i,t-\Delta t}, 0 \right) \right]} \cdot \left( m_{q,i,t} - m_{q,i,t-\Delta t} \right), \tag{B5}$$

where the values of $m_{q,k,t}$ on the right side of the equation are determined after the first limit has been applied for all size bins.

The Analytical Predictor of Condensation, with the mass-balance restrictions above, is unconditionally stable, since all final concentrations for gas and particle are bounded between 0 and $C_{tot}$, regardless of the time step.

## Appendix C: Brownian coagulation

In the model size distribution, particles from the first size section collide with particles from all other size sections. Particles from the second size section collide with particles from third to largest size section, and so on. The number concentration of particles in section $i$, $N_i$, increases if the colliding particles result in a particle of the same size as particles in section $i$. It decreases if particles in section $i$ coagulate with particles of other size sections or of the same section. When particles of volume $\upsilon_k$ and $\upsilon_j$ collide, the resulting particle has an intermediate volume $V_{k,j} = \upsilon_k + \upsilon_j$. If the intermediate volume falls between the two size sections $i$ and $i+1$, then the new particle is split between the two sections, constrained by volume conservation. Thus a size-splitting operator, the volume fraction $f_{k,j,i}$ for the partitioning to each model section $i$, is defined as in Jacobson (2005b):

$$f_{k,j,i} = \begin{cases} \left(\dfrac{\upsilon_{i+1} - V_{k,j}}{\upsilon_{i+1} - \upsilon_i}\right)\dfrac{\upsilon_i}{V_{k,j}} & \upsilon_i \leq V_{k,j} < \upsilon_{i+1} \qquad i < N_B \\ 1 - f_{k,j,i-1} & \upsilon_{i-1} < V_{k,j} < \upsilon_i \qquad i > 1 \\ 1 & V_{k,j} \geq \upsilon_i \qquad i = N_B \\ 0 & \text{all other cases} \end{cases} \tag{C1}$$

An advantage of this method is, that the volume fractions obtained in Eq. (C1) are independent of the representation of the size distribution.

## Appendix D: Correction factors for the coagulation rate

The free-molecular collision rate correction factor $W_k$ of the coagulation rate due to van der Waals forces is given by:

$$\begin{aligned} W_{k,i,j} \;=\; & \frac{-1}{2\left(r_i + r_j\right)^2 k_B T} \int_{r_i+r_j}^{\infty} \left(\frac{dE_{i,j}(r)}{dr} + r\frac{d^2E_{i,j}(r)}{dr^2}\right) \\ & \times \exp\left[\frac{-1}{k_B T}\left(\frac{r}{2}\frac{dE_{i,j}(r)}{dr} + E_{i,j}(r)\right)\right] r^2 dr \end{aligned} \tag{D1}$$

In the free molecular limit, there is no viscous interaction between particles.

The van der Waals interaction potential $E(r)$ is given by:

$$E_{i,j}(r) = -\frac{A_H}{6}\left[\frac{2r_ir_j}{r^2 - \left(r_i + r_j\right)^2} + \frac{2r_ir_j}{r^2 - \left(r_i - r_j\right)^2} + \ln\frac{r^2 - \left(r_i + r_j\right)^2}{r^2 - \left(r_i - r_j\right)^2}\right], \tag{D2}$$

where $A_H$ is the Hamaker constant which is specific for the van der Waals properties of each substance. Here, the Hamaker constant of water, $A_H/k_BT = 20$ is used for all particle types. The derivatives of the van der Waals interaction potential in Eq. (D1), $\frac{dE_{i,j}(r)}{dr}$ and $\frac{d^2E_{i,j}(r)}{dr^2}$, are obtained using the MATLAB® symbolic package.

The correction factor $W_c$ in the continuum regime is:

$$W_{c,i,j} = \frac{1}{\left(r_i + r_j\right) \cdot \int\limits_{r_i+r_j}^{\infty} \frac{D_{i,j}^{\infty}}{D_{r,i,j}}(r) \times \exp\left[\frac{E_{i,j}}{k_BT}\right] \frac{dr}{r^2}} \tag{D3}$$

Viscous force correction of the diffusion coefficient in the continuum regime is:

$$\frac{D_{i,j}^{\infty}}{D_{r,i,j}}(r) = 1 + \frac{2.6 r_i r_j}{\left(r_i + r_j\right)^2} \sqrt{\frac{r_i r_j}{\left(r_i + r_j\right)\left(r - r_i - r_j\right)}} + \frac{r_i r_j}{\left(r_i + r_j\right)\left(r - r_i - r_j\right)} \tag{D4}$$

In Eq. (D4), $D_{r,i,j}$ is a relative diffusion coefficient between particles $i$ and $j$, and $D_{i,j}^{\infty} = D_{p,i} + D_{p,j}$ is the sum of the individual diffusion coefficients of the two particles.

The integral in the correction factors $W_k$ and $W_c$ is approximated by numerical integration using Gauss-Legendre quadrature formula after transforming the variable $r$ using the relation $x = b/r$ (with the dimensionless coordinate $x$), so that the limits of the integral become 0 and $1/(1+a/b)$, $\int\limits_0^{1/(1+a/b)} (integrand)\frac{dx}{b}$, can be evaluated as function of $x$ (where $a = r_i$, $b = r_j$; and $b \geq a$).

**Appendix E: Dilution function for the Urban Case**

The first dilution stage of the diluted exhaust plume, between the upwind kerbside and the downwind kerbside of the street, was described with the jet plume dispersion model of Vignati et al. (1999). In this model, dispersion of the plume is calculated taking into account the atmospheric turbulence, the traffic-generated turbulence and the entrainment of fresh air due to the jet effect of the exhaust gas. The expression for the evolution of the plume cross-section during the first dilution stage is given by:

$$S(t) = \left(\sqrt{S_0} + t\sigma_w(0)\right)^2 - (t\alpha u_0)^2, \tag{E1}$$

where $S$ is the cross sectional area of the air parcel or exhaust plume (in $m^2$), $S_0$ is the cross-section of the initial air parcel (here: 0.8 m), $\sigma_w(0)$ is the initial entrainment velocity (in $m\,s^{-1}$) and $u_0$ is the initial exhaust gas velocity (here: 0.23 $m\,s^{-1}$).

The entrainment velocity is given by:

$$\sigma_w^2(t) = (\alpha u_{\text{street}})^2 + \sigma_{wt}^2 + \left(\alpha u_{\text{jet}}\right), \tag{E2}$$

where $u_{\text{street}}$ is the street-level wind speed, $\sigma_{wt}$ is the traffic-generated turbulence, and $u_{\text{jet}}$ is the plume jet velocity. The proportionality constant $\alpha$ is set to 0.1, typical for mechanically induced turbulence (Berkowicz et al., 1997). The traffic-generated turbulence is estimated using the traffic count, street width, horizontal area of a vehicle and the typical vehicle speed.

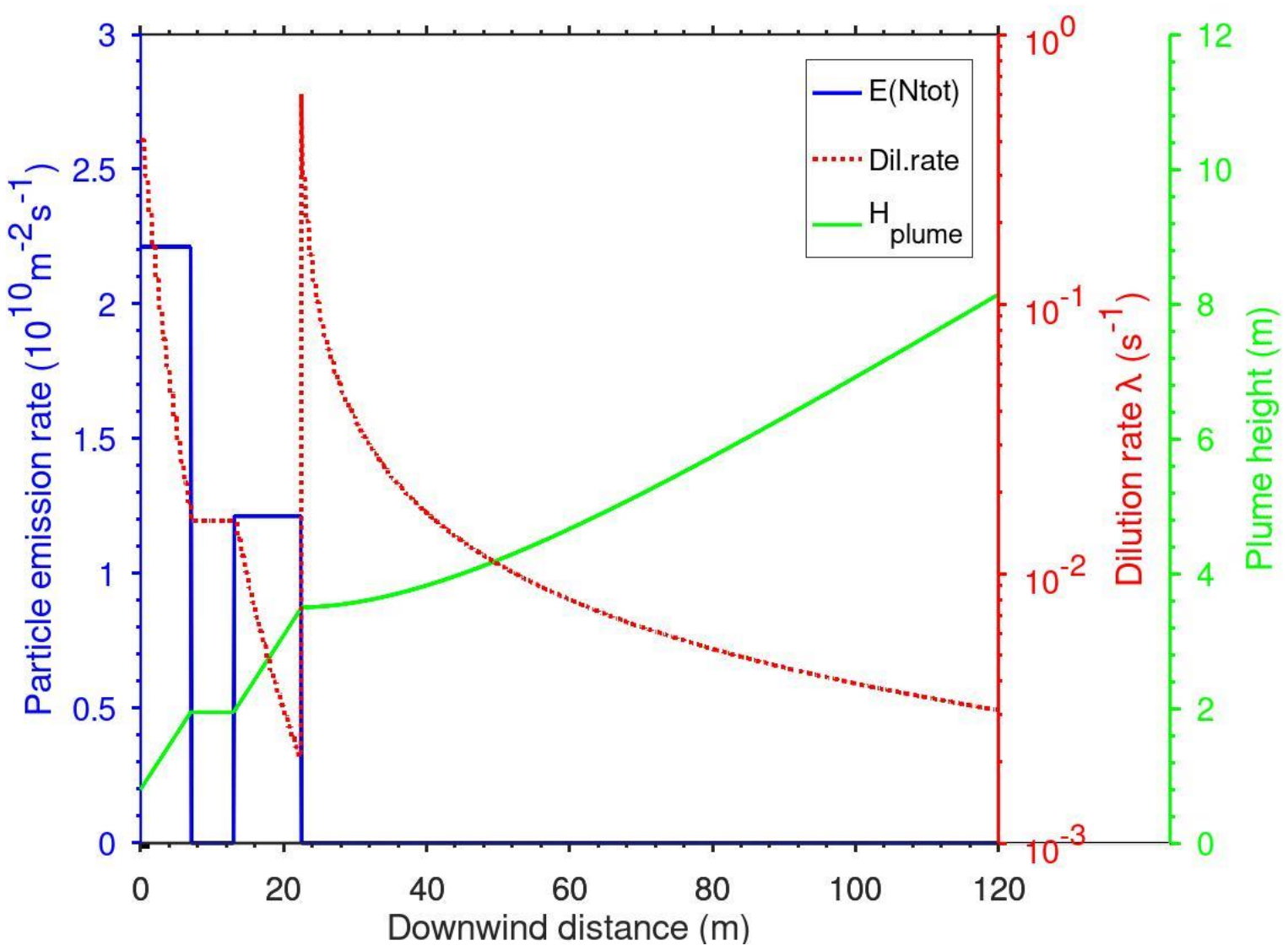

**Figure E1.** Plot of emission rate of exhaust particles in $10^{10}\,m^{-2}\,s^{-1}$ (blue line), dilution rate, $\lambda_{dil}$, (red dotted line), and air parcel height (green line) as function of time after simulation start in the "Urban Case" scenario. The plume height and the dilution rate are constant while the air parcel passes over the tram tracks in the middle of the street.

The evolution of the plume height, $H_p$, during the first stage is derived from Eq. (E2), assuming a circular plume cross-section:

$$H_{p,1}(t) = \sqrt{\frac{S(t)}{\pi}}. \tag{E3}$$

The dilution ratio $D_R$ in the vehicle exhaust plumes increases approximately linearly with time during the first seconds of the dilution. The dilution ratio is given by:

$$D_R(t) \cong 1 + \frac{dD_R}{dt}t, \tag{E4}$$

The change of the dilution ratio with time, $dD_R/dt$ is obtained from the derivation of $S(t)/S_0$,

$$\frac{dD_R}{dt} = \frac{-2\alpha^2 u_0^2 t + 2\sigma_w(0)\cdot\left(\sqrt{S_0} + \sigma_w(0)t\right)}{S_0}. \tag{E5}$$

The particle dilution rate as a function of time for the first dilution stage is:

$$\lambda_{dil}(t) = \frac{dD_R}{dt}\cdot\frac{1}{D_R^2}, \tag{E6}$$

For the second dilution stage, between the downwind kerbside and ambient, the dispersion situation was analysed with the simplified street canyon model SSCM, a component of EPISODE-CityChem (Karl et al., 2019), using realistic street canyon

geometry, line source emissions of total particles in both directions of the street and the meteorological conditions of the "Urban Case". Modelled total PN concentrations were obtained at certain receptor points, located perpendicular to the street in downwind direction, beginning at the kerbside, in distances of 10 m. A numerical power function was fit to the modelled PN concentration data. The resulting fit equation for total particle number concentration was found to be $N_{tot}\,(\mathrm{cm}^{-3}) = 1.24 \times 10^5\, d^{-0.306}$, with downwind distance $d$ from kerbside in m. The dilution parameter $b = 0.306 \pm 0.05$ is close to the reported value of 0.34 in Pirjola et al. (2012) that was derived from PN measurements. The obtained parameter $b = 0.306$ is used in Eq. (25) to calculate the change of particle number concentration with time due to dilution in the aerosol dynamics models.

The height of the air parcel, containing the vehicle exhaust, as function of time during the second dilution stage is given by:

$$H_{p,2}(t) = \sqrt{H_{p,0}^2 + \left(a' \cdot (10^{-3} U t)^{b'}\right)^2}, \tag{E7}$$

where $H_{p,0}$ is the height of the plume at the end of the first dilution stage, while $a'$ and $b'$ are dispersion parameters and depend on the atmospheric stability. For stable conditions prevailing in the Urban Case scenario, $a' = 61.14$ and $b' = 0.91$ (Petersen, 1980) were chosen. The evolution of air parcel height, dilution rate and particle emission rates during the Urban Case scenario simulation is shown in Fig. E1.

## Appendix F: Statistical indicators and model performance indicators

Statistical performance indicators for the model-observation comparison were calculated with the modStats function of the openair R package (Carslaw and Ropkins, 2012). The mean absolute error, MAE (also named mean gross error, MGE), is defined as:

$$\mathrm{MAE} = \frac{1}{N_\mathrm{o}} \sum_{n=1}^{N_\mathrm{o}} |\mathrm{M}_n - \mathrm{O}_n|\,, \tag{F1}$$

where M and O stand for the model and observation results, respectively, and $N_\mathrm{o}$ is the number of observations. The use of MAE compared to measures that are based on squared differences was preferred here, because the absolute values of the differences are less sensitive to high values.

Two measures of model performance were selected, the index of agreement (IOA) and the coefficient of efficiency (COE). In this study, the COE is used to rank the models according to their performance in predictive capability.

The calculation procedure of COE in openair is based on Legates and McCabe (1999). COE = 1 indicates a perfect model. COE = 0.0 indicates a model that is no better than the observed mean, therefore such a model can have no predictive advantage. If COE takes negative values, the model is less effective than the observed mean in predicting the variation in the observations. COE is defined as:

$$\mathrm{COE} = 1 - \frac{\sum_{n=1}^{N_\mathrm{o}} |\mathrm{M}_n - \mathrm{O}_n|}{\sum_{n=1}^{N_\mathrm{o}} |\mathrm{O}_n - \overline{\mathrm{O}}|}\,. \tag{F2}$$

The O with overbar is the observation mean.

The index of agreement (IOA) is a refined index for measuring model skill (Willmott et al., 2012). IOA spans values between -1 and +1 with values approaching +1 representing better model performance. When IOA = 0.0, it signifies that the sum of the magnitudes of the errors and the sum of the perfect-model deviation and observed deviation magnitudes are equivalent. Some caution is needed when IOA approaches -1, because it can either mean that the model-estimated deviations about O are poor estimates of the observed deviations or that there simply is little observed variability. IOA is defined as:

$$\mathrm{IOA} = \begin{cases} 1 - \dfrac{\sum_{n=1}^{N_o} |\mathrm{M}_n - \mathrm{O}_n|}{2\sum_{n=1}^{N_o} |\mathrm{O}_n - \overline{\mathrm{O}}|} & \text{if } \sum\limits_{n=1}^{N_o} |\mathrm{M}_n - \mathrm{O}_n| \leq 2\sum\limits_{n=1}^{N_o} |\mathrm{O}_n - \overline{\mathrm{O}}| \\ \\ \dfrac{2\sum_{n=1}^{N_o} |\mathrm{O}_n - \overline{\mathrm{O}}|}{\sum_{n=1}^{N_o} |\mathrm{M}_n - \mathrm{O}_n|} - 1 & \text{if } \sum\limits_{n=1}^{N_o} |\mathrm{M}_n - \mathrm{O}_n| > 2\sum\limits_{n=1}^{N_o} |\mathrm{O}_n - \overline{\mathrm{O}}| \end{cases} \quad \text{(F3)}$$

## Appendix G: Comparison of modelled and measured mass size distributions

Modelled mass size distributions ($dM/d\log D_p$) of total particles obtained for the reference run (all processes) and the sensitivity runs with different representation of condensable organic vapours were compared to the measured mass size distributions. The measured mass size distribution was obtained from particle number data observations with SMPS (138 size sections in the range of 3–420 nm; 150 s resolution data; on-board the mobile lab Sniffer), assuming a particle density of 1000 kg m$^{-3}$. For points A and D, the modelled mass size distributions of the reference run and the five sensitivity tests for condensation of organics are plotted together with the measured mass size distributions in Fig. G1.

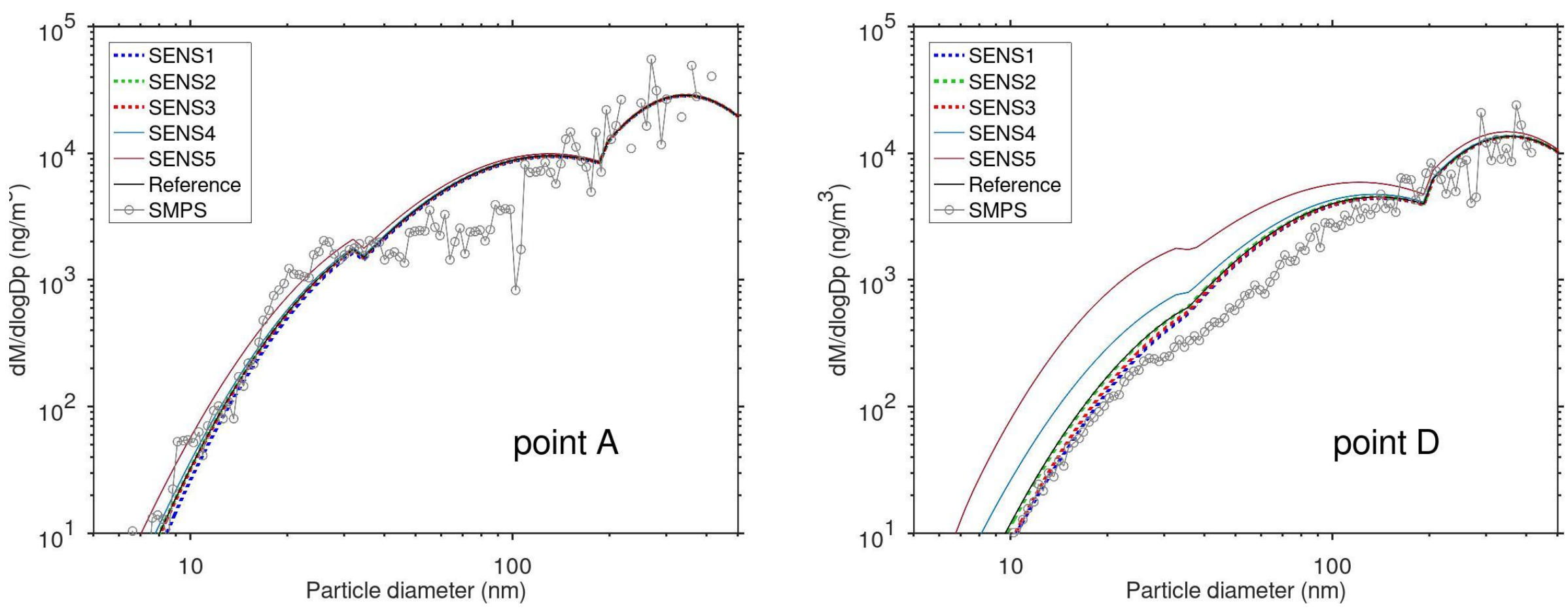

**Figure G1.** Comparison of particle mass size distribution for the diameter range 5–500 nm in the "Urban Case" simulation. Plots show the modelled mass size distribution from the reference run (including all processes) and the five condensation sensitivity tests (SENS1 to SENS5) with MAFOR together with the observed mass size distribution derived from SMPS measurements using particle density of 1000 kg m$^{-3}$ to convert from number to mass: a) size distribution of the total mass at point A; and b) size distribution of the total mass at point D.

The modelled mass size distribution obtained in the reference run matches the measured distribution at point A closely, except for the size range 40-100 nm where the model overestimates measured mass (Fig. G1a), mainly because of inaccurate particle emissions in this size range. Increased volatility of the semi-volatile organics (SENS1) and lower accommodation coefficient (SENS3) to some extent suppressed the condensation to the sub-10 nm particles. For the sensitivity run with adipic acid (SENS2) no deviation from the reference run is apparent in the mass size distributions. The 20-fold increase of SVOC emissions (SENS4) increases the mass concentrations of sub-10 nm particles at point D (by roughly a factor of two), but not at point A. The 50-fold increase of SVOC emissions (SENS5) increases mass concentrations of sub-10 nm particles at point A, still consistent with the measured mass size distribution. However, at point D the mass concentrations of particles with diameter $< 160$ nm are largely overestimated compared to the measurements. Given a factor of two uncertainty of the experimental mass concentration data (measurement error and uncertain particle density), the emission rate of condensable organics is bound between the reference emission ($EF_{SVOC} = 3.9 \times 10^{-7}$ g m$^{-1}$ veh$^{-1}$) as lower limit and the 20-fold emission (SENS4) as upper limit, for the model to be in agreement with observations. Based on MAFOR simulations, vehicle-emitted organics are thus determined to be on the order of $10^{-7}$ to $10^{-6}$ g m$^{-1}$ veh$^{-1}$.

*Author contributions.* MKa developed the model code of MAFOR v2, with LP contributing calculation routines for dry deposition and condensation coefficient, RS supporting the coupling to MECCA, and SJ contributing the ACDC/THN lookup-tables. All co-authors participated in developing the concept of the study. JK, MKa, LP, TG, SA and MKu designed the setup of the urban case simulation. LP prepared the experimental data for the urban case. LK did the pre-processing of the meteorological input data. MKa conducted the simulations with MAFOR v2, LK conducted the simulations with AEROFOR, and MKu conducted the simulations with SALSA. SA conducted a simulation with the dispersion-coagulation code LNMOM-DC and discussed results of the particle coagulation experiment. MKa developed the post-processing scripts for the statistical performance analysis. MKa, LP, MKu and TG analyzed the model data for the urban case intercomparison. MKa wrote the manuscript with contributions from all co-authors.

*Competing interests.* The authors declare that they have no conflict of interest

*Acknowledgements.* We thank Caroline Leck and Allan Gross for encouraging the publication of MAFOR as open source model. We want to thank Martin Seipenbusch (ParteQ GmbH) for permission to use experimental data on particle coagulation experiments. We are grateful to the following contributors for giving permission to publish their codes as part of open source MAFOR v2: Francis S. Binkowski for the aerosol water uptake routine, Tareq Hussein for the module on dry deposition onto rough surfaces, Anni Määttänen for the new parameterization on sulphuric acid-water nucleation, Joonas Merikanto for the THN code, Hanna Vehkamäki for the BHN code, and Fangqun Yu for the TIMN lookup-tables code. Rahul A. Zaveri is acknowledged for maintaining the MOSAIC code in WRF-Chem.

*Financial Support.* This research has been supported by the European Union's Horizon 2020 research and innovation programme under grant agreement No 814893 (the project "Shipping Contributions to Inland Pollution Push for the Enforcement of Regulations", SCIPPER) and under grant agreement No 874990 (the project "Evaluation, control and Mitigation of the EnviRonmental impacts of shippinG Emissions", EMERGE). This work reflects only the authors' view and the Innovation and European Climate, Infrastructure and Environment Executive Agency (CINEA) is not responsible for any use that may be made of the information it contains. This research has also received funding by the Academy of Finland for the project "Global health risks related to atmospheric composition and weather (GLORIA)".

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
