# Peer review of "Description and evaluation of the community aerosol dynamics model MAFOR v2.0"

_Geoscientific Model Development, 2021_

## Author Response (AR1)

**Changes to manuscript ms-nr gmd-2021-397**

**Description and evaluation of the community aerosol dynamics model MAFOR v2.0**

Matthias Karl (1), Liisa Pirjola (2,11), Tiia Grönholm (3), Mona Kurppa (3), Srinivasan Anand (4), Xiaole Zhang (5), Andreas Held (6), Rolf Sander (7), Miikka Dal Maso (8), David Topping (9), Shuai Jiang (10), Leena Kangas (3), and Jaakko Kukkonen (3, 12).

(1) Chemistry Transport Modelling, Helmholtz-Zentrum Hereon, 21502 Geesthacht, Germany.

(2) Department of Physics, University of Helsinki, P.O. Box 64, 00014 Helsinki, Finland.

(3) Atmospheric Composition Research, Finnish Meteorological Institute, P.O. Box 503, 00101 Helsinki, Finland.

(4) Health Physics Division, Bhabha Atomic Research Centre, Mumbai - 400085, India.

(5) Institute of Environmental Engineering (IfU), ETH Zürich, Zürich, CH-8093, Switzerland.

(6) Environmental Chemistry and Air Research, Technische Universität Berlin, 10623 Berlin, Germany.

(7) Air Chemistry Department, Max-Planck Institute of Chemistry, P.O. Box 3060, 55020 Mainz, Germany

(8) Aerosol Physics, Faculty of Engineering and Natural Sciences, Tampere University, PO Box 692, 33101 Tampere, Finland.

(9) Department of Earth and Environmental Science, University of Manchester, Manchester, Oxford Road, M13 9PL.

(10) School of Information Science and Technology, University of Science and Technology of China, Hefei, Anhui, 230026, China.

(11) Department of Automotive and Mechanical Engineering, Metropolia University of Applied Sciences, P.O. Box 4071, FI-01600, Vantaa, Finland.

(12) Centre for Atmospheric and Climate Physics Research, and Centre for Climate Change Research, University of Hertfordshire; College Lane, Hatfield, AL10 9AB, UK.

**Dear Prof Dr Samuel Rémy,**

We highly appreciate the reviews of our manuscript ms-nr gmd-2021-397 that we received from two anonymous referees. We have replied to their comments in the Open Discussion. We have addressed all specific comments in the revised manuscript as will be described below. We carefully considered the concerns expressed by the referees in our revision of the manuscript. In accordance with the reviewer comments, we have modified the manuscript for a clearer presentation of the new developments in the MAFOR model.

Below follows: (1) the point-by-point replies to the two reviewers, (2) a list of relevant changes in the manuscript, and (3) the revised manuscript with changes highlighted.

The comments of the reviewers have been presented in blue font, and our response as plain black text. We have considered the comments of the reviewers point-by-point. The response to the reviewers follows the sequence: (i) comment from reviewer, (ii) author's response, (iii) author's changes in manuscript.

Figure, table, section, and page numbers in the replies to the referees refer to the original manuscript. The revised manuscript with changes highlighted is sent along with this response.

This paper is an extensive presentation and evaluation of version 2 of MAFOR, an open source aerosol dynamics model coupled to a multiphase chemistry module (781 species and 2220 reactions in the gas phase, as well as 152 species and 465 reactions in the aqueous phase). First, the authors describe the structure of the model, the aerosol and chemical processes and the main updates compared to the first publication of the model. Then, they present the performance of the model with respect to its ability to predict particle and mass number size distributions. The new features of the model investigated, include the evaluation of (1) the model's sectional representation of the aerosol size distribution in a scenario of new particle formation in urban areas ("Case 1"); (2) Brownian coagulation under the condition of continuous injection of nanoparticles ("Case 2"); and (3) the dynamic treatment of semi-volatile inorganic gases by condensation and dissolution ("Case 3"). They also tested the model in a real-world scenario of a street canyon environment, in comparison with other aerosol process models (AEROFOR and SALSA) and experimental data. The authors conclude that the model is well suited for studying changes of the emitted particle size distributions by aerosol processes of organic vapours in urban environments and also for the simulation of new particle formation over multiple days. They also present some future developments of the model in view of application in urban settings.

The structure of the model is presented with clarity, giving the necessary critical information to the reader. The same goes for aerosol and chemical processes which are explained in sufficient detail to prevent potential misconceptions. Methods and data used, as well as the definitions of the scientific, regulatory and computational problem of interest are clearly stated and discussed. The manuscript is well written, has important environmental message, and should be of great interest to the readers. Overall, it is an important study, and should be considered for publication in GMD.

Response:

We thank the reviewer for their positive evaluation of the manuscript.

I have only one remark. The authors support that the main advantage of MAFORv2.0 is the consistent treatment of both the mass- and number-based concentrations of PM. I think that it would be of great interest for the reader if this consistency was discussed in more detail.

Response:

The authors thank the reviewer for this valuable remark. The consistent treatment of mass- and number-based concentrations of particulate matter in the model has several aspects: 1) initialization of the aerosol size distribution, 2) the insertion of particles from aerosol source emissions, 3) the mathematical solution of the aerosol dynamics processes, and 4) the comparability to observed PM mass concentrations and number concentrations.

In the MAFOR model, the aerosol is initialized based on the modal mass composition, which is then distributed over the size bins of the model (Eq. (21)) and converted to number based on the material density of the different aerosol components, assuming spherical particles. This way, it is assured that the initial aerosol is consistent in terms of mass and number. MAFOR v2.0 solves simultaneously for each size section, the number concentrations and mass concentrations as they change with time due to different aerosol dynamics processes in a given scenario. This has two advantages: 1) it takes into account the concurrent change of average particle density during the evolution of an aerosol size distribution in the prediction of number and mass concentrations, and 2) it represents the growth of particles in terms of number and mass. Finally, the output of modelled particle number size distribution and mass concentration size distribution can be directly compared to observed number and mass concentration size distributions, respectively.

Some of the above-mentioned aspects have uncertainties and limitations, which will lead to a certain deviation from the full consistency of number and mass.

The initialization based on mass concentrations of the PM components relies on a modal distribution, which may smooth variations between size sections of an observed aerosol size distribution. In addition, the mass composition in the nucleation, Aitken, Accommodation and coarse modes of the aerosol is sometimes not known and has to be guessed based on literature estimates.

In the real-world scenario of a street canyon environment, there is a problem that particle emissions are reported on the basis of numbers, while emissions in the MAFOR model are inserted on the basis of mass, and then converted to number based on density assumptions. The total PN emission factor is dependent on the set-ups of the measurements (Kukkonen et al., 2016): firstly it may include either only solid particles or solid and volatile PN, and secondly has a variable lower particle size cut-off, depending on the instrumental method employed.

In the case of the street canyon simulation, the PN emission factor was adopted from the study of Kurppa et al. (2020) and emissions were distributed over the particle size distribution so that the modelled size distribution after 5.5 m distance from start matched with the measurement of the particle size distribution on lane 2 of the street canyon. The attribution of particle emissions to a modal distribution in MAFOR v2.0 is a limitation of the model. The MAFOR model represented the variation of particle emissions between different size bins less well than the two other models, SALSA and AEROFOR, which used a bin-wise representation, in particular for the particles with sizes below 20 nm diameter.

When comparing the modelled total particle mass concentration distribution to observations from ELPI in the street canyon scenario, we have assumed that all particles are spheres and

have the same density of 1000 kg/m$^3$. The ELPI charging efficiency depends on particle mobility diameter, whereas the ELPI measures the aerodynamic diameter of particles. This dilemma is usually circumvented by assuming that the particles are unit density spheres, for which mobility diameter equals aerodynamic diameter. For soot particles that form as agglomerates of approximately spherical primary particles with 10–30 nm diameter, the effective density decreases with particle growth. This in turn narrows their aerodynamic size distribution relative to their mobility distribution. The uncertainty due to changes in effective density of soot particles are estimated to cause a systematic error for the determination of PM with ELPI of about 20 % (Maricq et al., 2006). Salo et al. (2019) compared ELPI+ to PM$_{10}$ cascade impactors in combustion emission measurements. ELPI+ mass concentrations were larger for most combustion cases, probably because firstly, the effective density of the particles was not the assumed unit density and secondly, volatile particles were measured by ELPI+, but not with the cascade impactors.

DeCarlo et al. (2004) mention two issues that affect the conversion of particulate matter mass to numbers: ultrafine particles with irregular shape and the internal void volumes of diesel soot agglomerates. Therefore, the evaluation of modelled total mass concentration in comparison against the measurements relies on the assumption of spherical particles without internal voids.

Changes in the manuscript:

We will include the above discussion on the consistent treatment of mass- and number-based concentrations of PM in a new section 4.2.3 in the revised manuscript.

References:

DeCarlo, P. F., Slowik, J. G., Worsnop, D. R., Davidovits, P., and Jimenez, J. L.: Particle morphology and density characterization by combined mobility and aerodynamic diameter measurements. Part 1: Theory, Aerosol Sci. Technol., 38(12), 1185–1205, doi:10.1080/027868290903907, 2004.

Kukkonen, J., Karl, M., Keuken, M. P., Denier van der Gon, H.A.C., Denby, B. R., Singh, V., Douros, J., Manders, A., Samaras, Z., Moussiopoulos, N., Jonkers, S., Aarnio, M., Karppinen, A., Kangas, L., Lützenkirchen, S., Petäjä, T., Vouitsis, I., and Sokhi, R. S.: Modelling the dispersion of particle numbers in five European cities, Geosci. Model Dev., 9, 451–478, doi:10.5194/gmd-9-451-2016, 2016.

Kurppa, M., Roldin, P., Strömberg, J., Balling, A., Karttunen, S., Kuuluvainen, H., Niemi, J.V., Pirjola, L., Rönkkö, T., Timonen, H., Hellsten, A., and Järvi, L.: Sensitivity of spatial aerosol particle distributions to the boundary conditions in the PALM model system 6.0, Geosci. Model Dev., 13, 5663–5685, https://doi.org/10.5194/gmd-13-5663-2020, 2020.

Maricq, M. M., Xu, N., and Chase, R. E.: Measuring particulate mass emissions with the Electrical Low Pressure Impactor, Aerosol Sci. Technol., 40(1), doi:10.1080/02786820500466591, 68–79, 2006.

Salo, L., Mylläri, F., Maasikmets, M., Niemelä, V., Konist, A., Vainumäe, K., Kupri, H.-L., Titova, R., Simonen, P., Aurela, M., Bloss, M., Keskinen, J., Timonen, H., and Rönkkö, T.: Emission measurements with gravimetric impactors and electrical devices: An aerosol instrument comparison, Aerosol Sci. Technol., 53(5), 526–539, https://doi.org/10.1080/02786826.2019.1578858, 2019.

**Referee #2**

The manuscript describes an open-source aerosol model, which includes all the basic aerosol microphysical processes as well as multiphase chemistry. It is an updated and extended version of an earlier published version MAFOR1.0. Such a model package can be useful to the aerosol modeling community, as such open-source models are still rare. I have some remarks that have to be addressed. Especially, I am concerned that the main improvements of the model (compared with the earlier version) might not be tested in the evaluation part.

Response:

We thank the reviewer for carefully evaluating our manuscript and for giving valuable comments and suggestions.

Given the complexity of the model, which includes multiphase chemistry in addition to aerosol dynamics, and can be operated in different ways, it is impossible to test and evaluate all new features in one paper. The main improvements of MAFOR v2 (compared with the earlier version v1) were tested in three numerical experiments ("Case 1", "Case 2", and "Case 3") that are given in the Supplement. Case 1 is a hypothetical case of an urban environment based on experimental data from the SAPPHIRE campaign in Helsinki (Hussein et al., 2007). This case was designed as an experiment for the numerical diffusion problem, but also included the new nucleation model of Määttänen et al. (2018). Case 2 evaluated the extension of the Brownian coagulation to consider the fractal geometry of soot particles, under the condition of continuous injection of nanoparticles. Case 3 evaluated the dynamic treatment of semi-volatile inorganic gases by condensation and dissolution. The improvement on the condensation/evaporation of semi-volatile organic vapors with different properties was evaluated partly in the street canyon case ("Urban Case"). An additional test will be performed that uses the Case 1 conditions to evaluate the new nucleation mechanism (Määttänen et al., 2018) and particle growth in MAFOR v2 in comparison to a simulation with the AEROFOR model.

A previous study by Pirjola et al. (2015) on formation of particles in the exhaust of a diesel engine equipped with an oxidative after-treatment system showed good agreement between MAFOR v1 and AEROFOR, when using the homogeneous nucleation of sulfuric acid along with heteromolecular nucleation between sulfuric acid and organic vapor molecules (HET). Both models were able to predict the particle number size distributions in agreement with the measured size distributions at different stages of dilution and ageing. Ship plume simulations with MAFOR v1 (Karl et al., 2020) that considered classical binary homogeneous $H_2SO_4$-$H_2O$ nucleation extended to high temperatures and coupling with gas-phase chemistry from MECCA v3.0 showed that the model was able to predict total PN concentrations in most of the 14 different ship plume events within ±50 % of the measurements. The basic photochemistry also worked reasonably in the ship plumes, with fast formation of sulfuric acid by addition of water molecules to the co-emitted $SO_3$ in the first seconds of the simulation. The initial peak of sulfuric acid quickly dropped to ambient levels, followed by in-plume photochemical formation

through oxidation of $SO_2$ by OH radical during daytime, resulting in maximum $H_2SO_4$ concentrations of $1-2 \times 10^8$ $cm^{-3}$ after the first minute in the ship plumes.

SOA formation with the new VOC chemistry (in CAABA/MECAA v4.0) has not been tested until now, but it is planned in the near future to study secondary aerosol formation in the vehicle exhaust with the updated model.

Changes in the manuscript:

We will add a new section 4.2.4 in the discussion chapter that points out how each of the new features of MAFOR v2.0 is evaluated and to make the numerical experiments of the Supplement more visible.

References:

Hussein, T., Kukkonen, J., Korhonen, H., Pohjola, M., Pirjola, L., Wraith, D., Härkönen, J., Teinilä, K., Koponen, I. K., Karppinen, A., Hillamo, R., and Kulmala, M.: Evaluation and modeling of the size fractionated aerosol particle number concentration measurements nearby a major road in Helsinki – Part II: Aerosol measurements within the SAPPHIRE project, Atmos. Chem. Phys., 7, 4081–4094, doi:10.5194/acp-7-4081-2007, 2007.

Karl, M., Pirjola, L., Karppinen, A., Jalkanen, J.-P., Ramacher, M. O. P., and Kukkonen, J.: Modeling of the concentrations of ultrafine particles in the plumes of ships in the vicinity of major harbors, Int. J. Environ. Res. Public Health, 17, 777, 1–24, https://doi.org/10.3390/ijerph17030777, 2020.

Määttänen, A., Merikanto, J., Henschel, H., Duplissy, J., Makkonen, R., Ortega, I. K., and Vehkamäki, H.: New parameterizations for neutral and ion-induced sulfuric acid-water particle formation in nucleation and kinetic regimes, J. Geophys. Res. Atmospheres, 123, 1269–1296, https://doi.org/10.1002/2017JD027429, 2018.

Pirjola, L., Karl, M., Rönkkö, T., and Arnold, F.: Model studies of volatile diesel exhaust particle formation: are organic vapours involved in nucleation and growth?, Atmos. Chem. Phys., 15, 10435–10452, https://doi.org/10.5194/acp-15-10435-2015, 2015.

**Specific Comments**

1. a) As a major advantage of the described model, "consistent treatment of both the mass and number-based concentrations of particulate matter" is listed (already in the abstract). If I have understood the details of MAFOR1.0 correctly, the choice of solution was already similar, which means that this feature is not a novelty in 2.0?

Response:

The authors thank the reviewer for this valuable remark. The simultaneous solution of number and mass-based concentrations was already implemented in the structure of MAFOR v1.0. As the reviewer correctly states, this is not a new feature of the model. We think that the simultaneous solution of number and mass-based concentration (in each size section) in this model makes it rather unique, since there are only few other models with inherently consistent

solution of both aerosol number and mass (e.g., two-moment sectional models). The "consistent treatment" refers to the simultaneous solution of the aerosol dynamics processes in terms of number and mass. This will be mentioned more clearly in the manuscript. The MAFOR model tracks for each size section, the total particle number concentration and the particles mass concentration of each aerosol component. This has two advantages: 1) it takes into account the concurrent change of average particle density during the evolution of an aerosol size distribution in the prediction of number and mass concentrations, and 2) it represents the growth of particles in terms of number and mass. In the MAFOR model, the particle emissions are input on mass basis. This procedure introduces some inaccuracy, as emitted particles are not always spherical and can contain internal void volume. This is a general problem for all aerosol process models. The emission treatment in MAFOR also causes inaccuracies in reproducing the bin-resolved emission distribution, in particular for particles with diameter smaller than 20 nm. We discuss the problems related to the consistency of mass- and number-based particle concentrations in more detail in our response to Referee #1.

Changes in the manuscript:

We clarify the sentence (Abstract):

*"The model simultaneously solves the time evolution of both the particle number and the mass concentrations of aerosol components in each size section. In this way, the model can also allow for the changes of the average density of particles."*

Changed sentence on page 4, lines 3-5:

*"The aerosol dynamics module of MAFOR simultaneously solves the time evolution of particle number concentration and mass concentration of aerosol components in each size section in a consistent manner. The model allows for the changes in the average density of particles and represents the growth of particles in terms of both the particle number and mass."*

Changed sentence on page 53, lines 2-3:

*"A major advantage of the model is the consistent treatment of particle number concentrations and mass concentrations of each aerosol component through the simultaneous solution of aerosol dynamics processes in terms of number and mass. This procedure allows the changes in the average density of particles to affect the predicted number and mass size distributions."*

1. b) In addition, it is stated that this is a feature that is an advantage "compared to the other sectional aerosol process models" (Summary, page 56, lines 28-30). Does this mean the models that were compared against here (AEROFOR, SALSA), or more generally? In either case, this is a strong statement, which needs more careful justification.

Response:

The referred sentence on page 56 will be removed.

1. c) E.g., in SALSA, you state that "SALSA outputs volume size distributions of particle components, which at known density can be translated to mass concentration" (page 53, lines 22-23). What is the inconsistency?

Response:

In the SALSA model, particle mass concentrations can only be determined from the volume distribution output. This involves the assumption of a constant average particle density during the simulation. However, the average density of particles is changing during the calculation, for instance by condensation of less dense particulate matter. In addition, the average density of particles can vary between different particle sizes.

1. d) And, further-furthermore, how about two-moment sectional models such as TOMAS (e.g. Lee and Adams, 2012) or GLOMAP (Spracklen et al., 2005)? Are they also prone to inconsistency regarding both mass and number-based concentrations?

Response:

The authors thank the reviewer for bringing those two-moment sectional models to our attention. While the single-moment sectional approaches usually only track either number or mass in each section, the two-moment sectional approach explicitly tracks both aerosol number (the zeroth moment) and mass (first moment) in each size section. The two-moment approach can conserve both number and mass very accurately (e.g. Adams and Seinfeld, 2002). Clearly, all two-moment models that track aerosol number and mass are unaffected by the inconsistency arising from representing particle growth only in terms of changes in particle number in each bin.

Changes in the manuscript:

We will include the explanations on single-moment and two-moment sectional approaches in section 2.1 ("Review of current aerosol process models").

*"First attempts to solve the stochastic collection equation for a droplet size distribution have used a single-moment sectional approach, which tracks either particle number or particle mass. Later, two-moment sectional models were developed, which explicitly track both particle number (i.e., zeroth moment) and the mass concentration of aerosol components (i.e., first moment) in each size bin, to predict the particle number and mass size distributions (Tzivion et al., 1987). The two-moment sectional approach can conserve both number and mass very accurately (Adam and Seinfeld, 2002). Two-moment sectional models have been implemented in global aerosol microphysics models for improving the understanding of the processes that control concentrations of cloud condensation nuclei (CCN), for example the climate model GISS-TOMAS (Lee and Adams, 2010) and the global offline-CTM model GLOMAP (Spracklen et al., 2005)."*

References:

Adams, P. J., and Seinfeld, J. H.: Predicting Global Aerosol Size Distributions in General Circulation Models, J. Geophys. Res., 107(D19), 4370, 1–23, doi:10.1029/2001JD001010, 2002.

Lee, Y. H. and Adams, P. J.: Evaluation of aerosol distributions in the GISS-TOMAS global aerosol microphysics model with remote sensing observations, Atmos. Chem. Phys., 10, 2129–2144, https://doi.org/10.5194/acp-10-2129-2010, 2010.

Spracklen, D. V., Pringle, K. J., Carslaw, K. S., Chipperfield, M. P., and Mann, G. W.: A global off-line model of size-resolved aerosol microphysics: I. Model development and prediction of aerosol properties, Atmos. Chem. Phys., 5, 2227–2252, https://doi.org/10.5194/acp-5-2227-2005, 2005.

Tzivion, S., Feingold, G., and Levin, Z.: An efficient numerical-solution to the stochastic collection equation, J. Atmos. Sci., 44, 3139–3149, 1987.

2. Page 4, lines 7-10, statement about integrating aerosol dynamics with gas phase chemistry. It is true that many of the aerosol dynamics models do not contain gas phase chemistry. However, many of them are designed to be coupled with a separate gas phase chemistry model. One example is the HAM-model (Bergman et al., 2021), which includes SALSA, and has been implemented into the ECHAM5 climate model. Would it thus be more 'fair' to compare the contents of MAFOR to HAM (and other such models) instead of the pure aerosol dynamic packages, when considering what they contain (Table1)?

Response:

The focus of the paper is on the evaluation of aerosol dynamics processes implemented in MAFOR v2 and their numerical solution. There are only a few other sectional aerosol dynamics models for use in atmospheric studies that are inherently coupled to a gas-phase chemistry scheme in their stand-alone version, as referred in the mentioned text on page 4 of the Introduction. It is noted that the Hamburg Aerosol Model (HAM) (Stier et al., 2005) which handles the emissions, removal and microphysics of aerosol particles model within ECHAM5 basically includes a chemistry scheme for the oxidation of DMS. In the previous publication of Karl et al. (2011), the coupling of aerosol dynamics with gas-phase chemistry, mainly with respect to gas-phase reaction products from DMS chemistry using a comprehensive reaction scheme, has been evaluated in a marine transport scenario under clear sky conditions for the sectional models MAFOR v1, AEROFOR and the monodisperse model MONO32 (Pirjola et al., 2003). However, such large-scale experimental field-data with publicly available observations for both the particulate phase and the gas-phase constituents under favorable meteorological conditions (clear sky and stably stratified boundary layer) are rare. Moreover, we feel that it is beyond the scope of this study to compare to the aerosol microphysical modules that are used in large scale 3-D modelling and climate models. Such a comparison would certainly require a dedicated study, entailing the implementation of different aerosol dynamics modules in the same climate model. We will add a statement in the manuscript, saying that many aerosol dynamics models are designed to be coupled with a separate gas phase chemistry module, when implemented in 3-D models.

Changes in the manuscript:

The sentences on page 4, lines 7-10, mentioned by the reviewer will be changed as follows:

*"The aerosol dynamics in MAFOR are coupled to a detailed gas-phase chemistry module, which offers full flexibility for inclusion of new chemical species and reactions. Many aerosol dynamics models are designed to be coupled with a separate gas-phase chemistry module, when implemented in atmospheric 3-D models. However, there exist only a few other aerosol dynamics models for use in atmospheric studies that inherently integrate gas-phase chemistry together with aerosol processes as a function of time."*

The text regarding implementation of SALSA in 3-D models in section 2.1 ("Review of current aerosol process models") will be changed as follows:

*"The aerosol process models M7 (Vignati et al., 2004) and SALSA (Kokkola et al., 2008), partly owing to their computationally efficiency, have been implemented into the 3-D aerosol-climate model ECHAM5 (Bergman et al., 2012). SALSA is a sectional aerosol module, developed with the specific purpose for implementation in large scale models. It is part of the Hamburg Aerosol Model (HAM) (Stier et al., 2005) that handles the emissions, removal and microphysics of aerosol particles, and the gas-phase chemistry of dimethyl sulphide (DMS) within ECHAM5."*

In section 4.3.2 ("Integration in 3-D atmospheric models") we add the following sentence:

*"The MAFOR box model inherently includes coupling to a detailed gas-phase chemistry. However, the aerosol dynamics solver can be applied as a separate module in 3-D atmospheric models."*

References:

Bergman, T., Kerminen, V.-M., Korhonen, H., Lehtinen, K. J., Makkonen, R., Arola, A., Mielonen, T., Romakkaniemi, S., Kulmala, M. and Kokkola, H.: Evaluation of the sectional aerosol microphysics module SALSA implementation in ECHAM5-HAMaerosol climate model, Geosci. Model Dev., 5, 845–868, https://doi.org/10.5194/gmd-5-845-2012, 2012.

Karl, M., Gross, A., Pirjola, L., and Leck, C.: A new flexible multicomponent model for the study of aerosol dynamics in the marine boundary layer, Tellus B, 63, 1001–1025, https://doi.org/10.1111/j.1600-0889.2011.00562.x, 2011.

Kokkola, H., Korhonen, H., Lehtinen, K. E. J., Makkonen, R., Asmi, A., Järvenoja, S., Anttila, T., Partanen, A.-I., Kulmala, M., Järvinen, H., Laaksonen, A., and Kerminen, V.-M.: SALSA - a Sectional Aerosol module for Large Scale Applications, Atmos. Chem. Phys., 8, 2469–2483, https://doi.org/10.5194/acp-8-2469-2008, 2008.

Pirjola, L., Tsyro, S., Tarrason, L., and Kulmala, M.: A monodisperse aerosol dynamics module – a promising candidate for use in the Eulerian long-range transport model, J. Geophys. Res., 108( D9), 4258, doi:10.1029/2002JD002867, 2003.

Stier, P., Feichter, J., Kinne, S., Kloster, S., Vignati, E., Wilson, J., Ganzeveld, L., Tegen, I., Werner, M., Balkanski, Y., Schulz, M., Boucher, O., Minikin, A., and Petzold, A.: The aerosol-climate model ECHAM5-HAM, Atmos. Chem. Phys., 5, 1125–1156, doi:10.5194/acp-5-1125-2005, 2005.

Vignati, E., Wilson, J., and Stier, P.: M7: An efficient size-resolved aerosol microphysics module for large-scale aerosol transport models, J. Geophys. Res., 109, D22202, doi:10.1029/2003JD004485, 2004.

3. I really appreciate the clarity by which the differences between versions 2.0 and 1.0 are explained (section 2, page 5). However, what remains unclear is what is the role of these new developments in improving the results presented in this manuscript (section 4), where dilution and aerosol dry deposition seem to dominate the dynamics, and, e.g. the new nucleation models seem to play no role. It feels like very little of the listed novelties are actually tested here???

Response:

We were aware of this difficulty when we selected the street canyon case ("Urban Case") for the evaluation of the model. The street canyon case was selected due to its relevance for defining the aerosol processes that need to be considered in urban dispersion models. The plume dispersion simulation considers the scale between the release of exhaust and the roadside, for which the aerosol dynamics processes are typically not resolved in city-scale dispersion models. Semi-volatile organic vapours can grow nucleation mode particles with a non-volatile core that formed in the vehicle exhaust before the dilution process (Rönkkö et al., 2007; Pirjola et al., 2015), without any significant chemical transformation in the atmosphere (Rönkkö et al., 2013). The treatment of condensable organic compounds plays a role in the evaluation (e.g., Figure 13) and the influence of different volatility was tested in section 4.1.3 ("Effect or influence of condensation/evaporation of organics"). Unfortunately, it was not possible to test the new development on formation of SOA from VOC oxidation (i.e. absorptive partitioning of organic vapours) in the street canyon case, because the gas-phase concentrations of VOC have not been measured. Note, that the new features of the nucleation model (sulphuric acid-water nucleation parameterization of Määttänen et al., 2018) and of the coagulation kernel (fractal aggregates of soot) were applied to constrain the uncertainties of the street canyon case, but did only have an limited effect on total PN concentrations and particle size distributions.

We have presented results from three numerical experiments in the Supplementary Materials, in which some of the new features of the model were evaluated in comparison to data from the literature.

In addition, we have now performed an evaluation of the implemented new nucleation model of Määttänen et al. (M2018) in a simulation of new particle formation in comparison to the AEROFOR model (that also applies M2018). Simulation of nucleation and particle growth was compared in a numerical experiment under clear sky conditions with zero background particles, mimicking conditions over the high Arctic in summer. The numerical experiment allows for comparing the rate of ion-induced and neutral nucleation, formation and growth of new particles between the two models.

SOA formation with coupled photochemistry and aerosol dynamics has been evaluated in a smog chamber experiment for the OH-initiated oxidation of 2-aminoethanol (Karl et al., 2012).

In that version of the MAFOR v1 model, the coupling was with the gas-phase chemistry scheme of MECCA v3.0. The main advantage of using the new version 4.0 of MECCA in MAFOR v2 is the much more detailed VOC chemistry of the Mainz Organic Mechanism (MOM). In a study of the oxidation processes in the Mediterranean atmosphere, simulated atmospheric hydroxyl radical (OH) concentrations with the CAABA/MECCA box model using MOM chemistry were in good agreement with in situ OH observations (Mallik et al., 2018).

Vehicles contribute to atmospheric PM concentrations not only through direct, primary PM emissions, but also even more significantly through gas-to-particle conversion of initially gaseous exhaust components and the photo-oxidation of emitted VOC (Nordin et al., 2013). While the gas-to-particle conversion of the semi-volatile vapours in the exhaust was evaluated in the simulation of the street canyon case, it was not possible to evaluate SOA formation through VOC photo-oxidation, as stated above. There is a need for investigation of the role of aromatic VOC and other gaseous precursors in the formation of SOA from gasoline and diesel vehicles. In a follow-up work, it is planned to simulate with MAFOR v2 the secondary aerosol formation in aged vehicle exhaust in a smog chamber experiment or in an oxidation flow reactor (OFR) that gives the potential aerosol mass. The model evaluation will be designed to consider the production of SOA-precursors from photochemical VOC oxidation using the mass-based formulation from the 2-D VBS framework for organic aerosol phase partitioning.

Changes in the manuscript:

We will add a new section 4.2.4 on the evaluation of the model improvements in the revised manuscript that details the evaluation of the new features one by one.

The results from the evaluation of the M2018 nucleation code in comparison against the AEROFOR model will be included in the manuscript (new section 4.2.4 and new Appendix H).

References:

Karl, M., Dye, C., Schmidbauer, N., Wisthaler, A., Mikoviny, T., D'Anna, B., Müller, M., Borrás, E., Clemente, E., Muñoz, A., Porras, R., Ródenas, M., Vázquez, M., and Brauers, T.: Study of OH-initiated degradation of 2-aminoethanol, Atmos. Chem. Phys., 12, 1881–1901, https://doi.org/10.5194/acp-12-1881-2012, 2012.

Mallik, C., Tomsche, L., Bourtsoukidis, E., Crowley, J. N., Derstroff, B., Fischer, H., Hafermann, S., Hüser, I., Javed, U., Keßel, S., Lelieveld, J., Martinez, M., Meusel, H., Novelli, A., Phillips, G. J., Pozzer, A., Reiffs, A., Sander, R., Taraborrelli, D., Sauvage, C., Schuladen, J., Su, H., Williams, J., and Harder, H.: Oxidation processes in the eastern Mediterranean atmosphere: evidence from the modelling of HOx measurements over Cyprus, Atmos. Chem. Phys., 18, 10825–10847, https://doi.org/10.5194/acp-18-10825-2018, 2018.

Määttänen, A., Merikanto, J., Henschel, H., Duplissy, J., Makkonen, R., Ortega, I. K., and Vehkamäki, H.: New parameterizations for neutral and ion-induced sulfuric acid-water particle formation in nucleation and kinetic regimes, J. Geophys. Res. Atmospheres, 123, 1269–1296, https://doi.org/10.1002/2017JD027429, 2018.

Nordin, E. Z., Eriksson, A. C., Roldin, P., Nilsson, P. T., Carlsson, J. E., Kajos, M. K., Hellén, H., Wittbom, C., Rissler, J., Löndahl, J., Swietlicki, E., Svenningsson, B., Bohgard, M., Kulmala, M., Hallquist, M., and

Pagels, J. H.: Secondary organic aerosol formation from idling gasoline passenger vehicle emissions investigated in a smog chamber, Atmos. Chem. Phys., 13, 6101–6116, https://doi.org/10.5194/acp-13-6101-2013, 2013.

Pirjola, L., Karl, M., Rönkkö, T., and Arnold, F.: Model studies of volatile diesel exhaust particle formation: are organic vapours involved in nucleation and growth?, Atmos. Chem. Phys., 15, 10435–10452, https://doi.org/10.5194/acp-15-10435-2015, 2015.

Rönkkö, T., Virtanen, A., Kannosto, J., Keskinen, J., Lappi, M., and Pirjola, L.: Nucleation mode particles with a non-volatile core in the exhaust of a heavy duty diesel vehicle, Environ. Sci. Technol., 41, 6384–6389, doi:10.1021/es0705339, 2007.

Rönkkö, T., Lähde, T., Heikkilä, J., Pirjola, L., Bauschke, U., Arnold, F., Schlager, H., Rothe, D., Yli-Ojanperä, J., and Keskinen, J.: Effects of gaseous sulphuric acid on diesel exhaust nanoparticle formation and characteristics, Environ. Sci. Technol., 47 (20), 11882–11889, 2013.

4. Table 5, page 40: As dry deposition is stated as the other major process (in addition to dilution) affecting the size distribution, and each of the models compared have a different dry deposition parameterization, is it possible that it is these differences (as well as how the particles are introduced into the beginning of the simulation) that explain much of the differences in the results.

Response:

In addition to dry deposition, the way particles are introduced in the beginning of the simulation (initial size distribution) and the numerical solver cause differences between the models. As already stated in the original manuscript (page 47, lines 3-4), the change of PN in simulations with AEROFOR was more strongly controlled by dilution than in simulations with the other models. The AEROFOR model solves the set of stiff differential equations of aerosol dynamics processes and dilution in one step with the NAG library, whereas SALSA and MAFOR use operator splitting. To assess the differences in the model results due to the application of different deposition schemes (as given in Table 5), additional model runs including all processes for the "Urban Case" scenario were performed with the MAFOR model using first SPF1985 (deposition scheme in AEROFOR) and second ZH2001 (deposition scheme in SALSA). The comparison of the final particle size distribution (at point D, after 78.5 s plume transport time) obtained from MAFOR runs with different dry deposition parameterizations is shown in the Figure C1 below.

The HU2012 deposition scheme that was used in the reference run with MAFOR is more efficient in removing particles > 10 nm diameter than the other two deposition schemes. However, differences between using SPF1985 or ZH2001 are very small, which means that the application of different dry deposition parameterizations was not the only reason for differences in the results.

[Figure]

Figure C1: Modelled size distribution with MAFOR using different dry deposition parameterizations at point D (after 78.5 s plume transport time). HU2012 is the reference configuration as shown in Figure 11 of the manuscript.

Changes in the manuscript:

Figure 4 and the text belonging to this figure was revised to consider the dry deposition velocities calculated with the ZH2001 parameterization for urban rough surfaces.

The following text on the effect of different dry deposition schemes on model results will be added in section 4.1.2 ("Importance of aerosol processes"):

*"Differences in the relative contribution of deposition in the models are most probably due to the different schemes for dry deposition in the models (Table 5). To assess the differences in the model results due to the application of different deposition schemes, additional model runs including all processes were performed with the MAFOR model using first the deposition scheme in AEROFOR, SPF1985, and second the deposition scheme in SALSA, ZH2001. The comparison of the final particle size distribution at point D is shown in Figure E2 (Appendix E), obtained from MAFOR runs with different dry deposition parameterizations. The HU2012 deposition scheme that was used in the reference run with MAFOR was more efficient in removing particles >10 nm diameter than the other two deposition schemes. However, differences between using either the scheme SPF1985 or ZH2001 were negligible, which implies that the application of different dry deposition parameterizations was not the main reason for differences of the predicted particle size distributions."*

The final size distribution plot (figure C1) has been included in Appendix E.

5. Page 30, discussion about numerical diffusion: The authors acknowledge that the fixed sectional discretization chosen is prone to numerical diffusion. They also state that this is circumvented by using enough bins. However, if the model is to be used in 3D atmospheric, or even climate models, one can typically not afford to use a very big number of bins, and this is why other methods, such as the moving center method (which is mentioned) has been proven useful. This, I believe, needs more clarification, especially since in the summary (page 58) MAFOR2.0 is envisioned "a state-of-the-art benchmark model" to be also implemented into earth system models.

Response:

It is true that large 3-D atmospheric models and climate models cannot afford to use a large number of size sections, and this has already been mentioned in the original manuscript in section 4.3.2 ("Integration in 3-D atmospheric models"). The reviewer suggests to use the moving center method due to its good performance when the size distribution is represented by a fewer number of size sections. This has already been stated in section 2.6 ("Numerical solution of the aerosol dynamics") on page 30, lines 12-20, of the original manuscript.

The fixed sectional grid has been chosen for the MAFOR model because of the advantages when simulating continuous new particle formation. Numerical diffusion of the fixed sectional method in the MAFOR model was addressed in section 2.6. An evaluation of the fixed sectional discretization has been performed in the numerical experiment Case 1, presented in the Supplement section S2. Notify that Case 1 shows that the 16 size bins causes ~10 % error and 32 bins only ~3 % error for the final total PN concentrations under those conditions. This small error is considered still acceptable when compared to measurement errors of the observed total PN concentrations. Further, the computational demand increases only slightly when using a larger number of size sections. In addition, we suggest to develop a mapping procedure between the aerosol dynamics module that attributes the higher number of size sections to the advection routine of the large scale model that uses a smaller number of size sections.

Changes in the manuscript:

Based on the above, we have modified the text on integration of the aerosol dynamics module into 3-D models in section 4.3.2 as follows:

*"With regard to implementation of the aerosol dynamics code into large scale atmospheric models it is of special interest to assess how much one can lower the accuracy of the size distribution description without compromising on the accuracy of the model results. The evaluation of the sectional size representation in Case 1 (Supplement Sect. S2) revealed that the use of 16 size sections causes a numerical error of ~10%, and the use of 32 size sections causes only an error of ~3% in the final total PN concentrations under those conditions. The error of both representations is considered still acceptable when compared to measurement errors of observed total PN concentrations. Further, the computational demand increases only slightly when using a larger number of size sections. Overall, the size representation using 32 size sections is adequate for the simulation of long periods, as the accuracy in terms of size distribution changes and total number concentration is sufficiently high, while the*

*computational demand is only 2% higher compared to the lowest tested resolution of 16 size sections.*

*Aerosol representations in large scale models are often limited to less than 20 size classes, as the particles in each size section have to be included in the advection routine and a higher number of advected species increases the computing time. Therefore, methods need to be developed for the mapping of the size representation used in the aerosol dynamics code and the advected particle species. The effect of changing the number of size classes in the 3-D model needs to be tested thoroughly.”*

6. The theory in Appendix B is a bit hard to follow. Is B2 a time-differenced result of B1? (together with B4?) And, where does B3 come from? For the reader, it would be clearer to first show the differential equation, the mention how it is discretized, and finally show the discretized equation.

Response:

We agree with the reviewer that the equations in Appendix B were difficult to follow. The description of the Analytical Predictor of Condensation in Appendix B was incomplete and also includes the solution for the growth by dissolution (equation B3). Appendix B will be revised according to the suggestions from the reviewer. We first present the differential equations that form the $N_B+1$ ordinary differential equations for condensation and also for dissolution. We then explain the integration to obtain a discretized equation for the mass concentration of compound q in size bin i. Next, we obtain the discrete equation for the final gas concentration with respect to the mass-balance equation. Finally, the limits are introduced to prevent negative values of the resulting mass concentrations.

Changes in the manuscript:

Appendix B has been revised according to the suggestions from the reviewer.

**List of relevant changes in the ms**

**Relevant text changes:**

The manuscript has been mainly changed to improve the description of the consistent treatment of the mass- and number-based concentrations of particulate matter and to improve the presentation of the evaluation of the new developments in MAFOR version 2.0. A discussion of the consistency of mass- and number-based concentrations of PM has been added in an additional discussion section 4.2.3 in the revised manuscript. An additional section 4.2.4 "Evaluation of the model improvements" has been added to briefly present the evaluation of each of the new developments.

**1.    Introduction**

Following the specific comment 1a) of Referee #2, the statement about the consistent treatment of both the mass and number-based concentrations of PM in the MAFOR model has been revised:

"The aerosol dynamics module of MAFOR simultaneously solves the time evolution of particle number concentration and mass concentration of aerosol components in each size section in a consistent manner. The model allows for the changes in the average density of particles and represents the growth of particles in terms of both the particle number and mass."

To clarify that many aerosol dynamics models can be coupled to a gas-phase chemistry mechanism when incorporated in three-dimensional models, the following was included in response to specific comment 2 of Referee #2:

"The aerosol dynamics in MAFOR are coupled to a detailed gas-phase chemistry module, which offers full flexibility for inclusion of new chemical species and reactions. Many aerosol dynamics models are designed to be coupled with a separate gas-phase chemistry module, when implemented in atmospheric 3-D models. However, there exist only a few other aerosol dynamics models for use in atmospheric studies that inherently integrate gas-phase chemistry together with aerosol processes as a function of time."

The evaluation of the new developments in MAFOR v2.0 is introduced as follows:

"Several of the new features of MAFOR version 2 were investigated in numerical scenarios and compared to reference data. Specifically, they included the evaluation of 1) the model's sectional representation of the aerosol size distribution in a scenario of new particle formation in urban areas ("Case 1"; Sect. S2, Supplementary Materials); 2) Brownian coagulation under the condition of continuous injection of nanoparticles ("Case 2"; Sect. S3); 3) the dynamic treatment of semi-volatile inorganic gases by condensation and dissolution ("Case 3"; Sect. S4); and 4) a new parameterization for nucleation in case of neutral and ion-induced particle formation (Appendix H)."

**2.        Section 2.1 "Review of current aerosol process models"**

We addressed the specific comment 1d) of Referee #2 about two-moment sectional models by including a paragraph on the single-moment and two-moment sectional approaches in section 2.1:

"First attempts to solve the stochastic collection equation for a droplet size distribution have used a single-moment sectional approach, which tracks either particle number or particle mass. Later, two-moment sectional models were developed, which explicitly track both particle number (i.e., zeroth moment) and the mass concentration of aerosol components (i.e., first moment) in each size bin, to predict the particle number and mass size distributions (Tzivion et al., 1987). The two-moment sectional approach can conserve both number and mass very accurately (Adam and Seinfeld, 2002). Two-moment sectional models have been implemented in global aerosol microphysics models for improving the understanding of the processes that control concentrations of cloud condensation nuclei (CCN), for example the climate model GISS-TOMAS (Lee and Adams, 2010) and the global offline-CTM model GLOMAP (Spracklen et al., 2005)."

We addressed the specific comment 2 of Referee #2 regarding the coupling of the aerosol dynamics model SALSA with a gas-phase chemistry module inside the ECHAM5 climate model by including a reference to this:

"The aerosol process models M7 (Vignati et al., 2004) and SALSA (Kokkola et al., 2008), partly owing to their computationally efficiency, have been implemented into the 3-D aerosol-climate model ECHAM5 (Bergman et al., 2012). SALSA is a sectional aerosol module, developed with the specific purpose for implementation in large scale models. It is part of the Hamburg Aerosol Model (HAM) (Stier et al., 2005) that handles the emissions, removal and microphysics of aerosol particles, and the gas-phase chemistry of dimethyl sulphide (DMS) within ECHAM5."

**3.        Section 2.3.5 "Dry deposition and wet scavenging of particles"**

In connection with the specific comment 4 of Referee #2, the dry deposition parameterization of Zhang et al. (2001; ZH2001) that is used in the SALSA model has been included in the comparison of dry deposition schemes. ZH2001 was implemented in the MAFOR v2.0 model and the dry deposition velocity of particles on rough urban surfaces was compared to the other dry deposition parameterizations that are included in the model. Figure 4 and the text belonging to this figure was revised to consider the dry deposition velocities calculated with the ZH2001 parameterization for urban rough surfaces. The comparison of the size-dependent deposition velocities obtained for the specific conditions was summarized:

"Size-dependent deposition velocities calculated with the SPF1985 and KS2012 schemes agree within a factor of two, except for large particles. Both curves have a minimum in the size range 0.2–0.5 μm diameter, while the curve from the ZH2001 scheme has a minimum at ~2 μm. For the HS2012 scheme, an upper limit value of the effective surface roughness length ($F^+ = 2.75$) was chosen, adequate for dry deposition to rough environmental surfaces, that results in higher deposition velocities for particles above 0.1 μm diameter compared to the other schemes. For particles in the size range between 0.01 and 0.5 μm, the calculated deposition velocities with HS2012 are nearly independent of particle size."

**4.      Section 4.1.2 "Importance of aerosol processes"**

Related to the same comment of Referee #2, the effect of using different dry deposition parameterizations on the model results in the evaluation of the "Urban Case" scenario has been analysed. To assess the differences in the model results due to the application of different deposition schemes (as given in Table 5), additional model runs including all processes for the "Urban Case" scenario were performed with the MAFOR model using SPF1985 (deposition scheme in AEROFOR), ZH2001 (deposition scheme in SALSA), and HU2012 (reference deposition scheme in MAFOR). The result of this analysis has been included in section 4.1.2:

"Differences in the relative contribution of deposition in the models are most probably due to the different schemes for dry deposition in the models (Table 5). To assess the differences in the model results due to the application of different deposition schemes, additional model runs including all processes were performed with the MAFOR model using first the deposition scheme in AEROFOR, SPF1985, and second the deposition scheme in SALSA, ZH2001. The comparison of the final particle size distribution at point D is shown in Figure E2 (Appendix E), obtained from MAFOR runs with different dry deposition parameterizations. The HU2012 deposition scheme that was used in the reference run with MAFOR was more efficient in removing particles >10 nm diameter than the other two deposition schemes. However, differences between using either the scheme SPF1985 or ZH2001 were negligible, which implies that the application of different dry deposition parameterizations was not the main reason for differences of the predicted particle size distributions."

The comparison of the final particle size distribution (at point D, after 78.5 s plume transport time) obtained from MAFOR runs with the different dry deposition parameterizations has been plotted in a new Figure E2, included in Appendix E.

**5.      New Section 4.2.3 "Consistent treatment of mass- and number-based concentrations of PM"**

A new section 4.2.3 "Consistent treatment of mass- and number-based concentrations of PM" has been added in the discussion part of the revised manuscript, in response to the main remark of Referee #1 and the specific comment 1 of Referee #2. In this new section, a discussion of the different aspects of the consistency in the treatment of mass- and number-based concentrations of particulate matter is presented. The discussion covers several aspects: 1) the initialization of the aerosol size distribution; 2) the insertion of particles from aerosol source emissions; 3) the mathematical solution of the aerosol dynamics processes; and 4) the comparability to observed PM mass concentrations and number concentrations.

In addition, the statement about the performance of MAFOR v2.0 in section 4.2.2 "Discussion of model performance" has been revised:

"Overall, the simulation of the "Urban Case" demonstrates the good performance of MAFOR v.2 in predicting particle number, size distribution and chemical composition of traffic exhaust aerosol. A major advantage of the model is the consistent treatment of particle number concentrations and mass concentrations of each aerosol component through the simultaneous solution of aerosol dynamics processes in terms of number and mass. This procedure allows the changes in the average density of particles to affect the predicted number and mass size distributions. An added value of the model is that

it can be used to determine the (order or magnitude) emission rate of SVOC by comparison between the modelled and the observed size distribution of total mass."

**6.    New section 4.2.4 "Evaluation of the model improvements"**

The main improvements of MAFOR version 2 (compared with the earlier version v1) were tested in three numerical experiments ("Case 1", "Case 2", and "Case 3") that are given in the Supplementary Materials. A new section 4.2.4 "Evaluation of the model improvements" has been included in the revised manuscript for a better presentation of the evaluation of the respective new developments. Text on the evaluation of the different new developments from section 2.3.3 "Coagulation" and section 2.4 "Dynamic partitioning of semi-volatile inorganic gases" was moved to this new section for better visibility.

In addition, we have performed an evaluation of the implemented new nucleation model of Määttänen et al. (M2018) in a simulation of new particle formation in comparison to the AEROFOR model (that also applies M2018). The numerical experiment allows for comparing the rate of ion-induced and neutral nucleation, formation and growth of new particles between the two models. The evolution of the modelled particle number size distribution in this numerical experiment was visualized in a new Figure 15. The results from the evaluation of the M2018 nucleation code in comparison against the AEROFOR model has been included in the new section 4.2.4 and in a new Appendix H.

**7.    Section 4.3.2 "Integration in 3-D atmospheric models"**

We addressed the specific comment 5 of Referee #2 about the selection of the fixed sectional method in view of implementation in large-scale atmospheric models in section 4.3.2:

"With regard to implementation of the aerosol dynamics code into large scale atmospheric models it is of special interest to assess how much one can lower the accuracy of the size distribution description without compromising on the accuracy of the model results. The evaluation of the sectional size representation in Case 1 (Supplement Sect. S2) revealed that the use of 16 size sections causes a numerical error of ~10%, and the use of 32 size sections causes only an error of ~3% in the final total PN concentrations under those conditions. The error of both representations is considered still acceptable when compared to measurement errors of observed total PN concentrations. Further, the computational demand increases only slightly when using a larger number of size sections. Overall, the size representation using 32 size sections is adequate for the simulation of long periods, as the accuracy in terms of size distribution changes and total number concentration is sufficiently high, while the computational demand is only 2% higher compared to the lowest tested resolution of 16 size sections.

Aerosol representations in large scale models are often limited to less than 20 size classes, as the particles in each size section have to be included in the advection routine and a higher number of advected species increases the computing time. Therefore, methods need to be developed for the mapping of the size representation used in the aerosol dynamics code and the advected particle species. The effect of changing the number of size classes in the 3-D model needs to be tested thoroughly."

**8.    Summary and Conclusions**

The paragraph about the main advantages of MAFOR v2.0 in the "Summary and Conclusions" has been deleted in the revised manuscript.

Further, the perspective of implementing MAFOR v2.0 in large-scale atmospheric models has been rephrased:

"The continued development of the open source code by the community is advised and steered by a consortium of aerosol scientists. Several aspects of the numerical solutions (efficient integration of number and mass concentrations, operator-splitting of processes, use of the fixed sectional method and low numerical diffusion) make the aerosol dynamics code a promising candidate for implementation into large scale atmospheric models."

**Tables:**

No changes have been made in the tables of the manuscript.

**Figures:**

Figure 4.

The plot of dry deposition velocity as function of particle diameter has been revised to include dry deposition velocities calculated with the ZH2001 parameterization for urban rough surfaces.

Figure 15.

The new Figure 15 illustrates the evolution of the modelled particle number size distribution in a 10-h simulation to compare the performance of the nucleation code M2018 in MAFOR v2.0 to that in AEROFOR.

Figure E2.

The new figure E2 in Appendix E shows the modelled particle number size distribution at point D (after 78.5 s plume transport time) obtained from simulations with MAFOR v2.0 for the "Urban Case" using different dry deposition parameterizations.

Figure H1.

The new Figure H1 in the new Appendix H compares several modelled parameters in a 10-h particle formation experiment with zero background particles (T = 267 K and RH = 90%) under clear sky conditions between MAFOR v2.0 and the AEROFOR model when the new binary nucleation parameterization M2018 is used.